# From short-term uncertainties to long-term certainties in the future evolution of the Antarctic Ice Sheet

Violaine Coulon [1,6] ✉, Ann Kristin Klose [2,3,6] ✉, Tamsin Edwards [4], Fiona Turner[4], Frank Pattyn [1] & Ricarda Winkelmann [2,3,5]

Robust projections of future sea-level rise are essential for coastal adaptation, yet they remain hampered by uncertainties in Antarctic ice-sheet projections–the largest potential contributor to sea-level change under global warming. Here, we combine two ice-sheet models, systematically sample parametric and climate uncertainties, and calibrate against historical observations to quantify Antarctic ice-sheet changes to 2300 and beyond. By 2300, the projected Antarctic sea-level contributions range from -0.09 m to +1.74 m under low emissions (SSP1-2.6, outer limits of 5-95% probability intervals), and from +0.73 m to +5.95 m under very high emissions (SSP5-8.5). Irrespective of the wide range of uncertainties explored, large-scale Antarctic ice-sheet retreat is triggered under SSP5-8.5, while reaching net-zero emissions well before 2100 strongly reduces multi-centennial ice loss. Yet, even under such strong mitigation, a significant sea-level contribution could still result from West Antarctica. Our results suggest that current mitigation efforts may not be sufficient to avoid self-sustained Antarctic ice loss, making emission decisions taken in the coming years decisive for future sea-level rise.

Observations over the past decades show that the Antarctic Ice Sheet is losing mass[1], mainly due to ocean-driven ice-sheet retreat in West Antarctica[2] and parts of East Antarctica[3,4], thereby contributing to global sea-level rise. The Sixth Assessment Report of the Intergovernmental Panel on Climate Change (IPCC-AR6) concluded that Antarctic mass loss is *likely* (66–100% outcome probability, see ref. 5 for definitions of likelihood terms and confidence levels) to continue throughout this century under all emission scenarios. By 2100, this contribution could amount to several tens of centimeters of sea-level rise, with the *medium confidence* projections ranging from −0.01 m to +0.41 m for Shared Socio-economic Pathway SSP1-2.6 and from 0.00 m to +0.57 m for SSP5-8.5 (5–95% percentile range). In a *low confidence* assessment that includes ice-sheet processes characterized by deep uncertainty[6], the upper bound may be higher[7,8].

The bulk of Antarctic mass loss, however, is expected beyond the end of this century[6], both because slowly-unfolding self-sustained ice loss may already have been triggered in recent decades[9,10] or could be initiated in the coming ones[11,12], and because changes in Antarctic climate are expected to intensify after 2100, particularly under higher-emission scenarios[6].

Yet, no projections of the Antarctic sea-level contribution over time are provided beyond 2150 in IPCC-AR6. Only the ice-sheet contribution at 2300 was assessed, ranging from −0.3 m to +3.1 m sea-level equivalent, and assigned *low confidence*, without any likelihood statement[6]. Given the substantial biophysical and socio-economic impacts of sea-level rise for low-lying coasts[13], this constitutes an important gap in global sea-level projections. The lack of confidence in multi-centennial Antarctic projections is because, at the time of IPCC-

[1]Université libre de Bruxelles (ULB), Laboratoire de Glaciologie, Brussels, Belgium. [2]Potsdam Institute for Climate Impact Research (PIK), Member of the Leibniz Association, Potsdam, Germany. [3]Department of Physics and Astronomy, University of Potsdam, Potsdam, Germany. [4]King's College London, Department of Geography, London, UK. [5]Integrative Earth System Science, Max Planck Institute of Geoanthropology, Jena, Germany. [6]These authors contributed equally: Violaine Coulon, Ann Kristin Klose. ✉e-mail: violaine.coulon@ulb.be; annkristin.klose@pik-potsdam.de

AR6, only a few individual studies had assessed Antarctic ice loss beyond 2100 under continued global warming, and these studies projected a wide range of outcomes (e.g., refs. 7,14,15). By contrast, most ice-sheet projections focused on shorter timescales, typically on the order of several decades[16–18].

Since IPCC-AR6, the Ice Sheet Model Intercomparison Project for CMIP6 (ISMIP6)–a multi-model ensemble–has provided extended projections of Antarctic sea-level contributions through 2300, with a main emphasis on very high emissions (SSP5-8.5 and the Representative Concentration Pathway RCP8.5)[19]. The focus on very high emissions[6,19] means that we face an imbalance between emission scenarios in sea-level projections, and lack estimates of the Antarctic sea-level contribution to be expected when constraining emissions.

Irrespective of the projection timescales considered, parametric uncertainties, for example, related to surface runoff, sub-shelf melt, or iceberg calving, remain largely unaccounted for in projections of Antarctic mass changes across multiple continental-scale ice-sheet models[20]. While some individual ice-sheet models have explored uncertainty in unconstrained parameters (e.g., refs. 7,10,21), existing multi-model intercomparisons typically include a wide range of ice-sheet models without consistently accounting for parametric uncertainty (e.g., refs. 16,19).

Furthermore, simulated ice-sheet changes over the past decades have rarely preceded Antarctic projections[16,19], even though historical trajectories have been suggested to influence the projected ice loss[22]. Covering the historical period would allow for conditioning projections of Antarctic mass changes on observations, addressing potential biases in the projected distribution of the ice-sheet contribution to sea-level rise[20]. So far, only a few studies have calibrated projections on the continental scale with transient satellite observations[7,23,24], and even fewer in a Bayesian framework[10,25].

To fill the gap in multi-centennial projections of Antarctic mass changes[26–28] and provide robust estimates of future sea-level change, we need to systematically account for as many sources of uncertainty as possible (such as future emissions, climate model, ice-sheet model structure and parametric uncertainty) while calibrating with past (observed) mass changes[20]. Here, we provide a robust assessment of the future evolution of the Antarctic Ice Sheet that, for the first time, jointly accounts for parametric, ice-sheet model structure, and climate model uncertainties in historically-calibrated projections through 2300 under a wide range of emissions, from low to very high. Future Antarctic climate trajectories to 2300 under the extended Shared Socio-Economic Pathways SSP1-2.6 and SSP5-8.5[29] are derived from four General Circulation Models (GCMs) from the sixth phase of the Coupled Model Intercomparison Project (CMIP6) and used to drive two ice-sheet models, with uncertainties in ice–climate interactions explored through a wide range of parameter perturbations. We quantify the contribution of these various sources of uncertainty to the projected range in ice-sheet changes using an analysis of variance, and assess the long-term consequences of the warming projected over the next centuries (that is, the committed ice-sheet response), by maintaining the changes in climate projected for 2300 through the end of the millennium.

## Results

### Future Antarctic climates

The Antarctic Ice Sheet may face a wide range of future climates over the coming centuries, depending on the emission pathway and strongly modulated by the climate model used (Fig. 1). Here, we base our analysis on a subset of CMIP6 GCMs–the UK Earth System Model (UKESM1-0-LL; ref. 30), the Institut Pierre-Simon Laplace global climate model (IPSL-CM6A-LR; ref. 31), the Community Earth System Model (CESM2-WACCM; ref. 32) and the Meteorological Research Institute Earth System Model (MRI-ESM2-0; ref. 33), which provide some of the few projections up to 2300 available in CMIP6 at the time

of this analysis[34]. These GCMs span a broad range of climate sensitivities, representative of the CMIP6 ensemble[35]. Despite significant improvements over previous generations, CMIP6 GCMs still show biases in Antarctic climate (e.g., refs. 36–38). For example, the upper Southern Ocean is generally too warm and fresh[36], and the Amundsen Sea Low remains poorly captured[37].

During the first half of this century, projected Antarctic atmospheric and oceanic warming is similar across the two emission scenarios. Divergence between the higher-emission (SSP5-8.5) and the lower-emission (SSP1-2.6) pathway emerges after 2050, eventually leading to significant changes in Antarctic climate on multi-centennial timescales (Fig. 1). Under SSP1-2.6, the four GCMs considered project a limited Antarctic-averaged atmospheric and oceanic temperature change of +1.0 °C – +3.6 °C and +0.2 °C – +0.7 °C, respectively, by 2300 compared to the 1995–2014 period. In contrast, the SSP5-8.5 scenario leads to substantial changes after the end of this century, with Antarctic-averaged atmospheric warming ranging from +12.0 °C to +17.0 °C and ocean warming from +1.7 °C to +3.1 °C by 2300 (Fig. 1).

Across the emission pathways, the projected changes in Antarctic climate over the next three centuries show similar characteristics for each CMIP6 GCM considered. Compared to the other three CMIP6 GCMs, MRI-ESM2-0 projects relatively slow and limited changes, especially under very high emissions (with Antarctic oceanic and atmospheric warmings of +1.7 °C and 12.2 °C, respectively, by 2300). Within each emission pathway, CESM2-WACCM predicts the strongest atmospheric warming of the four GCMs, along with a steady increase in circum-Antarctic ocean temperatures, reaching up to +0.7 °C under SSP1-2.6 and +2.0 °C under SSP5-8.5 by 2300. In contrast, projections by IPSL-CM6A-LR and UKESM1-0-LL show early and abrupt ocean warming in the first half of this century under both emission pathways. Under SSP5-8.5, these two GCMs yield the strongest multi-centennial Antarctic ocean warming, reaching +2.7 °C and +3.1 °C, respectively, by 2300.

### Bayesian calibration of the multi-model ensemble

To quantify the Antarctic contribution to sea-level rise and its associated uncertainties under these various future climates, we perform a 1400-member ensemble of simulations using two state-of-the-art ice-sheet models: Kori-ULB[10,39] and the Parallel Ice Sheet Model[40,41] (PISM; see Methods). Future Antarctic climates are derived from the four CMIP6 GCMs described above under the emission pathways SSP1-2.6 and SSP5-8.5 (Fig. 1). Key model parameters controlling the interaction of the ice sheet with the atmosphere and the ocean (Table 2) are varied using Latin Hypercube designs, which sample efficiently across the parameter space[42] (as used previously for ice-sheet projections by, for example, refs. 10,43). Our simulations begin in 1950 to allow, for each ice-sheet model, a Bayesian calibration with observations from the satellite era. Each ensemble member is attributed a weight by comparing the simulated ice-sheet evolution over the past decades with a series of mass balance estimates from the latest Ice Sheet Mass Balance Inter-comparison Exercise (IMBIE[1]; see Methods and ref. 10).

Through this calibration, the ice-sheet model ensembles are brought closer to the observed ice-sheet mass changes over the past decades (Supplementary Fig. 1a, b). This improved agreement between modeled and observed mass changes is quantitatively confirmed by a reduction in the Continuous Ranked Probability Score (CRPS) following the calibration (Supplementary Fig. 1e–l; see Methods for details). In addition, conditioning on the IMBIE record is effective in reducing the uncertainty in the hindcasts, especially for the Kori-ULB ensemble (Supplementary Fig. 1a) given the wider range of ice–ocean uncertainties explored (Table 2; see Methods).

Some regional biases in Antarctic mass changes, however, persist in the calibrated ensembles. For example, PISM simulations tend to underestimate the observed West Antarctic ice-sheet thinning (Supplementary Fig. 1d, k), whereas Kori-ULB trajectories capture this trend

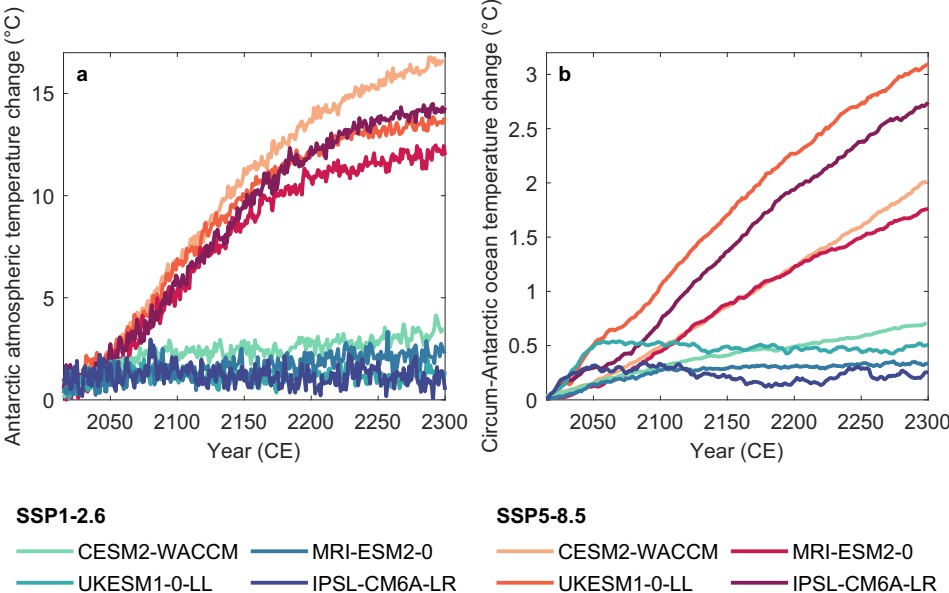

**Fig. 1 | Future Antarctic climates.** Future Antarctic climates projected by four CMIP6 GCMs (given by color) under emission pathways SSP1-2.6 and SSP5-8.5 (with respect to the time period 1995-2014). **a** Antarctic-averaged atmospheric temperature change. **b** Circum-Antarctic ocean temperature change, averaged over 400-800 m ocean depths.

more accurately (Supplementary Fig. 1c, g). Conversely, East Antarctic mass gain is slightly overestimated in the calibrated Kori-ULB ensemble compared to the IMBIE estimates (Supplementary Fig. 1f). These discrepancies between simulated and observed historical ice-sheet changes likely contribute to the overestimation of the ice-sheet net mass balance for both ice-sheet models, especially after 2000 (Supplementary Fig. 1a, b; see e.g., median of posterior distribution). Between 1992 and 2020, the calibrated median net mass balance amounts to $-77$ Gt yr$^{-1}$ ($-189$ Gt yr$^{-1}$ – $+1$ Gt yr$^{-1}$, 5–95% percentile range) for the Kori-ULB ensemble and to $-59$ Gt yr$^{-1}$ ($-157$ Gt yr$^{-1}$ – $+26$ Gt yr$^{-1}$) for the PISM ensemble, compared to the observed value of $-92 \pm 18$ Gt yr$^{-1}$[1] (see Supplementary Fig. 1e–l).

These regional discrepancies may lead to compensatory effects at the continental scale. For example, the calibration tends to favor PISM ensemble members that overestimate mass loss in the Antarctic Peninsula to offset the underestimation of West Antarctic mass loss. In contrast, for Kori-ULB, ensemble members that slightly overestimate West Antarctic mass loss are favored to counteract the model's bias towards mass gain in East Antarctica. Such compensations may reduce the ability of the calibration to fully capture regional dynamics, and can lead to biases in specific regions despite overall agreement at the continental scale (see Supplementary Fig. 1e–l and Methods).

## Projected Antarctic mass changes to 2300

Our multi-model ensemble supports emerging evidence suggesting little scenario-dependence of the Antarctic contribution to sea-level rise by 2100[10,17,24] (Fig. 2a, zoom-in). In particular, in the coming decades, ice-sheet trajectories under both emission pathways substantially overlap, with projected sea-level contributions ranging between $-0.03$ m and $+0.33$ m for SSP1-2.6 and between $-0.05$ m and $+0.37$ m for SSP5-8.5 by 2100 (Table 1). On multi-decadal timescales, mass loss remains confined to West Antarctica as well as the Antarctic Peninsula and is offset to some extent by East Antarctic mass gain (Table 1, Supplementary Fig. 2).

After 2100, the projected Antarctic sea-level contribution under both emission pathways starts to diverge, potentially leading to multi-meter ice loss under very high emissions. By 2300, mass loss under SSP5-8.5 ranges from $+0.73$ m to $+5.95$ m sea-level equivalent (Fig. 2a, Table 1), primarily driven by a significant grounding-line retreat in the

Amundsen Sea Embayment, the Siple Coast, and Weddell Sea regions (Fig. 3c, d). In addition, marine parts of the East Antarctic Ice Sheet, such as the Wilkes and Recovery catchments, potentially lose mass by 2300 (Fig. 3c, d). Multi-centennial ice loss from these East Antarctic marine catchments is only partially offset by accumulation in the interior of East Antarctica, with the lower bound of the 5–95% probability interval reaching less than $-0.20$ m sea-level equivalent by 2300 under SSP5-8.5 for both ice-sheet models (Table 1, Supplementary Fig. 2c, d).

When strongly constraining emissions according to SSP1-2.6, the projected Antarctic contribution to sea-level rise remains below $+1.75$ m by 2300 with 95% probability across both ice-sheet models (Fig. 2a, Table 1). Compared to SSP5-8.5, mass loss is substantially reduced due to a lower likelihood of extensive grounding-line retreat in West Antarctica. Nonetheless, the resulting sea-level rise may still require major coastal adaptation efforts. In East Antarctica, mass loss may be avoided by 2300 when following SSP1-2.6, with increased snow accumulation dominating the regional signal and leading to a negative median contribution to sea level for both ice-sheet models (Table 1, Supplementary Fig. 2c, d).

By 2100, our range of the projected Antarctic sea-level contribution is consistent with the IPCC-AR6 assessment[6] (based on their *medium confidence* projections; Fig. 2a), although our upper bound under SSP5-8.5 is slightly lower. The higher IPCC-AR6 upper bound reflects the LARMIP-2 estimates[18], which focus on the response to sub-shelf melting under high melt sensitivities representative of the Amundsen Sea Embayment[44,45]. By 2300, however, our historically calibrated ensembles of simulations yield a higher upper bound than IPCC-AR6[6] ($+4.10$ – $+4.40$ m versus $+3.10$ m in IPCC-AR6, 83rd percentiles; Fig. 2a) and than ISMIP6 under very high emissions[19] ($+5.10$ – $+5.95$ m, 95th percentiles, versus $+4.40$ m, the upper end of ISMIP6 model range). The higher upper bounds presented here and in ISMIP6 likely reflect the application of a range of extended CMIP6 climate trajectories beyond 2100, which were not considered in IPCC-AR6. The sampling of both ice-sheet model structure and wide parametric uncertainties explains our ensembles' higher upper bounds compared to ISMIP6, despite being constrained by observed mass changes. Yet, even with the strong SSP5-8.5 warming trajectories projected by some of the GCMs considered here, our estimates remain well below the

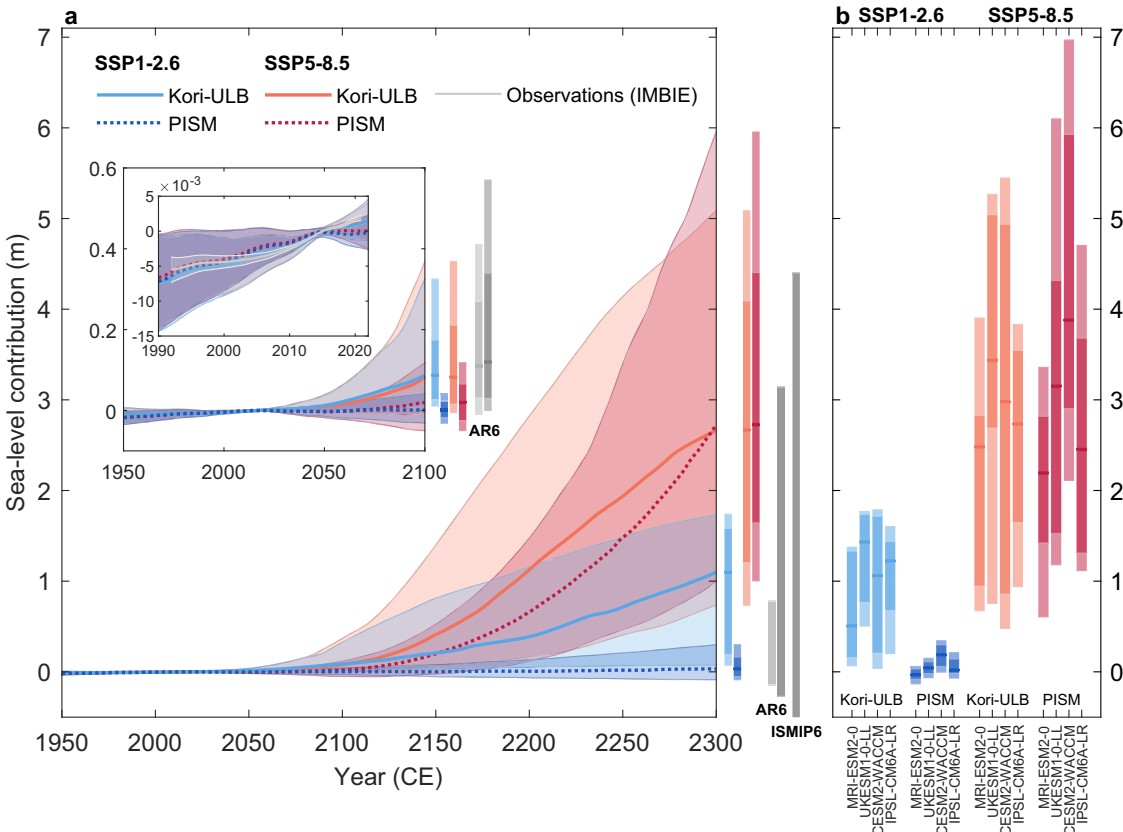

**Fig. 2 | Future sea-level contribution from the Antarctic Ice Sheet through 2300.** Future sea-level contribution from the Antarctic Ice Sheet through 2300 (in meters sea-level equivalent) in response to changes in Antarctic climate projected by four CMIP6 GCMs under emission pathways SSP1-2.6 (light/dark blue) and SSP5-8.5 (orange/red). Lines and shaded areas (in **a**) show the medians and 5-95% probability intervals of the calibrated probabilistic sea-level projections by the ice-sheet models Kori-ULB (solid lines and/or lighter colors) and PISM (dotted lines and/or darker colors). Two zoom-ins highlight (i) the period 1950-2100, and (ii) the

historical period 1990-2020, where observations from IMBIE[1] are shown for comparison (light gray line and shading). Boxes show the [5,17,50,83,95]th percentiles for the years 2100 and 2300. For comparison, IPCC-AR6 estimates[6] of the Antarctic contribution to sea-level change by 2100 ([5,17,50,83,95]th percentiles) and by 2300 (17–83% percentile ranges) are given in light (SSP1-2.6) and dark (SSP5-8.5) gray. An additional error bar at 2300 indicates the SSP5-8.5 ISMIP6 model range[19]. Boxes in **b** show [5,17,50,83,95]th percentiles of the Antarctic sea-level contribution, depending on the four CMIP6 GCMs and the ice-sheet models.

range projected by a single ice-sheet model that considers the Marine Ice Cliff Instability (MICI; +6.87 m − +13.55 m sea-level equivalent under RCP8.5[7]). These projections sampled only a limited subset of MICI-related uncertainty and did not account for broader parametric or climate model uncertainties, which would likely alter these ranges[46].

**Role of uncertainties in modulating future Antarctic mass loss**
Projected Antarctic mass loss is shaped by a shifting interplay between ice-sheet model structures, parameters governing the interactions with the ocean and the atmosphere, and diverging GCM climate trajectories. Structural differences between the ice-sheet models and parameters modulating sub-shelf melting dominate multi-decadal uncertainty. As changes in Antarctic climate intensify and diverge across emission scenarios, parametric uncertainty related to ice–atmosphere interactions and climate model uncertainty increasingly contribute to the spread in the projected Antarctic ice-sheet trajectories. Under very strong warming (SSP5-8.5), this shift in the dominant sources of uncertainty leads to a robust agreement across the ice-sheet models on a multi-meter sea-level contribution by 2300.

To quantify the relative contribution of these different sources of uncertainty in our ensembles, we perform a variance decomposition using an analysis of variance (ANOVA) framework (see Methods; Figs. 4 and 5). This approach attributes the spread in the projected Antarctic sea-level contribution to parameters governing the interaction of the ice sheet with the atmosphere and ocean, the choice of climate model, the ice-sheet model structure, and their two-way

interactions (the combined effect of two sources of uncertainty acting together).

In line with previous findings[19,47], the choice of ice-sheet model is a major source of uncertainty in Antarctic sea-level projections (Fig. 4), highlighting the influence of differences in model physics (e.g., basal friction laws, calving schemes; see Supplementary Table 1), numerical methods, and initialization approaches. Over the next decades, the influence of this structural uncertainty on the projected ice-sheet trajectories increases under both emission scenarios, with its main effect accounting for up to 40% of the total variance at the continental scale (Fig. 4a, b). This fraction is smaller than in previous assessments[19,47], likely because of differences in experimental design: our setup explores a wider range of parametric uncertainty but includes only two ice-sheet models, whereas refs. 19,47 sampled fewer parameters but a larger and more diverse set of models. In those earlier studies, parametric uncertainty was implicitly embedded in the structural uncertainty, whereas here both sources of uncertainty are explicitly quantified, along with their interactions. Our results suggest that the interactions between parametric and structural uncertainties significantly contribute to the total variance during the next few decades, adding to the main effect of the ice-sheet model structures. The uncertainty in the ice-sheet model structure is illustrated by the systematically higher Antarctic sea-level contribution projected by Kori-ULB compared to PISM throughout this century (Fig. 2a, zoom-in; Table 1). Under SSP1-2.6, PISM projects a zero median Antarctic sea-level contribution by 2100, with a 5-95% range of −0.03 m to +0.04 m.

**Table 1 | Projected sea-level contribution from the Antarctic Ice Sheet on different timescales, depending on future emissions**

| | Kori-ULB | | | | PISM | | | |
|---|---|---|---|---|---|---|---|---|
| | 2100 | 2300 | 2500 | 3000 | 2100 | 2300 | 2500 | 3000 |
| **Antarctic Ice Sheet** | | | | | | | | |
| SSP1-2.6 | 0.09 (0.03,0.17) [0.01, 0.33] | 1.10 (0.19, 1.58) [0.07, 1.74] | 1.65 (0.99, 2.21) [0.41, 2.31] | 2.04 (1.32, 2.84) [0.85, 3.41] | 0.00 (−0.02, 0.02) [−0.03, 0.04] | 0.03 (−0.05, 0.16) [−0.09, 0.30] | 0.10 (−0.04, 0.51) [−0.14, 1.00] | 0.81 (0.27, 3.34) [0.02, 5.28] |
| SSP5-8.5 | 0.08 (0.02, 0.21) [−0.01, 0.37] | 2.67 (1.20, 4.09) [0.73, 5.09] | 4.04 (2.81, 6.62) [1.44, 8.42] | 6.14 (3.72, 12.84) [1.88, 17.53] | 0.02 (−0.03, 0.07) [−0.05, 0.12] | 2.73 (1.64, 4.40) [1.00, 5.95] | 7.74 (5.00, 10.23) [3.39, 13.14] | 13.57 (10.35, 18.73) [7.81, 25.85] |
| **West Antarctic Ice Sheet** | | | | | | | | |
| SSP1-2.6 | 0.09 (0.05, 0.18) [0.03, 0.36] | 1.07 (0.28, 1.54) [0.16, 1.71] | 1.78 (0.93, 2.00) [0.38, 2.11] | 2.18 (1.44, 2.56) [1.26, 2.74] | 0.00 (−0.01, 0.01) [−0.01, 0.02] | 0.03 (0.00, 0.11) [−0.01, 0.19] | 0.12 (0.03, 0.39) [0, 0.84] | 0.73 (0.28, 2.70) [0.16, 4.60] |
| SSP5-8.5 | 0.10 (0.05, 0.22) [0.03, 0.44] | 2.29 (1.02, 2.96) [0.69, 3.19] | 3.18 (2.52, 3.60) [1.73, 3.85] | 3.63 (3.38, 4.24) [2.76, 4.41] | 0.00 (−0.01, 0.01) [−0.02, 0.03] | 1.39 (0.90, 2.71) [0.64, 3.58] | 4.47 (2.80, 5.47) [2.20, 5.86] | 6.16 (5.67, 6.67) [5.27, 6.91] |
| **East Antarctic Ice Sheet** | | | | | | | | |
| SSP1-2.6 | −0.02 (−0.03, 0.00) [−0.04, 0.03] | −0.06 (−0.11, 0.07) [−0.14, 0.20] | −0.10 (−0.20, 0.21) [−0.25, 0.32] | −0.19 (−0.41, 0.39) [−0.49, 0.82] | −0.02 (−0.03, 0.00) [−0.03, 0.00] | −0.06 (−0.10, −0.01) [−0.13, 0.05] | −0.07 (−0.15, 0.02) [−0.19, 0.11] | 0.01 (−0.19, 0.36) [−0.28, 0.67] |
| SSP5-8.5 | −0.03 (−0.05, −0.01) [−0.06, 0.01] | 0.24 (−0.03, 0.99) [−0.15, 1.66] | 0.64 (−0.10, 2.79) [−0.32, 4.50] | 2.24 (0.09, 8.28) [−0.79, 12.78] | −0.02 (−0.04, 0.00) [−0.05, 0.02] | 0.68 (0.20, 1.55) [−0.04, 2.33] | 2.41 (1.09, 4.22) [0.38, 6.96] | 6.59 (3.67, 11.17) [1.89, 18.05] |
| **Antarctic Peninsula** | | | | | | | | |
| SSP1-2.6 | 0.00 (0.00, 0.01) [−0.01, 0.02] | 0.00 (−0.01, 0.03) [−0.01, 0.03] | 0.00 (−0.01, 0.03) [−0.02, 0.03] | 0.01 (−0.01, 0.03) [−0.01, 0.04] | 0.02 (0.01, 0.03) [0.01, 0.03] | 0.05 (0.02, 0.08) [0.01, 0.11] | 0.07 (0.03, 0.11) [0.01, 0.17] | 0.13 (0.04, 0.21) [0.01, 0.27] |
| SSP5-8.5 | 0.01 (0.00, 0.02) [−0.01, 0.02] | 0.14 (0.11, 0.23) [0.07, 0.26] | 0.19 (0.14, 0.28) [0.11, 0.29] | 0.21 (0.16, 0.30) [0.12, 0.31] | 0.04 (0.02, 0.05) [0.01, 0.07] | 0.45 (0.31, 0.60) [0.25, 0.70] | 0.67 (0.51, 0.82) [0.43, 0.85] | 0.86 (0.72, 0.99) [0.66, 1.01] |

Ice sheet changes (in meters sea-level equivalent) on the continental scale and for different Antarctic Ice Sheet regions (West Antarctic Ice Sheet, East Antarctic Ice Sheet, and Antarctic Peninsula) under emission pathways SSP1-2.6 and SSP5-8.5 as determined by the ice-sheet models Kori-ULB and PISM. Given are the medians, (17%-83%) and [5%–95%] probability intervals, projected by 2100 and 2300 next to the committed ice loss by 2500 and 3000, compared to 2015.

Under SSP5-8.5, the median increases slightly to +0.02 m (-0.05 m to +0.12 m). In contrast, Kori-ULB projects substantially higher estimates: +0.09 m (+0.01 m to +0.33 m) under SSP1-2.6 and +0.08m (-0.01 m to +0.37 m) under SSP5-8.5. This ice-sheet model divergence mainly stems from a stronger dynamic ice loss in West Antarctica in the projections by Kori-ULB than PISM (Supplementary Fig. 2a, b), as a continuation of the trends simulated by each ice-sheet model over the historical period (Supplementary Fig. 1). While the ice-sheet model structure is the dominant source of uncertainty in West Antarctica and the Antarctic Peninsula throughout this century (Fig. 4c, e, f, h), in East Antarctica, it becomes negligible by the end of this century under both emission scenarios (Fig. 4d, g). In this region, both ice-sheet models consistently project a slight mass gain by 2100 (Table 1, Supplementary Fig. 2c, d).

Beyond 2100, the evolution of the structural uncertainty diverges between the two emission scenarios. Under the SSP1-2.6 warming trajectory, it continues to increase, reaching 50% of the total variance at the continental scale by 2300 (Fig. 4a). In contrast, under SSP5-8.5, the strong atmospheric and oceanic changes projected over Antarctica progressively override the trends inherited from the historical period. Consequently, structural uncertainty decreases and eventually becomes negligible on the continental scale by 2300. At the same time, the contribution of parameters related to ice–atmosphere interactions increases, especially in East Antarctica. By 2300, these parameters become the main source of uncertainty in the projected ice-sheet response under SSP5-8.5, accounting for 31% of the total variance (Fig. 4b). This reflects the growing influence of surface melt and runoff in a warming climate[10]. The shift in the dominant sources of uncertainty for this very strong warming scenario leads to robust agreement between the two ice-sheet models on a substantial Antarctic sea-level contribution by 2300 (Fig. 2a). Not only do we find a similar median ice loss of +2.67 m and +2.73 m sea-level equivalent in Kori-ULB and PISM, respectively, but also the 5–95% probability intervals are similar (Table 1). The increasing dominance of the climate pathway over the ice-sheet model structure is also particularly evident when explicitly including the emission pathways in the ANOVA (Supplementary Fig. 3).

Across both emission scenarios, the parameters controlling ice–ocean interactions play a key role in driving divergent mass loss trajectories, accounting for up to about 35% of the variance at the continental scale over the next three centuries (Fig. 4). This signal is largely driven by the Kori-ULB ensemble, where parametric uncertainty in ice–ocean interactions dominates the variance in the projected ice-sheet changes, except under very strong warming (SSP5-8.5 beyond 2250; Fig. 5a, b). In contrast, parameters associated with ice-atmosphere interactions consistently dominate the variance in the PISM ensemble (Fig. 5c, d). This difference in the dominant source of uncertainty between both ice-sheet models likely arises from differences in their ensemble designs: the Kori-ULB ensemble, by sampling a variety of sub-shelf melt parameterizations and associated parametric uncertainty in the effective ice–ocean heat flux, is more sensitive to ocean warming (in particular, to the early increase in circum-Antarctic ocean temperatures projected by UKESM1-0-LL and IPSL-CM6A-LR even under SSP1-2.6; Fig. 1). By comparison, the PISM ensemble relies on a single sub-shelf melt parameterization (the Potsdam Ice-Shelf Cavity mOdel, PICO[48]; see Methods), making it less sensitive to ocean warming. This difference in the sensitivity of the two ice-sheet models ensembles to changes in the ocean is also evident in the contribution of the two-way interaction between ice-sheet model and ocean-related parameters, which quantifies the divergent model responses to ocean forcing. This interaction term specifically captures the influence of the choice of sub-shelf melt parameterization on projected ice-sheet mass changes as well as the ice-sheet model sensitivity resulting from the initialization[49], which were hidden within the structural uncertainty in previous assessments[19,47]. These findings highlight the importance of sampling a range of sub-shelf melt parameterizations and associated parametric uncertainties in systematic ensemble designs, next to continued developments in the representation of ocean-induced melting in ice-sheet models. Similarly, the difference in the sensitivity of the two ice-sheet model ensembles to atmospheric forcing is captured by the two-way interaction between the ice-sheet model and atmosphere-related parameters, which emerges as a significant contributor to the total variance at the continental scale under very strong warming.

Beyond ice-sheet model structure and parametric uncertainties, the trajectory of future Antarctic ice loss is also strongly modulated by the choice of GCM, which explains up to 17% of the total variance in the Antarctic sea-level contribution by 2300, consistent with previous estimates[19]. Especially in East Antarctica, the GCM divergence beyond 2100 (Fig. 1) represents an important source of uncertainty (Fig. 4d, g). This influence is clearly reflected in the distributions of the Antarctic

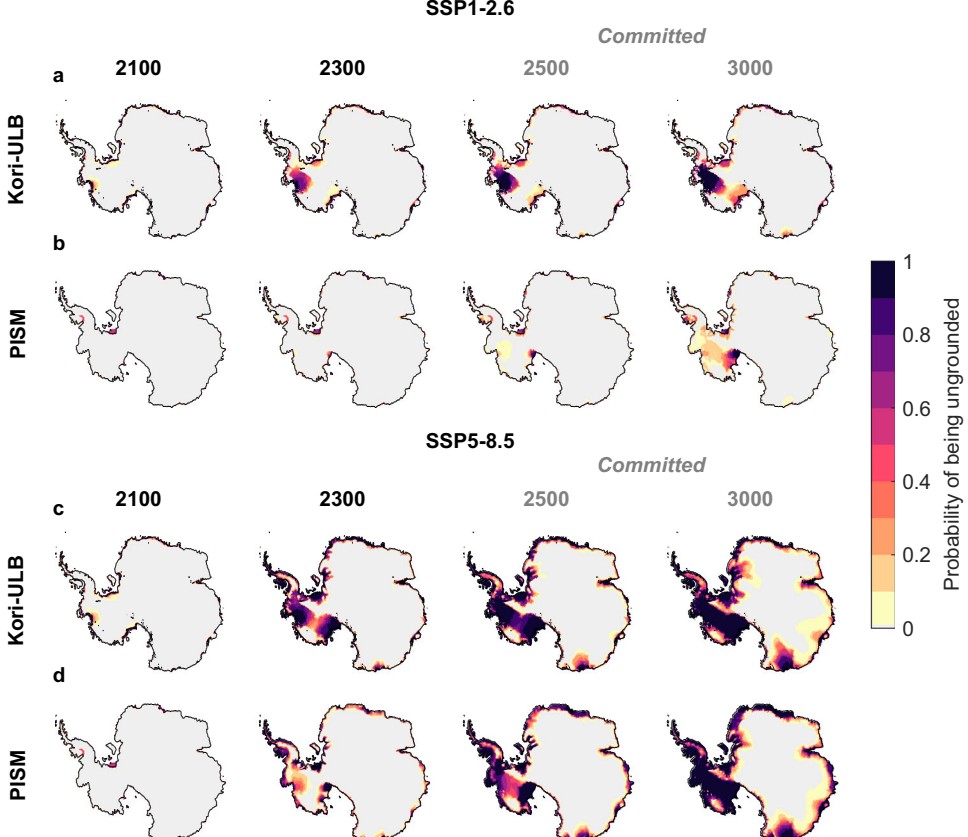

**Fig. 3 | Evolution of the Antarctic Ice Sheet over the next centuries up to the end of the millennium.** Evolution of the Antarctic Ice Sheet over the next centuries up to the end of the millennium under emission pathways SSP1-2.6 (**a**, **b**) and SSP5-8.5 (**c**, **d**) as determined by the ice-sheet models Kori-ULB (**a**, **c**) and PISM (**b**, **d**) by 2100 and 2300 as well as 2500 and 3000 (committed). Shown is the probability of being ungrounded at these different points in time throughout this millennium. Gray regions correspond to locations where there is a 0% probability of being ungrounded.

sea-level contribution by 2300 for the different GCMs, which in some cases do not even overlap in their 17–83% probability intervals (Fig. 2b; comparing, for example, distributions under MRI-ESM2-0 and UKESM1-0-LL). For example, the limited changes in Antarctic climate projected by MRI-ESM2-0 result in consistently lower ice-sheet sea-level contributions by 2300 in both the Kori-ULB and PISM ensemble (Fig. 2b). In contrast, the strong Antarctic atmospheric warming projected by CESM2-WACCM (Fig. 1a) amplifies Antarctic mass loss[10], leading to a multi-meter sea-level contribution under SSP5-8.5 emissions for both ice-sheet models (Fig. 2b). The early and abrupt ocean warming characteristic of UKESM1-0-LL and IPSL-CM6A-LR (Fig. 1b) triggers high sea-level contributions compared to the other GCMs in the ocean-sensitive Kori-ULB ensemble, even under SSP1-2.6. In the PISM ensemble, however, these same GCMs lead to lower sea-level contributions than for CESM2-WACCM due to the ensemble's atmospheric rather than oceanic sensitivity (Fig. 5).

**Certain long-term future of the Antarctic Ice Sheet**
Irrespective of the wide range of future Antarctic climates as well as the uncertainties in the ice-sheet model structure and parameters sampled in the ensembles, our projections reveal certainties for the long-term evolution of the Antarctic Ice Sheet to the year 3000: a collapse of the West Antarctic Ice Sheet under very high emissions, and substantially reduced sea-level contribution under low emissions.

Under the higher-emission pathway SSP5-8.5, a collapse of the West Antarctic Ice Sheet is triggered across our entire ensemble of simulations. This collapse could already unfold by 2300, is *likely* (66–100% outcome probability, see ref. 5 for definitions of likelihood terms and confidence levels followed here) by 2500 and becomes

*virtually certain* by 3000, as a committed response to the projected SSP5-8.5 changes in Antarctic climate (Fig. 3c, d). Such ice loss results in a West Antarctic contribution to sea-level rise ranging from +2.76 m to +6.91 m by 3000 (Table 1).

Self-sustained ice loss is also triggered in East Antarctica under the strong SSP5-8.5 warming trajectory, leading to *very likely* long-term ice-sheet retreat (Fig. 3c, d). This includes both ocean-driven grounding-line retreat in marine catchments and potential ice loss from regions grounded above sea level, amplified by the surface melt-elevation feedback[10]. As a result, the East Antarctic contribution to sea-level rise by 3000 shows a wide spread, ranging from −0.79 m to +18.05 m (Table 1). Our historically-calibrated projections, linking observed and projected changes to long-term dynamics, consistently indicate that grounding-line retreat in the East Antarctic Wilkes catchment is more likely than in the Aurora catchment under SSP5-8.5 (Fig. 3c, d). This suggests that the currently observed ocean-driven mass loss of Totten Glacier[4] (also captured by Kori-ULB during the historical period; Supplementary Fig. 1) does not necessarily imply a substantial long-term retreat. In contrast, even though the Wilkes catchment currently shows limited mass changes[50], grounding lines in this region are *very likely* to retreat over the long term in response to early-millennium SSP5-8.5 warming, consistent with the distinct glaciological and topographic settings of these catchments[51].

Under the lower-emission pathway SSP1-2.6, a long-term collapse of the West Antarctic Ice Sheet by 3000 cannot be ruled out, but is considerably less likely (Fig. 3a, b). Uncertainty arises from the Amundsen Sea Embayment, where the likelihood of retreat ranges from *about as likely as not* (PISM) to *virtually certain* (Kori-ULB), reflecting the trends projected by the ice-sheet models through 2300

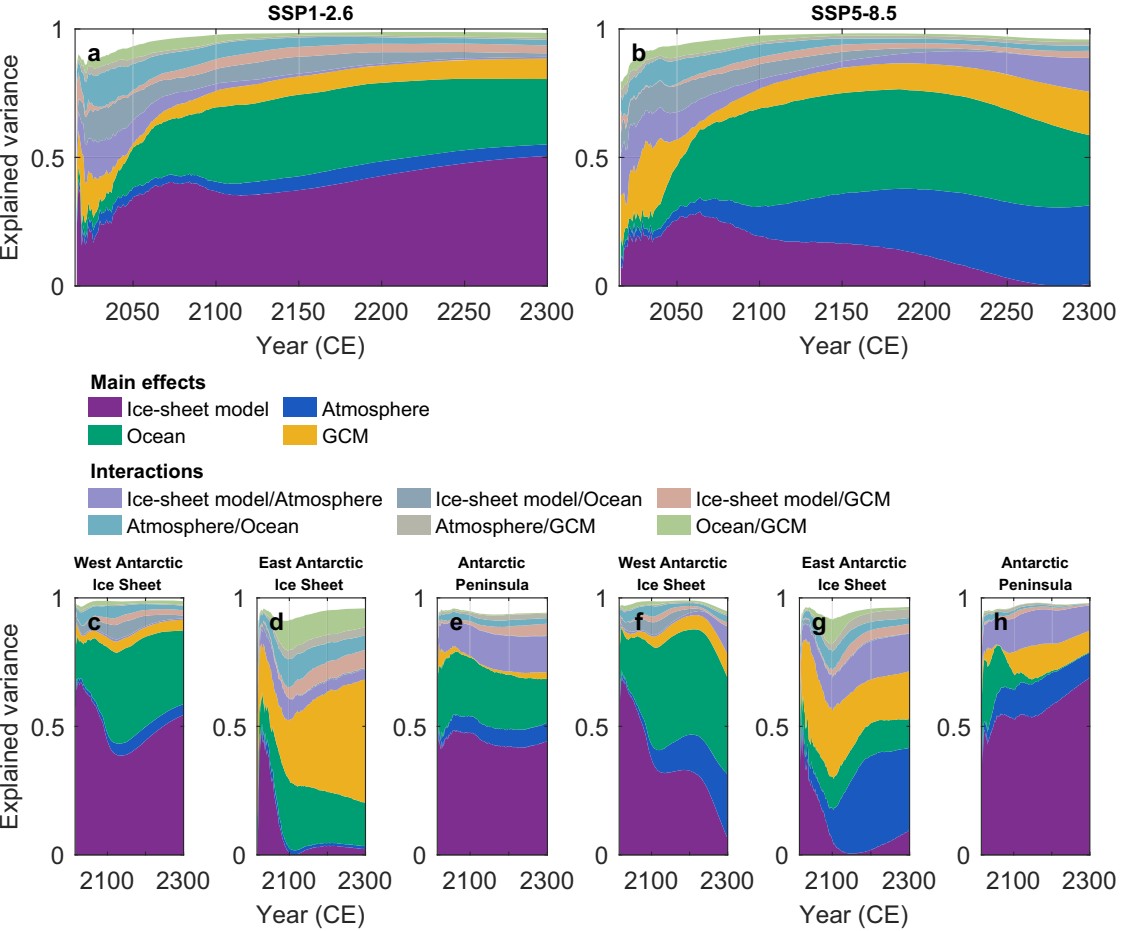

**Fig. 4 | Sources of uncertainty in the future sea-level contribution from the Antarctic Ice Sheet through 2300.** Explained variance (i.e., the fraction of total variance) in the Antarctic ice-sheet contribution to global mean sea level attributed to main effects of the ice-sheet model (purple), ocean-related parameters (green), atmosphere-related parameters (blue), CMIP6 GCMs (yellow), and their two-way interactions (lighter colors), based on ANOVA for the combined Kori-ULB and PISM ensembles. Each colored area represents the fraction of total ensemble variance explained by a given source of uncertainty or interaction over time. Fractions from all individual atmospheric and ocean-related parameters are aggregated into their respective categories. The white space above the stacked areas represents variance contributions from additional two-way and higher-order interaction terms that are not shown here, as well as residual unexplained variance. Results are shown for the Antarctic Ice Sheet (**a**, **b**), the West Antarctic Ice Sheet (**c**, **f**), the East Antarctic Ice Sheet (**d**, **g**), and the Antarctic Peninsula (**e**, **h**) under emission pathways SSP1-2.6 (**a**, **c**–**e**) and SSP5-8.5 (**b**, **f**–**h**).

(see previous section). In East Antarctica, constraining emissions under the SSP1-2.6 pathway largely prevents substantial self-sustained ice loss at least until 3000 (Fig. 3a, c), although modest mass loss could still contribute to a sea-level contribution of up to half a meter in both Kori-ULB and PISM (Table 1).

Overall, while very high SSP5-8.5 warming could potentially lead to substantial Antarctic ice loss of up to +25.85 m sea-level equivalent by 3000, following the SSP1-2.6 scenario limits the long-term Antarctic mass loss to below +5.28 m with a 95% probability (Table 1). Note that the spread in the Antarctic sea-level contribution over multi-centennial timescales (with higher amplitudes projected by PISM compared to Kori-ULB; Table 1) is partly due to differences in the simulated magnitude of the water-expulsion effect[52] (see Methods).

## Discussion
### Remaining uncertainties in the timing of ice loss
We provide a comprehensive assessment of the evolution of the Antarctic Ice Sheet to 2300 and up until the end of this millennium under a wide range of future climates by combining two ice-sheet models, systematically sampling parametric uncertainties, and historically calibrating against observations. While IPCC-AR6 assigned *low confidence* to the assessed multi-centennial Antarctic ice-sheet response[6],

the probabilistic projections presented here provide robust estimates of the Antarctic contribution to sea-level rise, which are needed to support long-term policy and decision-making with respect to global emissions mitigation.

Our approach also allows us to identify aspects of the long-term evolution of the Antarctic Ice Sheet for which we have *high confidence*: very strong early-millennium warming triggers substantial, self-sustained ice loss in both West and East Antarctica, with the long-term collapse of the West Antarctic Ice Sheet being *virtually certain*. In contrast, reaching net-zero emissions well before 2100 limits the long-term Antarctic contribution to sea level by avoiding substantial ice loss from East Antarctica over this millennium. However, even under such strongly constrained greenhouse gas emissions, widespread West Antarctic ice-sheet retreat cannot be fully excluded. Under continued current climate policies, close to a middle-of-the-road scenario (SSP2-4.5[5]; not assessed here due to the lack of projections beyond 2100[34]), the likelihood of significant West Antarctic ice loss is expected to increase. Assessing this wide range of emissions–from low to very high–suggests that some sea-level contribution from the Antarctic Ice Sheet requiring adaptation may be unavoidable, even under low emissions, and that this lower bound needs to be further constrained.

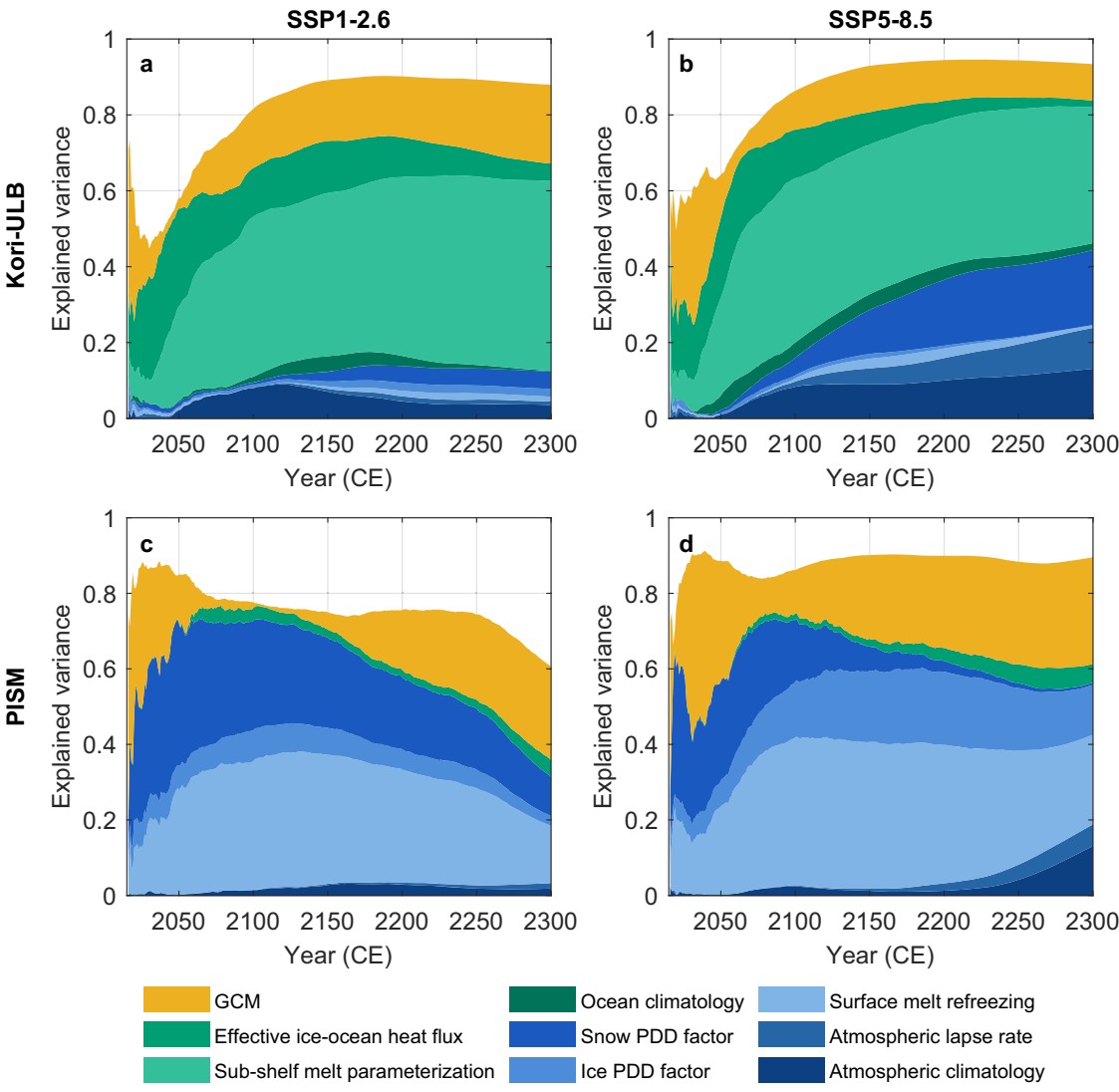

**Fig. 5 | Ice-sheet model specific sources of uncertainty in future sea-level contribution from the Antarctic Ice Sheet through 2300.** Explained variance (i.e., fraction of total variance) in the Antarctic ice-sheet contribution to global mean sea level attributed to main effects of CMIP6 GCMs (yellow), ocean-related parameters (green) and atmosphere-related parameters (blue), based on ANOVA for the ice-sheet models Kori-ULB (left column, **a**, **c**) and PISM (right column, **b**, **d**) under emission pathways SSP1-2.6 (**a**, **c**) and SSP5-8.5 (**b**, **d**). Each colored area represents the fraction of total ensemble variance explained by a given source of uncertainty or interaction over time. The white space above the stacked areas represents variance contributions from additional two-way and higher-order interaction terms that are not shown here, as well as residual unexplained variance. Note that only one ocean-related parameter (i.e., the effective ice--ocean heat flux) is included in the PISM ensemble (see Table 2).

In particular, while our results indicate that the long-term fate of the ice sheet is strongly determined by the emission scenario, uncertainties remain regarding the timing and pace of multi-decadal to centennial sea-level rise–that is, *when* and *how fast* a given amount of sea-level change may be exceeded (not *if*[53,54]). Reliable projections of the onset of collapse and the future rates of mass loss remain limited by uncertainties in both ice-sheet and climate models. In our ensemble, we identify a shift in the dominant sources of uncertainty over time: initially, the rate of ice-sheet retreat is mainly controlled by ice-sheet model structures, their distinct historical trends, and parameters controlling sub-shelf melting. This also reflects differences in ice-sheet model initialization, which here relies on two distinct approaches (see Methods) and is known to strongly influence the projected ice-sheet response, especially over the short term[12,49]. By contrast, from 2100 onwards, the emission scenario progressively emerges as the most significant uncertainty factor (see Supplementary Fig. 3), highlighting the importance of better sampling of SSPs in multi-centennial climate model projections. In the case of very strong warming, the external forcing dominates the ice-sheet response, overwriting the influence of

the ice-sheet model structure, and parameters controlling surface melt and runoff become the key source of uncertainty.

Due to computational constraints, not all uncertainties could be explored here. The timing of large-scale ice loss may also depend on dynamical processes not captured in our ensemble, such as damage[55] and hydrofracturing. These could promote earlier ice-shelf breakup[56,57] and potentially initiate MICI mechanisms[58,59], accelerating long-term West Antarctic Ice Sheet collapse, which is already suggested as *virtually certain* under very high emissions in our simulations even without MICI. However, these processes are difficult to constrain, observations are sparse (if they exist at all), and they are seldomly represented in large-scale ice-sheet models[7,14,60]. At the same time, grounding-line retreat in West Antarctica may be delayed by the faster viscoelastic response expected in this region due to weaker solid Earth structure[61,62]. In addition, we note that our ensemble uses a horizontal resolution of 16 km, which may affect the representation of narrow basin entry channels, such as those in the Wilkes and Aurora catchments, and consequently influence the timing of grounding-line retreat. Including such additional processes or uncertainties (e.g.,

related to sliding[63] and subglacial hydrology[64,65]) could not only shift or broaden the projected range of the Antarctic sea-level contribution, but also accelerate or delay ice-sheet retreat.

### Next decades decisive for Antarctica's long-term future

Much attention has recently been focused on uncertainties stemming from the dynamic sea-level contribution from Antarctica (e.g., ref. 6). Our study shows that substantial long-term Antarctic changes–some of which may be irreversible on human timescales[66]–may occur even if massive mitigation efforts are made. These risks should be considered in policy-making for both global emissions and coastal adaptation. In particular, a long-term collapse of the West Antarctic Ice Sheet appears unavoidable under very high emissions. Even if emissions are strongly constrained by reaching net zero before the end of this century, the Antarctic Ice Sheet may still contribute a limited but significant amount to sea-level rise over this millennium. These findings suggest that ongoing mitigation efforts may not be sufficient to prevent self-sustained Antarctic ice loss, making decisions on emissions reductions taken in the coming years and decades decisive for multi-centennial sea-level rise.

## Methods

### Ice-sheet models

The projections of Antarctic mass changes presented here have been carried out with the standalone state-of-the-art ice-sheet models Kori-ULB[10,39] and PISM[40,41]. All simulations were performed at a 16 km spatial resolution. The main characteristics of both ice-sheet models are summarized in Supplementary Table 1.

**Kori-ULB.** Kori-ULB is a vertically-integrated, thermomechanical, hybrid ice-sheet/ice-shelf model that combines the shallow ice (SIA) and shallow shelf (SSA) approximations for grounded ice, while only the shallow shelf approximation is applied for floating ice shelves to represent ice flow[40,41]. All Kori-ULB projections are based on the model setup described in ref. 10. To represent grounding-line migration, a flux condition based on ref. 67 is prescribed at the grounding line, following the approach by refs. 68,69. This parameterization has been shown to produce responses to the loss of buttressing within the range of other ice-sheet models (using different ice-flow approximations), even at coarser resolutions[10,69]. Basal shear stress and velocities are related in a Weertman sliding law with a basal sliding exponent $m = 3$. The field of basal friction coefficients is inferred following the inverse method of ref. 70 (see below). Basal melting underneath the floating ice shelves is calculated by different sub-shelf melt parameterizations, such as the Potsdam Ice Shelf Cavity mOdel[48] (PICO), the plume model[71], and simplified parameterizations[72,73]. Calving at the ice front is determined by the combined penetration depths of surface and basal crevasses (parameterized as a function of the ice velocity, the accumulated strain rate, and the ice thickness), relative to the total ice thickness[14,59]. The response of the bedrock to changes in ice and ocean load is included using the Elastic Lithosphere-Relaxed Asthenosphere (ELRA) model, with a flexural rigidity of the lithosphere of $10^{25}$ N m and a relaxation time of 3000 years[61,74].

Initial ice-sheet conditions are obtained by a transient inverse simulation following refs. 70,75 under constant historical climatic boundary conditions associated with the year 1950 (see below). To minimize the difference between the modeled and observed ice thickness[51], basal friction coefficients under grounded ice and sub-shelf melt rates under floating ice are adjusted iteratively[10]. The calving front is fixed at its observed present-day position. To limit the impacts of transitioning from the balance sub-shelf melt rates derived during the transient inverse simulation to those determined by a sub-shelf melt parameterization, the initialization is followed by a short 10-yr relaxation run applying historical atmospheric and oceanic boundary conditions.

**PISM.** Projections with the thermomechanically-coupled ice-sheet model PISM use the model setup of ref. 12, based on a modified version of PISM release v1.0[9,76]. Ice velocities resulting from the shallow ice and shallow shelf approximations are superimposed over the entire ice-sheet/ice-shelf domain in this hybrid ice-flow model. The position of the grounding line results from hydrostatic equilibrium, and its movement is simulated on the subgrid scale, with a linear interpolation of the basal shear stress across the grounding line[77]. The sliding of the ice sheet over the underlying bedrock is represented by a general power law of the form

$$\boldsymbol{\tau}_b = -\tau_c \frac{\mathbf{u}_b}{u_{th}^q |\mathbf{u}_b|^{1-q}} \tag{1}$$

relating the basal shear stress $\boldsymbol{\tau}_b$ and shallow shelf approximation basal sliding velocities $\mathbf{u}_b$. A threshold velocity $u_{th} = 100$ m yr$^{-1}$ and sliding exponents $q = 0.5$ and $q = 0.75$ (depending on the initial state associated with each atmospheric climatology, see below) are chosen. The yield stress $\tau_c$ depends on the effective pressure of the subglacial till and the till friction angle. The till friction angle, a material property of the subglacial till, is parameterized depending on the bed elevation[78]. Sub-shelf melt rates are computed by PICO[48] based on the ocean climatology of ref. 79 with basin-wide temperature corrections to match observed aggregated melt rates close to the present day[9,48]. The rate of iceberg calving at the margins of the ice shelves follows the eigencalving law[80], combined with a minimum thickness criterion of 50 m at the calving front, and the removal of ice beyond present-day extents of the Antarctic Ice Sheet[81]. A viscoelastic Earth-deformation model[82,83] is used to include the response of the bedrock and seafloor to changing ice loads. Here, the upper mantle is characterized by a viscosity of $1 \times 10^{21}$ Pa s and a density of 3300 kg m$^{-3}$, and for the lithosphere, a flexural rigidity of $5 \times 10^{24}$ N m is chosen.

Initial ice-sheet states are the result of a spin-up approach under constant historical climatic boundary conditions representing the year 1950. As in ref. 12, a full-physics spin-up ensemble with varying model parameters related to basal sliding next to the SIA enhancement factor is run, starting from a thermal equilibrium obtained under a constant ice-sheet geometry (from ref. 84). The initial ice-sheet state performing well in a scoring[22,81] based on the observed ice-sheet geometry[84] and ice velocities[85] is chosen[12].

### Sea-level contribution calculation.

The calculation of the ice-sheet sea-level contribution in this study is based on the method proposed by ref. 86. This approach estimates the sea-level contribution from a marine ice sheet simulated by an ice-sheet model by considering changes in ice volume above floatation, along with correction terms. These corrections account for the density difference between fresh meltwater and saline ocean water, as well as for changes in mass above floatation that occur for ice grounded below sea level due to isostatic changes, but without actual changes in the sea-level contribution. Specifically, the method includes the water-expulsion effect[52] for this bedrock correction term. Consequently, sea-level contributions depend on the magnitude of bedrock changes across the entire grid. Since the two ice-sheet models used in this study have distinct bedrock adjustment models, they produce varying isostatic responses to changes in ice loading, resulting in different magnitudes for the bedrock correction. In particular, the response to ice unloading in PISM extends beyond the current ice-sheet extent (Supplementary Fig. 4), including under floating ice shelves and in the open ocean, leading to higher corrections and therefore higher sea-level contribution estimates at multi-centennial timescales[19] (Table 1).

### Antarctic climate over the historical period and until 2300

Changes in Antarctic climate are applied to the standalone ice-sheet models based on independent climate model projections (P1 in

Table 2). As in refs. 10,12, atmospheric and oceanic anomalies derived from the climate model projections are combined with present-day atmospheric and oceanic climatologies (P2 and P3 in Table 2). Atmospheric conditions for the 1995–2014 reference period are based on the polar-oriented regional climate models Modèle Atmosphérique Régional (MARv3.11[87]) and the Regional Atmospheric Climate MOdel (RACMO2.3p2[88]). Different initial states are constructed for each of these atmospheric climatologies. Similarly, we use ocean present-day conditions based on either the two-dimensional fields from ref. 79, or the three-dimensional fields from ref. 73 (for Kori-ULB only).

For the atmosphere, monthly-mean air temperature and sublimation anomalies, as well as precipitation ratios (thus avoiding negative absolute precipitation[89]) with respect to the 1995–2014 mean seasonal variations are used. Ocean changes are represented by yearly-averaged temperature and salinity anomalies relative to the 1995–2014 mean. Missing values on the continental shelf (given the coarse horizontal resolution of GCMs) and in currently ice-covered regions are extrapolated following ref. 90.

The ice-sheet simulations begin in the year 1950, allowing for conditioning on observations over the satellite era. Historical simulations are run to the year 2015 using changes in the atmosphere and the ocean derived from CMIP5 NorESM1-M[91], given its performance over the historical period around Antarctica[92]. Future changes in Antarctic climate from 2015 to 2300 under the extended Shared Socio-Economic Pathways SSP1-2.6 and SSP5-8.5[29] are based on a subset of CMIP6 GCMs (MRI-ESM2-0, IPSL-CM6A-LR, CESM2-WACCM, and UKESM1-0-LL; P1 in Table 2).

Sub-shelf melt rates are computed by physically-based parameterizations of varying complexity[48,71,93] $M_{param}$ (P4 in Table 2), given the local thermal forcing approximated from far-field GCM ocean properties[94], and depending on a parameter that modulates the effective ice-ocean heat flux $\Gamma_{eff}$ (P5 in Table 2).

Instead of directly applying the surface melt and runoff from the GCMs, these are determined as a function of monthly air temperatures and precipitation in a positive-degree-day (PDD) approach[10,95,96], depending on the degree-day factors for the melting of ice $K_{ice}$ and snow $K_{snow}$ (P6 and P7 in Table 2). Natural variability is considered using a standard deviation of 4 °C and 5 °C for Kori-ULB and PISM, respectively. In Kori-ULB, the refreezing of surface melt is governed by a simple thermodynamic parameterization and, among others, controlled by the thickness of the thermally active layer $d_{ice}$, while a constant fraction of surface melt refreezes based on the refreezing factor $\psi$ in PISM (P8 in Table 2). PDD-based projections of the surface mass balance and its components have shown good agreement with climate models in terms of both pattern and magnitudes[10]. Near-surface air temperatures are corrected for changes in the ice-sheet surface elevation following the atmospheric lapse rate $\gamma_{atm}$ (P9 in Table 2), thereby accounting for the surface melt-elevation feedback.

## Ensemble design

Perturbed-parameter ensembles were produced with each ice-sheet model. These ensembles include up to nine parameters, depending on the ice-sheet model, that govern interactions at the ice-atmosphere and ice-ocean interface. The parameters and their uncertainty ranges (chosen to be as wide as physically plausible[97]) follow ref. 10 for Kori-ULB, and are very similar for PISM. They are summarized in Table 2. We use a Latin hypercube sampling to generate a 400-member and 300-member ensemble for Kori-ULB and PISM, respectively, assuming uniform prior distributions for all parameters. Each ensemble is then run under both SSP1-2.6 and SSP5-8.5 scenarios, resulting in a total of 1400 simulations.

The designed ensembles include discrete and continuous parameters. The discrete parameters P1 to P3 account for climate forcing uncertainties by including a subset of CMIP6 GCMs (P1) as well as different present-day atmospheric and oceanic climatologies (P2 and

P3). Note that P3 only applies to Kori-ULB's ensemble, while the PISM ensemble only uses ref. 79. We cover uncertainties in ice–ocean interactions by using a variety of sub-shelf melt parameterizations (P4), and sampling, for each of these parameterizations, uncertainty in the parameter that modulates the effective ice–ocean heat flux, i.e., the sensitivity of sub-shelf melt to ocean thermal forcing (P5). Note that PISM's ensemble does not include a sampling from P4, as only PICO is used (Table 2). Finally, continuous parameters P6 to P9 capture uncertainties in ice–atmosphere interactions, i.e., the degree-day factors for the melting of ice (P6) and snow (P7), and the thickness of the thermally active layer or the refreezing factor in Kori-ULB and PISM, respectively, both influencing the refreezing of surface meltwater (P8), and the atmospheric lapse rate (P9).

## Calibration

We perform a calibration of each ice-sheet model's ensemble of simulations independently in a Bayesian framework, as in ref. 10. In particular, a likelihood score is assigned to each ice-sheet model's ensemble member $j$ based on the discrepancies between a number $N$ of observed ice-sheet net mass balance estimates $obs_i$, $i \in \{1 \dots N\}$ from the latest Ice Sheet Mass Balance Inter-comparison Exercise (IMBIE[1]; see Supplementary Table 2), which are averaged over several time-periods during the past decades, and the corresponding modeled mass balance $mod_i^j$. Assuming independence and a normal distribution of the discrepancies between the observed and simulated ice-sheet mass balance, the multivariate Gaussian likelihood function is given by

$$s_j = \exp\left(-\frac{1}{2}\sum_{i=1}^{N}\left(\frac{mod_i^j - obs_i}{\sigma_i}\right)^2\right). \tag{2}$$

Here, $\sigma_i^2$ denotes the discrepancy variance, determined from the observational $\sigma_i^{obs}$ and structural $\sigma_i^{mod}$ error (approximating the model uncertainties) as $\sigma_i^2 = (\sigma_i^{obs})^2 + (\sigma_i^{mod})^2$.

Following refs. 10,21,98, we assign the structural error by multiplying the observational error, here by a factor of 3 for the Kori-ULB and PISM ensembles. The magnitudes of the structural errors (and therefore discrepancy variances) were chosen to avoid heavily weighted scores for a small number of ensemble members[98]. These values reflect a judgment that our ability to model reality is far lower than the ability to measure it[21,25,97].

With a normalization, a weight $w$ is then created for each ensemble member with

$$w_j = \frac{s_j}{\sum s_j}. \tag{3}$$

The influence of the calibration and the calibrated trends over the historical period is shown in Supplementary Fig. 1 for each ice-sheet model ensemble. The posterior distributions of the parameter space are given in Supplementary Figs. 5 and 6.

To evaluate the effect of calibration on the ensembles' distribution of the projected Antarctic sea-level contribution, we calculate the Continuous Rank Probability Score (CRPS), following the approach of ref. 99. The CRPS measures the accuracy and sharpness of a probabilistic forecast by quantifying the difference between the ensemble's cumulative distribution function (CDF) and that of the observation (treated as a step function). Specifically, the CRPS for an ensemble with CDF F(x) and an observation $x_{obs}$ is given by:

$$CRPS = \int_{-\infty}^{\infty}\left[F(x) - H(x - x_{obs})\right]^2 dx \tag{4}$$

where H is the Heaviside step function centered at the observed value $x_{obs}$. A lower CRPS indicates a distribution that more closely matches

**Table 2 | Parameters governing ice–climate interactions and respective uncertainty ranges covered in the ice-sheet model ensembles**

| Parameter | Uncertainty range | | | |
|---|---|---|---|---|
| | **Kori-ULB** | | **PISM** | |
| **P1** CMIP6 GCM climate | MRI-ESM2-O | | | |
| | UKESM1-O-LL | | | |
| | CESM2-WACCM | | | |
| | IPSL-CM6A-LR | | | |
| **P2** Atmospheric present-day climatology | MARv3.11[87] | | | |
| | RACMOv2.3p2[88] | | | |
| **P3** Oceanic present-day climatology | ref. 79 | | ref. 79, with basin-wide temperature corrections[9] | |
| | ref. 73 | | – | |
| **P4** Sub-shelf melt parameterization ($M_{param}$) | PICO ($M_{PICO}$[48]) | | PICO ($M_{PICO}$[48]) | |
| | Plume ($M_{Plume}$[71]) | | – | |
| | Local-quadratic ($M_{quad}$[72,106]) | | – | |
| | ISMIP6 non-local quadratic ($M_{JD20}$[73]) | | – | |
| | ISMIP6 non-local quadratic, with dependency on local slope ($M_{JD20s}$[73]) | | – | |
| **P5** Effective ice–ocean heat flux ($\Gamma_{eff}$) | $\gamma_T^*$ in $M_{PICO}$ | $0.1$–$10 \times 10^{-5}$ m s$^{-1}$ | $\gamma_T^*$ in $M_{PICO}$ | $0.1$–$10 \times 10^{-5}$ m s$^{-1}$ |
| | $C_d^{1/2}\Gamma_{TS}$ in $M_{Plume}$ | $1$–$10 \times 10^{-4}$ | – | |
| | $\gamma_T$ in $M_{quad}$ | $1$–$10 \times 10^{-4}$ m s$^{-1}$ | – | |
| | $\gamma_0$ in $M_{JD20}$ | $1$–$4 \times 10^{4}$ m yr$^{-1}$ | – | |
| | $\gamma_0$ in $M_{JD20s}$ | $1$–$4 \times 10^{6}$ m yr$^{-1}$ | – | |
| **P6** Degree-day factor for the melting of ice ($K_{ice}$) | 4–12 w.e. mm PDD$^{-1}$ | | | |
| **P7** Degree-day factor for the melting of snow ($K_{snow}$) | 0–6 w.e. mm PDD$^{-1}$ | | | |
| **P8** Thickness of the thermally active layer ($d_{ice}$) | 0–15 m | | | |
| Refreezing factor ($\psi$) | | | 0.2–0.8 | |
| **P9** Atmospheric lapse rate ($\gamma_{atm}$) | 5–12 °C km$^{-1}$ | | | |

For Kori-ULB, the parameter $\Gamma_{eff}$ originally takes a value within the range of [0–1], as defined by the Latin hypercube sampling. It is then applied to the uncertainty range of the parameter associated with $M_{param}$. For instance, for the $j$th simulation of the ensemble, if $M_{param}^j$ is the local quadratic parameterization $M_{quad}$, a $\Gamma_{eff}^j$ of 0.5 would correspond to a $\gamma_T$ of $5.5 \times 10^{-4}$ ms$^{-1}$.

the observation. Here, we calculate the CRPS for the 1992-2020 average Antarctic mass balance at the continental scale (i.e., as used for the calibration) as well as for different regions (West Antarctica, East Antarctica, and Antarctic Peninsula).

To assess the influence of calibration, we calculate CRPS values normalized by the CRPS of the prior distribution. A normalized CRPS below 1 indicates improved agreement with observations through the calibration. The probability density functions of the prior and posterior distributions, together with the normalized CRPS values, are shown for each ice-sheet model and region in Supplementary Fig. 1e–l. At the continental scale, the calibration significantly reduces the CRPS for both ice-sheet model ensembles (normalized CRPS of 0.27 for Kori-ULB and 0.44 for PISM). However, this is not necessarily the case at the regional scale. For Kori-ULB, the calibration decreases the CRPS performance for the East Antarctic Ice Sheet. For PISM, the CRPS increases for the Antarctic Peninsula and remains largely unchanged for the West Antarctic Ice Sheet. This can be explained by regional biases in the ice-sheet models ensembles: the Kori-ULB ensemble tends to overestimate mass gain in East Antarctica, while the PISM ensemble underestimates West Antarctic mass loss. These regional biases persist in the calibrated ensembles and may lead to overfitting. For instance, underestimated West Antarctic mass loss in the PISM ensemble is compensated for by favoring ensemble members with mass loss in the Antarctic Peninsula beyond observed values during the calibration. Similarly, in the Kori-ULB ensemble, ensemble members slightly overestimating mass loss in West Antarctica are

favored to offset model drift in the East Antarctic Ice Sheet. The influence of the structural error on the posterior distributions of the Antarctic sea-level contribution and the normalized CRPS values is shown in Supplementary Fig. 7 and in Supplementary Table 3. Posterior medians are overall insensitive to the choice of structural error ($\sigma_i^{mod} = 2, 3, 4 \times \sigma_i^{obs}$), whereas the 5–95% intervals widen with increasing structural error, particularly at longer timescales. We find that assuming a structural error equal to three times the observational uncertainty improves the continental-scale CRPS values while mitigating overfitting at the regional scale.

### Sensitivity analysis

To quantify the contribution of different sources of uncertainty to the projected Antarctic contribution to global mean sea-level rise, we perform a variance-based sensitivity analysis using analysis of variance (ANOVA[100]), following the approach of refs. 19,47. This method allows us to partition the total variance in the projected Antarctic ice-sheet changes into contributions from individual sources of uncertainty sampled in our ensemble (main effects, representing the fraction of variance explained independently by each source of uncertainty), as well as interactions between them. Interaction terms represent the fraction of variance arising from the combined influence of two sources of uncertainty. For example, the ice-sheet model/GCM interaction captures how the influence of the ice-sheet model depends on the choice of the GCM. These interactions capture non-additive responses that cannot be attributed to any single source of uncertainty alone,

meaning that the combined influence of two sources of uncertainty is not simply the sum of their individual effects. In other words, interactions capture variance that emerges only when parameters are varied simultaneously, and cannot be attributed to any one parameter alone.

The ANOVA is performed on the calibrated ensembles using weighted linear regression models and computing variance contributions via weighted sums of squares. We use Type-I (sequential) sums of squares, which allows variance to be attributed incrementally to each factor. The calibration weights $w_i$ are passed directly to the MATLAB `fitlm` function.

We apply the ANOVA separately to each ice-sheet model ensemble (Fig. 5), using all model-specific sources of uncertainty as input factors. In these individual analyses, we focus only on the main effects and do not explicitly analyze interactions between them.

We also perform an ANOVA on the combined ensemble to estimate the contribution of structural differences between the ice-sheet models to the total uncertainty (Fig. 4). The sources of uncertainty are grouped into four broader categories: atmospheric parameters, oceanic parameters, GCM, and ice-sheet model (Fig. 4). Atmospheric parameters include the atmospheric present-day climatology (P2; Table 2), the atmospheric lapse rate (P9) and parameters controlling surface melt (P6 and P7) and runoff (P8). Oceanic parameters include the oceanic present-day climatology (P3), the sub-shelf melt parameterization (P4) and the effective ice–ocean heat flux (P5). Because of the sequential sum of squares and differences in the parameter spaces of the Kori-ULB and PISM ensembles (some parameters are varied only in Kori-ULB and are fixed in PISM; Table 2), we list the ice-sheet model as the first factor in the ANOVA model. This ensures that structural differences between the ice-sheet models are attributed first, preventing artificial inflation of variance contributions to be assigned to parameters (such as the sub-shelf melt parameterization) that would otherwise absorb structural uncertainty.

Potential additional two-way interactions (for example, within a given category), higher-order interactions involving three or more parameters, and residual unexplained variance may also contribute to the variance but are not explicitly included. These omitted contributions are represented by the white space in Figs. 4 and 5.

## Data availability
The data generated in this study and needed to produce the figures and tables, as well as the scripts, have been deposited in the Zenodo repository under accession code: https://doi.org/10.5281/zenodo.17432520[101]. The CMIP6 forcing data used in this study[102–105] are accessible through the CMIP6 search interface (https://esgf-node.llnl.gov/search/cmip6/).

## Code availability
The source code for the Kori-ULB ice-sheet model is publicly available on GitHub via https://github.com/FrankPat/Kori-dev (last access: 14 August 2024). The source code for PISM is publicly available on GitHub via https://www.pism.io (last access: 14 August 2024). The code version of both models used in this study is deposited in the Zenodo repository under accession code: https://doi.org/10.5281/zenodo.17432520[101].

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

## Acknowledgements

This publication was supported by PROTECT. This project has received funding from the European Union's Horizon 2020 research and innovation program under grant agreement no. 869304, PROTECT contribution number 165 (V.C., A.K.K., T.E., F.T., F.P., and R.W.). This study has also been supported by the European Union's Horizon 2020 research and innovation program under Grant Agreement no. 820575 (TiPACCs; A.K.K. and R.W.). V.C. acknowledges funding by the Fonds de la Recherche Scientifique de Belgique (F.R.S.-FNRS) with an F.R.S.-FNRS Postdoctoral Researcher Fellowship. Computational resources for Kori-ULB simulations have been provided by the Consortium des Équipements de Calcul Intensif, funded by the Fonds de la Recherche Scientifique de Belgique (F.R.S.-FNRS) under Grant No. 2.5020.11 and by the Walloon Region. The authors gratefully acknowledge the European Regional Development Fund, the German Federal Ministry of Education and Research and the Land Brandenburg for supporting this project by providing resources on the high-performance computer system at the Potsdam Institute for Climate Impact Research. Development of PISM is supported by NASA grants 20-CRYO2020-0052 and 80NSSC22K0274, as well as NSF grant OAC-2118285. We acknowledge the World Climate Research Program's Working Group on Coupled Modeling, which is responsible for CMIP, and we thank the climate modeling groups (whose models are listed in Table 2 of this paper) for producing and making their model output available.

## Author contributions

The study was conceived by V.C. and A.K.K. together with R.W. and F.P. A.K.K. and V.C. developed the experimental setup and design, with contributions from T.E. and F.T. V.C. and A.K.K. processed the forcing data, set up and initialized the ice-sheet models and performed the model simulations. A.K.K. and V.C. analyzed the data, produced the figures and wrote the original manuscript draft, supported by T.E., F.P., F.T., and R.W.

## Funding

## Competing interests

The authors declare no competing interests.
