## [Transparent Peer Review file · Nature Communications]

From short-term uncertainties to long-term certainties in the future evolution of the Antarctic Ice Sheet

Corresponding Author: Ms Ann Kristin Klose

Version 0:

Reviewer comments:

Reviewer #1

(Remarks to the Author)

General comments:

This manuscript reports a multi-model, historically-calibrated, numerical modelling experiment to evaluate how the Antarctic Ice Sheet will respond to future warming scenarios. The manuscript uses end-member Shared Socio-Economic Pathways (SSPs) for their projections of how the ice sheet may change up to the years 2300 and 3000.

The modelling experiment design and methods appear robust, at least to someone with limited direct modelling knowledge and expertise. The paper is well-written with reasonably good figures and diagrams, and an acceptable overall structure. However, throughout the manuscript, I did struggle to identify a clear take-home message that the paper was conveying. The overall findings are very much in accordance with previously published ice sheet modelling experiments of future Antarctic Ice Sheet change: i.e. glaciological change, and sea level rise, is relatively limited up to 2100 with little variation between SSPs, but then after 2100 the future scenarios of changes are higher magnitude, and there is divergence between the model outputs dependent on SSP used etc. What the authors appear to be stating, is that they have done the following differently to previous studies: (1) extended model runs up to 2300 and beyond (i.e. to yr 3000); (2) calibrated their model runs against historical observed ice sheet change (i.e. since 1950); and (3) assessed a range of model parametric uncertainties. I do not know the Antarctic Ice Sheet modelling literature sufficiently well enough to distinguish whether these are new and novel approaches to the previous experiments done by others. However, it is clear to me that it felt like I had to dig out this potential novelty from the manuscript. The manuscript didn't necessarily present the findings and key messages particularly well. Below in my specific comments I have tried to make some suggestions for approaches to restructure some of the sections/paragraphs and for the text to be more specific, so that the reader is better signposted to the key messages and novelty of what has been done.

Specific comments:

L12-13: Should read "United Kingdom of Great Britain and Northern Ireland" (England, Wales, Scotland & associated islands alone is simply 'Great Britain')

L20: "warming" – of what? Ocean, atmosphere etc.?

L20-24: I suspect this can be further condensed into 2-3 lines.

L24-25: "perturbed-parameter ensembles" is not terribly accessible language for a Nature Comms abstract aimed at a broader audience.

L32: "strong changes" – be specific here and quantify what is meant by strong.

L39-40: "coming years to decades" – be specific about timescale here (e.g. before 2050).

Abstract (Lines 20-40): Based on the abstract alone, I struggled to understand what the key novel message of the manuscript was. The modelling results replicate what we know already. That's not necessarily an issue, as reproducibility is good, but it would be good for the abstract (and wider manuscript) to deliver a new contribution to knowledge in some way. We already know that West and East Antarctica will change dramatically by the year 3000 if we continue with business-as-usual type scenarios, and that if we want to do something about sea level rise then we need to implement strong mitigation asap.

L45: "and parts of East Antarctica"?

L52-54: Based on my reading of the manuscript (particularly line 80), this is a key sentence as it is key to the experiment design (i.e. what this experiment does that the previous experiments didn't). It's effectively what is new and novel about this

work. However, I fear it is a bit isolated here and doesn't really follow from the paragraph it sits at the end of. I wonder if a bit of re-structuring of the intro would help, either by: (option 1) open the paragraph with "Parametric uncertainties used in continental-scale ice-sheet models have not been systematically explored", then follow with details of the problem and explain what can be done to address this. Follow this approach for paragraphs 2&3 too (i.e. few current modelling experiments extend to 2300/historical calibration); or (option 2): state in opening paragraph what your model experiment will do (i.e. explore parametric uncertainties, model to 2300, calibrate against historical observations etc.) then state why this is a step change in approach/knowledge. Either of these two options would substantively help to emphasise the different approach being taken here. At present, the interested reader has to seek the info out and solve the puzzle of where it is and how it connects across the introduction, and the wider manuscript.

L59: Delete "what is more"

L62-64: Again, this is a key message for the manuscript. Don't hide it at the end of the paragraph. Move it to start of paragraph (option 1 above) or move to start of introduction section.

L66: "robust projections" – the implication is that this modelling experiment is more robust than previous ones. Is this the case?

L65-73: The key message of this paragraph is totally lost. It seems to me that the "uncertainty" angle should be merged with issue raised in L52-54 (i.e. parametric uncertainty), whilst this paragraph should be all about the issues of model experiment timescales being too short (i.e. not extending beyond 2200). Longer model experiments (e.g. to 2300) allow coastal planning to be implemented effectively. Also, why all the italics in this paragraph?

L73: unclear to me what is "poorly quantified". The "interests of coastal planning stakeholders"?

L72: "coastal planning stakeholders" – they are briefly mentioned here, and then again later in the manuscript, but without who they are being specified, nor any effort being attempted at the model experiment results being distilled in a way that is useful for these so-called "stakeholders". Perhaps just drop reference to them or be more explicit about who they are and why they are important.

L74-84: This is where the authors are stating what is coming in the manuscript, whilst also making the case for the novelty of their approach. If the rest of the intro section were rewritten to build a strong argument and case for why previous model experiments were limited, this paragraph could be much more effective and hard hitting.

L79: what is "CMIP"? Define at first use.

L80 (and L52-53) what are these "parametric uncertainties"? – give detail and examples.

L88-89: Acronym soup. Please write out model names in full here for the wider readership of the manuscript.

L92: detail examples of the biases in Antarctic climate in those papers here – they could be critical for evaluating the quality of the work done here.

Figure 1: I am unclear what purpose figure 1c serves. It doesn't seem to be discussed in the text in any substantive way. It certainly isn't referred to specifically, whereas 1a and 1b are. It wasn't clear to me why the "relation between Antarctic-averaged atmospheric and circumantarctic ocean temperature changes" was important to include in the main body of the text.

L99: provide temperature change values as a range, as done later in paragraph for SSP5-8.5 scenario?

L104-112: Paragraph structure could be improved (e.g. split into temperature and ocean temperature change) to make it easier for the reader to distil the primary messages being conveyed. Currently, the atmosphere and ocean changes are too blended to make it clear which emission pathway does what.

L137-140: "likely contribute" – can the authors add a few words here to assess or quantify the likely overestimation of ice sheet net mass balance (i.e. not just link to an extended data figure).

Figure 2: I find this figure extremely challenging to extract information from. The overlapping colours on 2a are simply an overlapping blend of purple/brown, whilst the symbols on 2b cannot be discriminated easily. Suggest the authors explore how to improve the quality of the layout and presentation of this figure.

L151: "By 2300, the modelled Antarctic....."?

L153-154: Annotate PIG and TG on Figure 3 or remove reference to them here. Why are these glaciers mentioned specifically, when there are no glaciers referred to for Weddell Sea/Siple Coast?

L156: why do Wilkes and Recovery "potentially" lose mass? Geomorphologically, it is perhaps arguable whether Recovery is a "subglacial basin", in the sense of Aurora and Wilkes, too. Perhaps refer to Recovery and Wilkes catchments?

L163-167: This seems to be a key message of this paragraph, but is lost at the end, almost as a throwaway comment "In fact.....". It deserves a paragraph of its own, and some additional detail. It may be the key message of this manuscript that is of interest to a wider audience (e.g. a non-discipline specific audience).

L168: Suggest “aligns broadly” is changed to “are consistent with”

L171-172: References are required after “...loss to 2300.”

L183-185: Previously, the manuscript makes a big argument about there being a lack of modelling up to 2300, but here it reports that there the ice sheet model intercomparison project has modelling up to 2300. It would be helpful if the authors could explain this apparent contradiction, given that model runs to 2300 are argued as being something the authors do here that is ‘different’ from previous model experiments.

L185-190: More information is required here about why the lack of an overlap is important. What does this finding tell us about the likely processes operating on the Antarctic Ice Sheet and how realistic MICI is?

L236-257: As a non-modeller, I found it difficult to takeaway the key messages from this section. Perhaps it is a failing on my part, but the alternative is that it is because the key messages the authors are trying to convey are not being signposted effectively enough.

L258-266: Moving this section earlier in this section (perhaps to current line 240?), might likely help address the issues I found with L236-256. This is a key message to convey. Move it to the start of the section, headline what is important, and then provide the detail.

L268-270: A section opening sentence that does not really say very much. Authors could delete this entirely and simply start with L271, or they could make the statement more specific and less generic. They could consider moving lines 302-305 to the start of this section too.

L274: rather than “end of this millenium, be specific and say year 3000.

L273-277: a long sentence. Authors should consider breaking it into multiple sentences.

L278: What is meant by “tipping dynamics”? If they are hinting at tipping points here, then the end of the sentence (i.e. “very likely long-term ice sheet retreat”) does not fulfil the definition of a tipping point. As the authors do not assess potential irreversibility in their experiments, I recommend removing reference to “tipping” or “tipping points”.

L284-286 (& follow on L287-289): I don’t find this finding a surprise given the different glaciological and topographic geometries of these two basins. Have previous modelling publications suggested otherwise?

L294: state year 3000, rather than “end of millenium”?

L296: Instead of “likely as not”, what about “highly likely”?

L297: “...reflecting the trend in the ice sheet model experiments up to 2300....”?

L299: Be specific about what is meant by “over the next centuries”. 2 centuries? 3, 4?

L310-311: I’d argue that L312-348 are mainly about model uncertainties and limitations rather than “Next decades decisive for Antarctica’s long-term future”. As such, I recommend the authors consider changing the title of this section.

L312-315: Another rather general opening sentence, that doesn’t really make a clear statement about the findings of the modelling experiments undertaken in this study. For the manuscript to be more effective, it needs to have much more targeted and hard-hitting opening sentences for each section. I recommend that the authors carefully review all parts of the manuscript to ensure that it is fully delivering the key messages they want it to make. At present, there is a bit of a disconnect between what I suspect they think they are saying, and what they are saying in the manuscript. The key messages need to be at the forefront. For example, if ensemble modelling of the type done here (historically calibrated, running to 2300 (and/or 3000), with assessment of parametric assumptions etc.) is vital, and not done in previous studies, then make it clear what it enables. What is the step change? E.g. “Historically-calibrated ensemble modelling enables us to deliver XYZ.....”.

L314: Again there is a mention of “coastal adaptation planning”, but no details on specifically how the work done here feeds into such activities.

L324-348: All about model uncertainty and limitations, rather than ‘on-topic’ (i.e. next decades decisive.....) for this section. If the authors persist with this content here (perhaps with a modified section heading), it would be good if the authors can evaluate the implications of a model horizontal resolution of 16 km on their model results. This must have implications for modelling retreat via Wilkes vs Aurora Basin, simply because of the widths of the basin entry ‘channels’.

L351: “some of which may be irreversible” – the authors may wish to reconsider this given that they ‘only’ model up to year 3000.

L349-359: this content is more relevant to the section heading than lines 324-348. However, it doesn’t really present anything that we didn’t really know already from other work (e.g. substantial long-term Antarctic changes....will occur unless massive mitigation efforts are made). The authors need to make a much stronger case for why what they have done here provides something new and novel. What is the original contribution to knowledge they are making? We know that the “next

few years and decades are decisive for the multicentennial fate of the Antarctic Ice Sheet.” What is it that this study has done that is different from previous work? The message that needs to be provided in bold letters is probably the one that the authors outline in the intro section, the rigour of the modelling and analysis that is done here (i.e. assessment of parametric uncertainty, model historical-calibration etc.). However, the way the paper is currently written doesn’t relay that message effectively at the present time.

Neil Ross
Newcastle University

Reviewer #2

(Remarks to the Author)

Coulon et al. use multiple models and statistical sampling to project long term sea level rise commitments under different emissions pathways. They find that near term projections are strongly influenced by ice sheet model sensitivities, especially under limited warming scenarios. However, long-term projections are dominated by changes in Antarctic climate, particularly under unmitigated emissions. The study emphasises the importance of strong emissions reductions today to prevent multi-metre contribution to sea level rise from both the East and West Antarctic Ice Sheets.

General comments

I found this to be an impressive study and, with a few minor edits and some further clarifications, this paper would be a useful addition to Nature Comms. It is a significant and much-needed contribution to sea level rise projections over multi-centennial timescales. As the authors note, there are currently few studies that have looked at ice sheet changes over longer timescales. The use of multiple models, their calibration with historical data, use of statistical sampling, and consideration of various emission scenarios provide an exceptionally robust and comprehensive analysis of the future evolution of the Antarctic Ice Sheet and its potential contribution to sea level rise. Furthermore, the authors account for a wide range of uncertainties in climate and ice sheet responses. The methodology is especially comprehensive and both the interpretation of the data and conclusions drawn are sound.

Specific comments

L49-52 Add reference to: J.L. Bamber, M. Oppenheimer, R.E. Kopp, W.P. Aspinall, R.M. Cooke, Ice sheet contributions to future sea-level rise from structured expert judgment, Proc. Natl. Acad. Sci. U.S.A. 116 (23) 11195-11200, <https://doi.org/10.1073/pnas.1817205116> (2019). The Pollard and DeConto model isn't the only approach that suggests high SLR contributions.

L55-59 Clarity of these sentences could be improved. Replace the colon with a full stop and make this section more concise.

L64 No need for brackets here.

L67-73 Consider rephrasing to emphasise the importance of near-term emissions reductions, as this has clearer policy relevance than adaptation planning on multi-centennial timescales.

L75 Please state how far “beyond”

L78-81 Consider adding some examples of parametric uncertainties.

L120-140 Links go to the wrong tables and figures.

L120 Adding a brief explanation for using Latin Hypercube sampling would help readers unfamiliar with this method understand its advantages. Add references to previous use for some helpful context.

L171-172 “Only a few individual studies have explored the Antarctic ice loss to 2300.” Add references here.

L107, L213, L245, L261 and Fig 1 caption: Change to circum-Antarctic

Figure S1: Consider using a larger scale.

Reviewer #3

(Remarks to the Author)

My comments can be found in my review attachment.

Version 1:

Reviewer comments:

Reviewer #1

(Remarks to the Author)

The authors have undertaken a very thorough, comprehensive and thoughtful response to all reviewers' comments, and the manuscript is now in excellent shape. The tracked change document clearly evidences substantive rewriting and reorganisation of the manuscript text, to deliver a much improved and highly robust manuscript. The figures and diagrams are also much improved.

I particularly appreciate the very considered and positive way in which the authors approached suggestions for improving the overall messaging of the manuscript. As someone who is not a modeller, I was wary of providing such suggestions, feeling slightly out of my depth(!). However, I am delighted to see how effectively the changes made by the authors have improved the manuscript. The introduction is now focused, well structured, and justifies this study and the knowledge gap it addresses very effectively. The introduction, and the closing sections of the manuscript (lines 355-469), now accurately reflect and fairly represent, the quality, depth, and novelty of the research done in this study. I can easily identify and absorb the important information provided by this important study now. If I, as a non-modeller with slightly rusty mathematical and statistical skills, can do this then the authors can be confident that they are now able to target and impact a very broad audience with their manuscript.

I also fully understand the author team's logic where they did not implement recommendations from the reviewers, and agree with their decision making where changes have not been made.

I consider it a privilege to have been able to contribute (in a minor reviewer way) to the development of this manuscript, and it was a pleasure to read the very robust and focused revised manuscript. The authors have approached the reviews in a very positive and constructive way and have every right to be proud of what they have produced. To quote reviewer 3, the work is "truly impressive", as well as now being very accessible. I look forward to using the published version of the manuscript for both research and teaching purposes.

I have one minor comment (relating to lines 366-367) – I didn't think there was 6.91 m of West Antarctic ice to lose? Is this a typo?

Neil Ross
Newcastle University

Reviewer #2

(Remarks to the Author)

I appreciate the effort that the authors have put into the revisions. I'm happy that they have addressed all of my comments. I have no additional suggestions and believe the manuscript is now in good shape and ready for publication.

Reviewer #3

(Remarks to the Author)

This manuscript provides a robust analysis of the future evolution of the Antarctic Ice Sheet (AIS) and its contribution to sea level rise under two contrasting emission scenarios: low-emission (SSP1-2.6) and high-emission (SSP5-8.5). Using an ensemble modeling approach within a Bayesian framework, the study addresses uncertainties in ice-climate interactions (ocean and atmosphere) and ice sheet model structure (Kori and PISM).

The findings highlight that, for projections up to 2100, uncertainties are primarily driven by ice sheet model choice and ice-ocean interactions. Beyond 2100, however, the emission scenario becomes the dominant factor. Under SSP5-8.5, the study predicts a high likelihood of significant mass loss from the West Antarctic Ice Sheet (WAIS) and destabilization of subglacial basins in the East Antarctic Ice Sheet (EAIS), such as Wilkes. In contrast, adherence to SSP1-2.6, aligned with the Paris Agreement, substantially reduces the risk of WAIS collapse and prevents irreversible EAIS loss.

The manuscript is well-written, concise, and methodologically sound. Figures are clear, and the structure—from introduction to discussion—is well-organized. This work is a valuable contribution to the field, complementing prior studies like ISMIP and extending the understanding of ice-climate interactions and SSP uncertainties. The use of IPCC-aligned uncertainty terms enhances accessibility for a broader audience.

The authors have addressed all major concerns from my previous review, particularly regarding methodology and scenario interpretation. I now consider this paper ready for publication as is, with only minor optional refinements to further polish the details.

Minor comments:

L36-38: I agree with both parts of the sentence, but I am unsure whether the second part is a direct consequence of the first, or if that is what you intended to show. Your results indicate that SSP1-2.6, which is more optimistic than current mitigation

efforts, may not be sufficient to entirely avoid self-sustained Antarctic ice loss, and this loss is significantly less likely under SSP1-2.6 compared to SSP5-8.5, which represents a more pessimistic scenario than current mitigation efforts. These are two distinct findings, yet the current wording implies that the second result is a consequence of the first. I recommend rephrasing for clarity.

L88-89: all ice-sheet models using inverse method are calibrated with satellite observations. Perhaps you mean transient satellite observations?

Fig. 2: I really like the updated style of the figures! The colors are visually appealing and work well, though I hope they won't confuse readers familiar with the IPCC reports, where light blue is often used for SSP1-1.9 and light red for SSP3-7.0. As it stands, the figure is very clear and effective. For panel b, however, the names of the GCMs on the bottom axis appear a bit crowded and stacked—this might just be my perspective, but it could benefit from slight adjustments to improve readability.

L289-290: It seems that the negligible structural uncertainty by 2300 may result from opposing effects between the Antarctic Peninsula (where this uncertainty is quite significant) and the West Antarctic Ice Sheet. Additionally, since there are only two ice sheet models, when the medians of the two models intersect, their contribution to the overall uncertainty effectively drops to zero. However, I suspect that this uncertainty would begin to grow again beyond 2300. This changes the conclusion of this sentence somewhat.

L295-300: Similar to my previous comment, I think it is likely coincidental that both models produce the same median at this point. The curves appear quite different, and it seems probable that beyond 2300, PISM would project higher mass loss. For example, introducing a third model would likely alter these conclusions.

Fig. 4: Why is the total so far from 1 at the beginning? It looks much better in Fig. 5. From my perspective, this discrepancy might arise from the ISM/atmosphere and ISM/ocean uncertainties, which could be much higher initially—especially ISM/ocean. By 2100, most of the uncertainty comes from Kori, and its uncertainty is largely driven by ocean processes. Therefore, ISM/ocean should be the dominant source of uncertainty in 2100 in Fig. 4.

After reviewing the methodology, I realized this wasn't immediately clear. In Fig. 4, you are not showing all ocean-related uncertainties but only the common source of uncertainty explored in both models: the ice-ocean heat flux. However, given the importance of other ocean-related uncertainties in Kori, it seems likely that these would account for much of the white space in Fig. 4. It would be helpful to include all ocean-related uncertainties in Fig. 4. Even though PISM does not explore sub-shelf melt parameterization, it uses Mpico, which falls within Kori's parameter space. This suggests it should be possible to account for this uncertainty.

Here's a potential approach: For Kori, you have the same number of members for each parameter ($400/5 = 80$ members per parameter). For Mpico (parameter 0), you have 80 members from Kori and 300 members from PISM. You could shuffle these 380 members to create five smaller groups of 80 members each. (I realize $5 \times 80 \neq 380$, but if you instead use groups of 20 members, this could work: 19 groups of 20 members with Mpico, 4 groups of 20 members with Plume, etc.). Afterward, you could compute the variance between these groups to estimate the variance due to sub-shelf melt parameterization. Dividing this variance by the total variance would give you the contribution of this uncertainty. A similar approach could be applied for oceanic present-day climatology.

For the variance from the interaction between ISM and ice shelf parameterization, I think the value should be exactly the same: though I haven't thought through all the details and have just made a quick sketch, it seems that by using the previous method you split the variance due to the melt parameterization half in the main effect and half in the interaction. Another way to show this uncertainty is to attribute all the variance to the interaction: you compute the variance due to melt parameterization in the Kori ensemble and divide it by the total variance of both ensembles.

Finally, I didn't see any explicit mention that these sensitivity indices are computed on the calibrated ensemble (though I assume they are). It might be helpful to clarify this in your description of the sensitivity analysis, including how you account for the weights in this process. Also, if this is the case, the method proposed above would need to be modified slightly. Instead of forming groups with the same number of members, you would create groups with the same total weight.

L453: A good citation to add for sliding and subglacial hydrology: Zhao, C., Gladstone, R., Zwinger, T. et al. Subglacial water amplifies Antarctic contributions to sea-level rise. *Nat Commun* 16, 3187 (2025). <https://doi.org/10.1038/s41467-025-58375-4>

L466-469: Like in the abstract, I'm not sure I understand why the second part of the sentence is presented as a consequence of the first part. To me, the second part seems to be a consequence of the lower likelihood of extreme sea level rise under SSP1-2.6.

Methods:

L518-519: I'm a bit confused about the q value. Does it depend on your spin-up calibration (L536-538)? In the end, do you obtain a single q value for all members, or does it vary between members based on the scoring described in L537?

L592-593: I don't quite understand how natural variability is accounted for. Do you vary the temperature using a Gaussian distribution with a variance of 4°C and 5°C?

L666: Fortunately, the CRPS decreases across the entire domain, as expected since the data was used for calibration! It would have been a major red flag if that weren't the case. That said, while it doesn't always decrease at the regional scale, it does in most cases, which is a good sign for robustness.

Hope it helps, and good luck with your revisions!

Eliot Jager, University of Helsinki

**Response to the comments of the reviewers for the manuscript
'From short-term uncertainties to long-term certainties in the future evolution of the
Antarctic Ice Sheet'**

by V. Coulon, A.K. Klose, T. Edwards, F. Turner, F. Pattyn, and R. Winkelmann

We would like to thank the reviewers for carefully reading our manuscript and for their efforts in creating their review comments. We considered their suggestions thoroughly and adapted the manuscript accordingly.

We provide a point-by-point response to all comments below. The reviewers' comments are given in bold and black font, the authors' reply in normal and blue font. A revised manuscript and a manuscript highlighting the changes are attached.

In the revised manuscript, we have particularly addressed the following points:

- We have restructured the **Introduction** to highlight the novelty of our work.
- We have updated the **calibration** of our ice-sheet models ensembles and demonstrate the benefits of this calibration approach based on the Continuous Ranked Probability Score (CRPS).
- We have **quantified the roles of the various sources of uncertainty** sampled in our ensemble in the range of projected Antarctic ice-sheet changes based on an analysis of variance (ANOVA). This analysis quantitatively underlines the key messages with respect to the roles of uncertainties in modulating future Antarctic mass loss. The respective section has also been restructured and reformulated accordingly.
- We have restructured and reformulated major parts of the **Results** to clarify the findings and key messages of our work.

We are grateful for the opportunity to further improve our manuscript and are looking forward to your feedback.

Yours Sincerely,

Violaine Coulon, Ann Kristin Klose, Tamsin Edwards, Fiona Turner, Frank Pattyn, and Ricarda Winkelmann

Reviewer #1 (Remarks to the Author)

General comments

This manuscript reports a multi-model, historically-calibrated, numerical modelling experiment to evaluate how the Antarctic Ice Sheet will response to future warming scenarios. The manuscript uses end-member Shared Socio-Economic Pathways (SSPs) for their projections of how the ice sheet may change up to the years 2300 and 3000.

The modelling experiment design and methods appear robust, at least to someone with limited direct modelling knowledge and expertise. The paper is well-written with reasonably good figures and diagrams, and an acceptable overall structure. However, throughout the manuscript, I did struggle to identify a clear take-home message that the paper was conveying. The overall findings are very much in accordance with previously published ice sheet modelling experiments of future Antarctic Ice Sheet change: i.e. glaciological change, and sea level rise, is relatively limited up to 2100 with little variation between SSPs, but then after 2100 the future scenarios of changes are higher magnitude, and there is divergence between the model outputs dependent on SSP used etc. What the authors appear to be stating, is that they have done the following differently to previous studies: (1) extended model runs up to 2300 and beyond (i.e. to yr 3000); (2) calibrated their model runs against historical observed ice sheet change (i.e. since 1950); and (3) assessed a range of model parametric uncertainties. I do not know the Antarctic Ice Sheet modelling literature sufficiently well enough to distinguish whether these are new and novel approaches to the previous experiments done by others. However, it is clear to me that it felt like I had to dig out this potential novelty from the manuscript. The manuscript didn't necessarily present the findings and key messages particularly well. Below in my specific comments I have tried to make some suggestions for approaches to restructure some of the sections/paragraphs and for the text to be more specific, so that the reader is IPCC-AR6 better signposted to the key messages and novelty of what has been done.

We thank the reviewer for this constructive and thoughtful review, which has been extremely helpful in improving the readability of the manuscript as well as the clarity of its novelty and key messages. We agree that these aspects were not sufficiently emphasized in the original version.

Significant Antarctic mass changes are expected after 2100, both because slowly-unfolding self-sustained ice loss could be triggered in the coming decades, and because climate changes are expected to intensify beyond the end of the century, especially under higher-emission scenarios. However, in IPCC-AR6 (Fox-Kemper et al., 2021), multi-centennial Antarctic sea-level projections were assigned low confidence and not associated with a likelihood statement. This is due to the small number of available studies that had assessed Antarctic ice loss beyond 2100 under continued global warming and the large spread among them (e.g. DeConto and Pollard, 2016; Bulthuis et al., 2019; DeConto et

al., 2021).

Since IPCC-AR6, ISMIP6-2300 (Seroussi et al., 2024) has made an important step forward by providing a large multi-model ensemble (16 ice-sheet models) projecting the Antarctic ice-sheet evolution to 2300. This assessment mainly focused on very high emissions (SSP5-8.5 and RCP8.5), while the lower-emission pathway was represented by a single General Circulation Model (GCM). In addition, ISMIP6-2300 did not include any systematic parametric uncertainty sampling and did not calibrate projections against historical observations. These two aspects are not specific to ISMIP6 but remain broader challenges in Antarctic modelling for providing robust projections: parametric uncertainty has rarely been covered consistently across ice-sheet models, and only a few studies have calibrated projections on the continental scale with satellite observations (Golledge et al., 2019; DeConto et al., 2021; Lowry et al., 2021), and even fewer in a Bayesian framework (Ritz et al., 2015; Coulon et al., 2024).

Our study builds on and extends previous work in several key ways:

1. **Emission pathway uncertainty** – balanced assessment of both low (SSP1-2.6) and very high (SSP5-8.5) emissions, each with four GCMs.
2. **Parametric uncertainty** – systematic sampling of parameters controlling ice–climate interactions across two continental-scale ice-sheet models.
3. **Historical calibration** – Bayesian calibration against observed ice-sheet mass change over the past decades, which increases the robustness of our projections.
4. **Extended timescales** – projections to 2300, and extended until 3000, which allows us to explore the long-term consequences of early-millennium warming.

These elements together provide, to our knowledge, the first comprehensive multi-centennial assessment of the Antarctic ice-sheet evolution that consistently integrates historical calibration, a wide range of emission pathways and parametric uncertainty across two ice-sheet models.

Based on this framework, we find:

- Under very high emissions (SSP5-8.5), Antarctic ice loss could reach several meters by 2300, while reaching net-zero emissions well before 2100 (SSP1-2.6) limits mass loss to below 1.75 m. Even then, the associated sea-level rise would still require major coastal adaptation.
- The dominant sources of uncertainty shift over time: ice-sheet changes are initially most sensitive to ice-sheet model structures and parameters controlling sub-shelf melting, but with stronger warming (projected after 2150 under SSP5-8.5), the role of GCM trajectories and surface melt/runoff parameters becomes increasingly important.
- By 3000, we can assign high confidence to substantial, self-sustained Antarctic mass loss under strong early-millennium warming, including a virtually certain collapse of the West Antarctic Ice Sheet within our probabilistic projections. While substantial East Antarctic ice loss may be avoided under low emissions, widespread West Antarctic retreat is less likely but cannot be excluded.

While some of these aspects in the future Antarctic ice-sheet evolution are consistent with previous work, our framework highlights the (committed) ice-sheet response in a more robust way than before by providing calibrated probabilistic time series of Antarctic ice-sheet changes to 2300 under different SSPs. Given the balanced assessment under SSP1-2.6 and SSP5-8.5, this allows us, for example, to give robust estimates of Antarctic mass changes under constrained emissions.

In the revised manuscript, we have restructured the Introduction to highlight these points of novelty more clearly, and revised the Results and Discussion to emphasize the key take-home messages. We have also added an explicit ANOVA-based quantification of the relative roles of emissions, GCMs, ice-sheet model structures, and parametric uncertainties, which quantitatively underlines the key messages in the Section on “Role of uncertainties in modulating future Antarctic mass loss” and extends previous uncertainty quantifications (e.g. Seroussi et al., 2023; Seroussi et al., 2024).

Specific comments

L12-13: Should read “United Kingdom of Great Britain and Northern Ireland” (England, Wales, Scotland & associated islands alone is simply ‘Great Britain’)

Thanks, this has been updated as “UK”.

L20: “warming” – of what? Ocean, atmosphere etc.?

“warming” has been replaced by “global warming”.

L20-24: I suspect this can be further condensed into 2-3 lines.

Agreed, we have reformulated this part of the Abstract.

L24-25: “perturbed-parameter ensembles” is not terribly accessible language for a Nature Comms abstract aimed at a broader audience.

Agreed, we have replaced “perturbed parameter ensembles” by “systematically sampling parametric and climate model uncertainties”. Note that examples for parameters governing ice–climate interactions are given in the Introduction, as suggested in another comment.

L32: “strong changes” – be specific here and quantify what is meant by strong.

We find that changes in the Antarctic climate start to dominate the ice-sheet evolution after 2150 under the higher-emission pathway. This can be inferred from the ANOVA analysis that has been added in the revised manuscript. We have reformulated this sentence, stressing the conditions (“after 2150 under very high emissions”) for this shift in the dominant source of uncertainty.

L39-40: “coming years to decades” – be specific about timescale here (e.g. before 2050).

We agree that specifying a timescale would be helpful, but in this case it is difficult to be more precise. Given that even the lower-emission scenario may lead to significant West Antarctic ice loss, our intent is to emphasize that any emission reduction in the near-term will be helpful in reducing the likelihood of substantial mass loss later in this century and beyond. We tried to reformulate this sentence to clarify the context.

Abstract (Lines 20-40): Based on the abstract alone, I struggled to understand what the key novel message of the manuscript was. The modelling results replicate what we know already. That’s not necessarily an issue, as reproducibility is good, but it would be good for the abstract (and wider manuscript) to deliver a new contribution to knowledge in some way. We already know that West and East Antarctica will change dramatically by the year 3000 if we continue with business-as-usual type scenarios, and that if we want to do something about sea level rise then we need to implement strong mitigation asap.

We have reformulated the Abstract and major parts of the manuscript to clarify the novelty of our work. Please also see our response to the general reviewer comment for a more detailed discussion of the novelty of our work and respective adjustments in the revised manuscript.

L45: “and parts of East Antarctica”?

Agreed, we have adjusted the formulation as suggested.

L52-54: Based on my reading of the manuscript (particularly line 80), this is a key sentence as it is key to the experiment design (i.e. what this experiment does that the previous experiments didn’t). It’s effectively what is new and novel about this work. However, I fear it is a bit isolated here and doesn’t really follow from the paragraph it sits at the end of. I wonder if a bit of re-structuring of the intro would help, either by: (option 1) open the paragraph with “Parametric uncertainties used in continental-scale ice-sheet models have not been systematically explored”, then follow with details of the problem and explain what can be done to address this. Follow this approach for paragraphs 2&3 too (i.e. few current modelling experiments extend to 2300/historical calibration); or (option 2): state in opening paragraph what your model experiment will do (i.e. explore parametric uncertainties, model to 2300, calibrate against historical observations etc.) then state why this is a step change in approach/knowledge. Either of these two options would substantively help to emphasise the different approach being taken here. At present, the interested reader has to seek the info out and solve the puzzle of where it is and how it connects across the introduction, and the wider manuscript.

We fully agree and thank you for these very relevant suggestions. As developed above, we have completely restructured the Introduction to better highlight the novelty of our work, following your advice as closely as possible.

L59: Delete “what is more”

Ok.

L62-64: Again, this is a key message for the manuscript. Don't hide it at the end of the paragraph. Move it to start of paragraph (option 1 above) or move to start of introduction section.

Agreed, thank you for the suggestion. As developed above, we have completely restructured the Introduction, and hope that this clarifies the key messages.

L66: “robust projections” – the implication is that this modelling experiment is more robust than previous ones. Is this the case?

Yes, for the reasons outlined above (joint sampling of uncertainties in the ice-sheet model structure, climate models, emissions and parameters as well as calibrating against observations), we believe that our work provides a significant step forward with respect to a robust and comprehensive assessment of the future evolution of the Antarctic Ice Sheet compared to the few previous studies that have assessed the Antarctic contribution to sea-level rise beyond 2100.

L65-73: The key message of this paragraph is totally lost. It seems to me that the “uncertainty” angle should be merged with issue raised in L52-54 (i.e. parametric uncertainty), whilst this paragraph should be all about the issues of model experiment timescales being too short (i.e. not extending beyond 2200). Longer model experiments (e.g. to 2300) allow coastal planning to be implemented effectively. Also, why all the italics in this paragraph?

Agreed. We have included two distinct paragraphs addressing the uncertainty and timescales in the restructured Introduction to clarify the key messages and novelty of our work. The italics have been removed.

L73: unclear to me what is “poorly quantified”. The “interests of coastal planning stakeholders”?

We were referring to “high-end projections”. This sentence has been removed in the revised manuscript.

L72: “coastal planning stakeholders” – they are briefly mentioned here, and then again later in the manuscript, but without who they are being specified, nor any effort being attempted at the model experiment results being distilled in a way that is useful for these so-called “stakeholders”. Perhaps just drop reference to them or be more explicit about who they are and why they are important.

We agree. As suggested, we have removed the reference to coastal planning stakeholders in the revised manuscript.

L74-84: This is where the authors are stating what is coming in the manuscript, whilst also making the case for the novelty of their approach. If the rest of the intro section

were rewritten to build a strong argument and case for why previous model experiments were limited, this paragraph could be much more effective and hard hitting.

Thank you for this relevant and constructive comment. We have substantially restructured the Introduction following your advice, and hope that these revisions clarify the novelty of our work.

L79: what is “CMIP”? Define at first use.

We have included a definition of “CMIP” in the revised manuscript.

L80 (and L52-53) what are these “parametric uncertainties”? – give detail and examples.

We have added examples for parametric uncertainties in the revised manuscript.

L88-89: Acronym soup. Please write out model names in full here for the wider readership of the manuscript.

Agreed. We have added the full names of the GCMs in the revised manuscript.

L92: detail examples of the biases in Antarctic climate in those papers here – they could be critical for evaluating the quality of the work done here.

Thank you for this suggestion. We have included the examples that the upper Southern Ocean is generally too warm and fresh (Beadling et al., 2020), and that the Amundsen Sea Low remains poorly captured (Bracegirdle et al., 2020) in GCMs of the Coupled Model Intercomparison Project Phase 6 (CMIP6).

Figure 1: I am unclear what purpose figure 1c serves. It doesn’t seem to be discussed in the text in any substantive way. It certainly isn’t referred to specifically, whereas 1a and 1b are. It wasn’t clear to me why the “relation between Antarctic-averaged atmospheric and circumantarctic ocean temperature changes” was important to include in the main body of the text.

We agree that panel c in Figure 1 was not included in the main text in a detailed way, so we have removed it in the revised manuscript.

L99: provide temperature change values as a range, as done later in paragraph for SSP5-8.5 scenario?

Agreed, thank you for the suggestion. We have added the range of the projected Antarctic-averaged atmospheric and oceanic temperature change under SSP1-2.6.

L104-112: Paragraph structure could be improved (e.g. split into temperature and ocean temperature change) to make it easier for the reader to distil the primary

messages being conveyed. Currently, the atmosphere and ocean changes are too blended to make it clear which emission pathway does what.

Thank you for this comment. However, the goal of this paragraph is to focus on characteristic atmospheric and oceanic changes in Antarctica for each GCM considered in this work that are independent of the emission pathway. For example, projections by CESM2-WACCM indicate the strongest atmospheric warming in Antarctica of the four GCMs considered within each emission pathway. IPSL-CM6A-LR and UKESM1-0-LL project an early and abrupt Antarctic ocean warming in the first half of this century under both emission pathways.

In the revised manuscript, we have added an introductory sentence to this paragraph, clarifying its main aim, and have reformulated parts of the paragraph. We hope that these adjustments clarify the main aim and key message of this paragraph.

L137-140: “likely contribute” – can the authors add a few words here to assess or quantify the likely overestimation of ice sheet net mass balance (i.e. not just link to an extended data figure).

We have added a comparison of the calibrated median ice-sheet net mass balance and the observed mass balance between 1992 and 2020 in the revised manuscript to quantify the overestimation of the ice-sheet net mass balance.

Figure 2: I find this figure extremely challenging to extract information from. The overlapping colours on 2a are simply an overlapping blend of purple/brown, whilst the symbols on 2b cannot be discriminated easily. Suggest the authors explore how to improve the quality of the layout and presentation of this figure.

Thank you for this important comment. In the revised manuscript, we have adjusted Figure 2 as follows:

- The colouring and linestyle, representing the two ice-sheet models and emission scenarios, have been changed.
- Box-and-whiskers showing the Antarctic sea-level contribution projected by the ice-sheet models in 2100 and 2300 have been added in Figure 2a.
- The markers in Figure 2b, indicating the different GCMs, have been removed to improve clarity.

L151: “By 2300, the modelled Antarctic.....”?

We have added “projected” (instead of “modelled”) in the revised manuscript.

L153-154: Annotate PIG and TG on Figure 3 or remove reference to them here. Why are these glaciers mentioned specifically, when there are no glaciers referred to for Weddell Sea/Siple Coast?

This is a good point. We have removed Pine Island and Thwaites glaciers from the text, and now refer to the Amundsen Sea Embayment.

L156: why do Wilkes and Recovery “potentially” lose mass? Geomorphologically, it is perhaps arguable whether Recovery is a “subglacial basin”, in the sense of Aurora and Wilkes, too. Perhaps refer to Recovery and Wilkes catchments?

We find that grounding-line retreat in the Wilkes and Recovery catchments may occur by 2300 under the SSP5-8.5 emission pathway for some ensemble members, but is overall less likely than retreat in West Antarctica. This is reflected by a probability of being ungrounded of < 0.5 in Figure 3c and d.

In the revised manuscript, “subglacial basin” has been changed to “catchment”.

L163-167: This seems to be a key message of this paragraph, but is lost at the end, almost as a throwaway comment “In fact.....”. It deserves a paragraph of its own, and some additional detail. It may be the key message of this manuscript that is of interest to a wider audience (e.g. a non-discipline specific audience).

Thanks for pointing this out. In the revised manuscript, a separate paragraph is focused on the Antarctic sea-level contribution under the SSP1-2.6 emission pathway, starting with this key message.

L168: Suggest “aligns broadly” is changed to “are consistent with”

Ok.

L171-172: References are required after “...loss to 2300.”

Given the restructuring of this section, this sentence has been removed in the revised manuscript.

L183-185: Previously, the manuscript makes a big argument about there being a lack of modelling up to 2300, but here it reports that there the ice sheet model intercomparison project has modelling up to 2300. It would be helpful if the authors could explain this apparent contradiction, given that model runs to 2300 are argued as being something the authors do here that is ‘different’ from previous model experiments.

We understand the confusion. As developed above, projections of the Antarctic ice-sheet evolution through 2300 have indeed been carried out in ISMIP6 (Seroussi et al., 2024), contributing to filling the gap in global sea-level projections beyond 2100 in IPCC-AR6. However, our study goes beyond the ISMIP6-2300 ensemble in several key aspects. In addition to considering a wide range of emissions, from low to very high (balancing SSP1-2.6 and SSP5-8.5), we extend the ISMIP6 framework by (i) systematically sampling parametric uncertainty in ice–climate interactions in a coordinated ensemble design across two continental-scale ice-sheet models, and (ii) calibrating the projections against historical observations. This has been clarified in the revised Introduction.

L185-190: More information is required here about why the lack of an overlap is important. What does this finding tell us about the likely processes operating on the Antarctic Ice Sheet and how realistic MICI is?

Our ensemble shows a higher upper bound than ISMIP6-2300 (Seroussi et al., 2024), but is still below the uncertainty range provided by DeConto et al. (2021) using an ice cliff collapse parameterization. However, it should be noted that DeConto et al. (2021) sample a very limited range of uncertainty associated with the Marine Ice Cliff Instability and not any other parameter or climate model uncertainty, which would probably alter these ranges towards an overlap (Li et al. 2023). This sentence has been rephrased as “*Yet, even with the strong SSP5-8.5 warming trajectories projected by some of the GCMs considered here, our estimates remain well below the range projected by a single ice-sheet model that considers the Marine Ice Cliff Instability (MICI; +6.87 m-+13.55 m of sea-level equivalent under RCP8.5). These projections sampled only a limited subset of MICI-related uncertainty and did not account for broader parametric or climate model uncertainties, which would likely alter these ranges (Li et al. 2023).*”

L236-257: As a non-modeller, I found it difficult to takeaway the key messages from this section. Perhaps it is a failing on my part, but the alternative is that it is because the key messages the authors are trying to convey are not being signposted effectively enough.

In the revised manuscript, we have completely restructured and reformulated this section on the “Role of uncertainties in modulating future Antarctic mass loss”. In addition, we have included an ANOVA-based quantification of the relative contribution of the different sources of uncertainty to the spread in Antarctic sea-level projections, as suggested by Reviewer 3. We hope that these revisions clarify the key messages of this section.

L258-266: Moving this section earlier in this section (perhaps to current line 240?), might likely help address the issues I found with L236-256. This is a key message to convey. Move it to the start of the section, headline what is important, and then provide the detail.

Given the restructuring of the section on the “Role of uncertainties in modulating future Antarctic mass loss”, this paragraph on high-end sea-level contribution – which was distracting from the key messages – has been removed in the revised manuscript.

L268-270: A section opening sentence that does not really say very much. Authors could delete this entirely and simply start with L271, or they could make the statement more specific and less generic. They could consider moving lines 302-305 to the start of this section too.

We have reformulated this statement to be more specific, by briefly stating the key ‘certainties’ that our analysis reveals with respect to the committed response of the Antarctic Ice Sheet by year 3000.

L274: rather than “end of this millenium, be specific and say year 3000.

We have changed “end of this millennium” to “3000” in the revised manuscript.

L273-277: a long sentence. Authors should consider breaking it into multiple sentences.

Thanks. We have split up this sentence in the revised manuscript.

L278: What is meant by “tipping dynamics”? If they are hinting at tipping points here, then the end of the sentence (i.e. “very likely long-term ice sheet retreat”) does not fulfil the definition of a tipping point. As the authors do not assess potential irreversibility in their experiments, I recommend removing reference to “tipping” or “tipping points”.

We have changed “Tipping dynamics” to “Self-sustained ice loss”.

L284-286 (& follow on L287-289): I don’t find this finding a surprise given the different glaciological and topographic geometries of these two basins. Have previous modelling publications suggested otherwise?

Thank you for this point. Indeed, the higher likelihood of grounding-line retreat in the Wilkes catchment compared to the Aurora catchment under higher-emission scenarios is consistent with their distinct glaciological and topographic settings (Morlighem et al., 2020). Previous modelling studies have highlighted the potential vulnerability of both catchments (e.g., Golledge et al., 2015; DeConto & Pollard, 2016; Garbe et al., 2020; Ritz et al., 2015), but have not provided historically-calibrated projections to 2300 and beyond that would allow a direct, quantitative comparison of the multi-centennial evolution of Wilkes versus Aurora. Our framework addresses this gap, linking observed and projected changes to long-term dynamics and providing more robust estimates of future retreat across timescales. In the revised manuscript, these aspects have been added.

L294: state year 3000, rather than “end of millenium”?

We have changed “end of this millennium” to “3000” in the revised manuscript.

L296: Instead of “likely as not”, what about “highly likely”?

Throughout the whole manuscript, we follow IPCC language (see Box 1.1 in Chen et al., 2021) when it comes to likelihood statements as the one in L296. In the revised manuscript, we would like to continue following this language, and thus use “about as likely as not”. We have added a reference to this IPCC language at its first use, and have put all likelihood terms in italics for clarity.

L297: “...reflecting the trend in the ice sheet model experiments up to 2300....”?

We have changed “reflecting the trend in the ice sheet model experiments up to 2300” to “reflecting the trends projected by the ice-sheet models up to 2300”.

L299: Be specific about what is meant by “over the next centuries”. 2 centuries? 3, 4?

We have changed “over the next centuries” to “at least until 3000”.

L310-311: I’d argue that L312-348 are mainly about model uncertainties and limitations rather than “Next decades decisive for Antarctica’s long-term future”. As such, I recommend the authors consider changing the title of this section.

We agree. We have changed the title of this section to “Remaining uncertainties in the timing of ice loss”, and kept the title “Next decades decisive for Antarctica’s long-term future” for the concluding section.

L312-315: Another rather general opening sentence, that doesn’t really make a clear statement about the findings of the modelling experiments undertaken in this study. For the manuscript to be more effective, it needs to have much more targeted and hard-hitting opening sentences for each section. I recommend that the authors carefully review all parts of the manuscript to ensure that it is fully delivering the key messages they want it to make. At present, there is a bit of a disconnect between what I suspect they think they are saying, and what they are saying in the manuscript. The key messages need to be at the forefront. For example, if ensemble modelling of the type done here (historically calibrated, running to 2300 (and/or 3000), with assessment of parametric assumptions etc.) is vital, and not done in previous studies, then make it clear what it enables. What is the step change? E.g. “Historically-calibrated ensemble modelling enables us to deliver XYZ.....”.

Thank you for this very helpful comment. Following this advice, we have reformulated major parts of the manuscript to better highlight the key messages. In particular, in this section, we clearly emphasize the step change provided by our study: historically-calibrated ensemble modelling in a Bayesian framework to 2300 under SSPs (balancing low- and very-high emissions), combining two ice-sheet models, and systematically sampling parametric uncertainties. This allows us to provide robust, probabilistic projections of the Antarctic contribution to sea-level rise, addressing a gap in IPCC-AR6. We hope these revisions clarify the main findings of our study and their significance.

L314: Again there is a mention of “coastal adaptation planning”, but no details on specifically how the work done here feeds into such activities.

We have changed the focus of this statement to reflect the need of robust multi-centennial projections for “long-term policy and decision-making” instead of “coastal adaptation planning”.

L324-348: All about model uncertainty and limitations, rather than ‘on-topic’ (i.e. next decades decisive.....) for this section. If the authors persist with this content here (perhaps with a modified section heading), it would be good if the authors can evaluate the implications of a model horizontal resolution of 16 km on their model results. This must have implications for modelling retreat via Wilkes vs Aurora Basin, simply because of the widths of the basin entry ‘channels’.

Thank you for this suggestion. We have added a note at the end of this section to acknowledge that our ensemble uses a horizontal resolution of 16 km, which may affect the representation of narrow basin entry channels (e.g., in Wilkes and Aurora catchments) and thus influence the timing of grounding-line retreat.

L351: “some of which may be irreversible” – the authors may wish to reconsider this given that they ‘only’ model up to year 3000.

We have changed “some of which may be irreversible” to “some of which may be irreversible on human timescales”. We agree that we do not specifically assess the reversibility of the projected ice-sheet changes here. However, based on previous modelling studies (e.g. Garbe et al., 2020) and our process-understanding with respect to relevant feedback mechanisms (e.g. Weertman, 1974; Schoof, 2007), together with response timescales of the Antarctic Ice Sheet (reaching up to multiple centuries to millennia; e.g. Golledge et al., 2015; Klose et al., 2024), we can expect these changes to be irreversible on human timescales (Lee et al., 2021).

L349-359: this content is more relevant to the section heading than lines 324-348. However, it doesn’t really present anything that we didn’t really know already from other work (e.g. substantial long-term Antarctic changes....will occur unless massive mitigation efforts are made). The authors need to make a much stronger case for why what they have done here provides something new and novel. What is the original contribution to knowledge they are making? We know that the “next few years and decades are decisive for the multicentennial fate of the Antarctic Ice Sheet.” What is it that this study has done that is different from previous work? The message that needs to be provided in bold letters is probably the one that the authors outline in the intro section, the rigour of the modelling and analysis that is done here (i.e. assessment of parametric uncertainty, model historical-calibration etc.). However, the way the paper is currently written doesn’t relay that message effectively at the present time.

As mentioned in our response to previous related comments, we have significantly restructured and reformulated the manuscript and hope that those key messages are now efficiently conveyed.

**Neil Ross
Newcastle University**

Reviewer #2 (Remarks to the Author)

Coulon et al. use multiple models and statistical sampling to project long term sea level rise commitments under different emissions pathways. They find that near term projections are strongly influenced by ice sheet model sensitivities, especially under limited warming scenarios. However, long-term projections are dominated by changes in Antarctic climate, particularly under unmitigated emissions. The study emphasises the importance of strong emissions reductions today to prevent multi-metre contribution to sea level rise from both the East and West Antarctic Ice Sheets.

General comments

I found this to be an impressive study and, with a few minor edits and some further clarifications, this paper would be a useful addition to Nature Comms. It is a significant and much-needed contribution to sea level rise projections over multi-centennial timescales. As the authors note, there are currently few studies that have looked at ice sheet changes over longer timescales. The use of multiple models, their calibration with historical data, use of statistical sampling, and consideration of various emission scenarios provide an exceptionally robust and comprehensive analysis of the future evolution of the Antarctic Ice Sheet and its potential contribution to sea level rise. Furthermore, the authors account for a wide range of uncertainties in climate and ice sheet responses. The methodology is especially comprehensive and both the interpretation of the data and conclusions drawn are sound.

We thank the reviewer for this positive evaluation of our manuscript, and have included the specific comments given below in the revised version.

Specific comments

L49-52 Add reference to: J.L. Bamber, M. Oppenheimer, R.E. Kopp, W.P. Aspinall, R.M. Cooke, Ice sheet contributions to future sea-level rise from structured expert judgment, Proc. Natl. Acad. Sci. U.S.A. 116 (23) 11195-11200, <https://doi.org/10.1073/pnas.1817205116>; (2019). The Pollard and DeConto model isn't the only approach that suggests high SLR contributions.

Thank you for suggesting this reference. In the revised manuscript, the Introduction has been restructured and reformulated. In particular, we have included the IPCC-AR6 assessment assigned medium confidence, and have added a note on potentially higher upper bounds in the Antarctic sea-level contribution, as estimated when including MICI (DeConto et al., 2021) or in the structured expert judgement by Bamber et al. (2019).

L55-59 Clarity of these sentences could be improved. Replace the colon with a full stop and make this section more concise.

Agreed. In the revised manuscript, we have restructured and reformulated the Introduction.

L64 No need for brackets here.

Ok.

L67-73 Consider rephrasing to emphasise the importance of near-term emissions reductions, as this has clearer policy relevance than adaptation planning on multi-centennial timescales.

Thank you for the suggestion. In the revised manuscript, we have restructured and reformulated the Introduction. We feel that the emphasis on near-term emission reduction is no longer relevant here. However, we address this aspect in the final section of the revised manuscript

L75 Please state how far “beyond”

We have added “to the end of this millennium” in the revised manuscript.

L78-81 Consider adding some examples of parametric uncertainties.

We have added examples for parametric uncertainties in the revised manuscript.

L120-140 Links go to the wrong tables and figures.

Thanks for pointing this out. We have made sure that the links go to the correct tables and figures in the revised manuscript.

L120 Adding a brief explanation for using Latin Hypercube sampling would help readers unfamiliar with this method understand its advantages. Add references to previous use for some helpful context.

We agree. We have added a brief explanation of the Latin Hypercube sampling and respective references in the revised manuscript.

L171-172 “Only a few individual studies have explored the Antarctic ice loss to 2300.” Add references here.

In the revised manuscript, a modified version of this sentence has been moved to the Introduction, and relevant references have been added.

L107, L213, L245, L261 and Fig 1 caption: Change to circum-Antarctic

We have changed “circumantarctic” to “circum-Antarctic” in the revised manuscript.

Figure S1: Consider using a larger scale.

Done.

Reviewer #3 (Remarks to the Author)

My comments can be found in my review attachment.

Summary

This article examines the future of the Antarctic Ice Sheet (AIS) and its impact on sea level rise using two ensemble models calibrated within a Bayesian framework for two extreme emission scenarios: low emission (SSP1-2.6) and very-high emission (SSP5-8.5). Each ensemble addresses a broad spectrum of uncertainties associated with ice-climate interactions (ocean and atmosphere). The findings indicate that uncertainties in historical and short to long-term projections (up to 2100) predominantly stem from the selection of the ice sheet model (Kori or PISM), ice-climate interactions, and the climate model (AOGCM). In contrast, over multi-centennial to millennium timescale (from 2100 to 3000), the chosen emission scenario emerges as the most significant uncertainty factor. Notably, under the extremely high emission scenario (SSP5-8.5), there is a high likelihood of substantial mass loss from the West Antarctic Ice Sheet (WAIS) and a significant probability of loss from certain subglacial basins of the East Antarctic Ice Sheet (EAIS), such as Wilkes. Conversely, adhering to the Paris Agreement and limiting warming to 2 degrees (SSP1-2.6) considerably reduces the likelihood of WAIS loss and prevents self-sustained ice loss from the EAIS.

I enjoyed reading this manuscript, which is well-written and succinct! The figures are clear, as are the introduction, methods, results, and discussion sections. I believe this is an important paper and serves as a valuable complement to the ISMIP work on the AIS. It perfectly complements Seroussi et al. (2024) by more thoroughly exploring ice-climate interactions and SSP uncertainties. The goal of reaching a broader audience has been very successfully achieved by using terms characterizing uncertainties similar to those used by the IPCC (very likely, virtually certain, etc.).

Thank you for the positive feedback, and the very constructive and relevant suggestions provided in the following comments.

Major comments

I really appreciate the work of this team, and this article seems like an excellent culmination of their previous research. However, I am somewhat uncertain about the added value of this paper, as it primarily appears to summarize the findings for a broader audience from Klose et al. (2024). The main distinction seems to be the inclusion of the Bayesian calibration proposed in Coulon et al. (2024), which significantly affects the results for Kori due to differences in the prior ensemble compared to Klose et al. (2024), but has minimal impact on the PISM results. Additionally, there seems to be insufficient evidence demonstrating the superior performance of the posterior ensemble over the prior ensemble during the

observation period. Figure A1 indicates a significant reduction in uncertainty for the Kori posterior ensemble, yet this is not observed for PISM, and the posterior median occasionally falls outside the observational range for both models.

Thank you for this constructive comment.

We would like to clarify that the work presented here is not a summary of previous findings from Klose et al. (2024), but is based on new ensembles of projections specifically designed for this study. In these ensembles, various sources of uncertainty are considered to provide a robust assessment of the future evolution of the Antarctic Ice Sheet through 2300 and extended to the end of this millennium. Compared to Klose et al. (2024), parametric uncertainty in ice–climate interactions are consistently included here for both Kori-ULB and PISM, next to uncertainty in future emissions and climate model uncertainty, which represents a substantial advance: To the best of our knowledge, such a comprehensive assessment of the Antarctic ice-sheet changes beyond the end of this century, especially for the SSPs and calibrated with observations, does not exist so far. It directly addresses the ‘low confidence’-statement on the estimated sea-level contribution from the Antarctic Ice Sheet through 2300 in IPCC-AR6 (Fox-Kemper et al., 2021) and substantially adds to Seroussi et al. (2020) and Seroussi et al. (2024). To clarify the novelty of our work, the Introduction has been significantly restructured, following comments from Reviewer 1. For a broader discussion of the novelty of our study, we also refer to our response to the related general comment of Reviewer 1.

That said, we agree that the added value of the Bayesian calibration was not sufficiently clear in the original manuscript. The calibration has been revised and its benefits are demonstrated using Continuous Rank Probability Scores, as suggested below. In addition, explanations regarding the stronger reduction in uncertainty for the Kori-ULB ensemble than for the PISM one, and the median occasionally falling outside of the observational range are provided in our responses to specific comments below.

To enhance the paper and further distinguish it from Klose et al. (2024), I suggest two modifications:

1. Rewrite the section “Role of uncertainties in modulating future Antarctic mass loss”: Your ensemble adds substantial value to the findings of Seroussi et al. 2023 and 2024 by delving deeper into ice-climate interactions and SSP uncertainty. It would be beneficial to include a figure showing the time evolution of Sobol indices/percentage of variance (similar to Fig. 9 in Seroussi et al. 2023, Fig. 15 in Seroussi et al. 2024, and Fig. 6 in Jager et al. 2024). This would provide strong visual support for this section and clarify which sources of uncertainty are most significant up to the year 2300. Additionally, a new paragraph comparing this with Fig. 15 from Seroussi et al. 2024 could emphasize the importance of considering uncertainty from ice-climate interactions and SSP.

We would like to thank you for this very relevant suggestion. In the revised manuscript, we have quantified the contribution of the different sources of uncertainty included in our ensembles by calculating the percentage of variance using ANOVA, both by combining the PISM and Kori-ULB ensembles to quantify the structural uncertainty, and separately for each

ice-sheet model ensemble. These results are shown in Figures 4 and 5 of the revised manuscript, as well as in Extended Data Figure 3. We believe this addition significantly improves the manuscript and provides strong visual support for the discussion on the role of the different sources of uncertainty.

The section has been rewritten accordingly, with a clearer statement on the added value of our approach and a comparison to the ISMIP6-2300 ensemble.

2. Demonstrate the improvement of the ensemble through Bayesian calibration:

In Jager et al. 2024, I proposed a methodology to determine whether the posterior ensemble is better than the prior ensemble. Similar to practices in machine learning and hydrology, it involves splitting your observation dataset into a training set and a testing set. I believe this method could be easily applied to your study, but if you have another approach to demonstrate the benefits of Bayesian calibration, I would be delighted to learn about it. Here is a suggestion for implementing it:

a. Split the ensemble into a training set and a testing set. Given that you have 18 observations across 3 sectors, consider using observations from only 2 sectors (5 per sector) as the training set, with the remainder as the testing set.

b. Use the training set for Bayesian calibration to determine your weights.

c. Evaluate the performance of the posterior ensemble against the prior one using the testing set. Various metrics, such as the continuous rank probability score (CRPS) from Eq. 3 in Jager et al. 2024, can be used for this comparison. Additional metrics are available here: [<https://www.cawcr.gov.au/projects/verification/>].

d. Restart with another training and testing set.

Again, thank you for the relevant comment and suggestion. You are correct to say that the effect of the calibration is more evident for the Kori-ULB ensemble than for the PISM ensemble. We attribute this to the following two main reasons:

- The PISM ensemble produces a much smaller spread than the Kori-ULB one by design (less ice–ocean uncertainties were explored). Obtaining a larger reduction in variance from the PISM prior to posterior would require reducing the model discrepancy, but this would imply giving more weight or trust to one ice-sheet model over the other, which is something we wanted to avoid.
- The PISM ensemble underestimates the observed mass loss of the West Antarctic Ice Sheet. In contrast, for Kori-ULB, a model drift leads to an overestimation of mass gain in East Antarctica. This influences the efficiency of the calibration using the regional mass balance estimates from IMBIE, which is more pronounced for the PISM ensemble than for Kori-ULB (given that West Antarctica dominates the historical mass changes signal).

For these reasons, we have revised our calibration and now use continental-scale observational estimates instead of regional ones. This allows for a more efficient calibration of the PISM ensemble.

However, this reduces the number of observational constraints (6 values), making it difficult to apply the cross-validation methodology you suggest (splitting the data into training and testing sets). That said, we agree that it is helpful to use a quantitative metric to assess the

influence of the calibration. Therefore, we now calculate the Continuous Rank Probability Score (CRPS) for Antarctic-wide, West Antarctic, East Antarctic, and Antarctic Peninsula mass-balance observations over the 1992-2020 period using IMBIE estimates (Otosaka et al., 2023). The normalised CRPS values, as well as the comparison between prior and posterior distributions, are now shown in Extended Data Figure 1. We hope this effectively illustrates the benefits of the calibration for both ice-sheet models.

We also acknowledge the limitation of calibrating against continental rather than regional metrics. To compensate for the model drift towards mass gain in East Antarctica in Kori-ULB, ensemble members characterized by a slight overestimation of West Antarctic mass loss are favored in the calibration. For PISM, the calibration tends to favor ensemble members that overestimate mass loss in the Antarctic Peninsula to offset the underestimation of West Antarctic mass loss. A similar statement has been added to the section “Bayesian calibration of the multi-model ensemble” of the revised manuscript.

I've noticed that some statements in the paper may not fully align with the widely accepted understanding of SSPs, emissions, and mitigation efforts, or at least differ from my personal understanding. This issue seems to extend beyond your study and reflects a broader trend in our community, where SSP5-8.5 is still often treated as a 'business-as-usual' scenario, which it is not. According to the latest IPCC Summary for Policymakers (2022), the policies currently implemented are projected to lead to a warming of about 3.2°C (ranging between 2.2°C and 3.5°C, with medium confidence). This projection is slightly higher than SSP2-4.5 (around 2.8°C) but lower than SSP3-7.0 (around 3.9°C), and significantly lower than SSP5-8.5 (around 4.8°C). Other literature also suggests that we are not currently on a trajectory consistent with SSP5-8.5 (Raftery et al., 2017; Hausfather and Peters, 2020; Hausfather and Moore, 2022; Pielke et al., 2022).

Given this context, you have two options:

1. Perform new simulations with an intermediate SSP such as SSP2-4.5: This approach would allow you to explore whether current policies are sufficient to prevent significant contributions from the AIS, or if more aggressive policies are needed. It also provides valuable information to stakeholders by representing a 'most probable/likely path', serving as a median path between two less likely extremes.

2. Revise some of your statements downwards: While this option is simpler as it does not require running new simulations (and I understand that there are limited climate models that have run until 2300 with SSP2-4.5, though some exist from CMIP5 with RCP4.5, e.g., Hezel et al. 2014), it would significantly reduce the impact of your paper. Specifically, it would necessitate revising the last high-impact sentence in the abstract: 'Our results imply that strong mitigation efforts are needed in the coming years to decades to reduce the risk of self-sustained sea-level rise over the next centuries.'" In this second case, it would also be appropriate to add a paragraph to discuss this limitation of the study in the 'Next decades decisive for Antarctica's long-term future' section. This would be an opportunity to push for studies with better sampling of SSPs, which seem to be the main source of uncertainty after 2100.

Thank you for this important comment.

We agree that SSP5-8.5 does not correspond to a 'current policy' or a 'business-as-usual' scenario. For clarification, in the revised version, we have described SSP5-8.5 as "very high emissions" (see also our response to a later comment).

That said, we still believe that assessing Antarctic ice-sheet changes under this very high-emissions pathway remains valuable. For example, given that important processes and feedbacks (for example related to the carbon cycle) remain not (or only partially) represented in climate models (Canadell et al., 2021; Chen et al., 2021; Riahi et al., 2022), high-end warming of the magnitude projected under SSP5-8.5 cannot be fully excluded. In this context, our results provide insights into this potential tail of climate futures.

We also fully agree that additional projections under a middle-of-the-road scenario such as SSP2-4.5 would be important to better characterize future ice-sheet trajectories under different climate policies. SSP2-4.5 has, however, only been assessed over this century in CMIP6 (O'Neill et al., 2016). Therefore, no SSP2-4.5 climate projection was available at the time of this assessment, and, to our knowledge, only a single GCM has provided SSP2-4.5 (GISS-E2-1-G) to 2300 since. While climate projections for RCP4.5 are available after 2100, we would here like to use the updated SSPs. For these reasons, it is currently not possible to consistently include such a middle-of-the-road scenario within our probabilistic framework over multi-centennial timescales. We have added a sentence emphasising the need for a better sampling of SSPs beyond 2100 in climate model projections in the Discussion.

By bracketing future forcing from the SSP1-2.6 to the SSP5-8.5 pathways, we are able to explore the full range of scenarios used in IPCC-AR6 in a balanced way. We can thus substantially extend previous estimates with a strong focus on higher-emission pathways (e.g. Fox-Kemper et al., 2021; Box 9.4; Seroussi et al., 2020; Seroussi et al., 2024). By doing so, we show that while following the SSP1-2.6 pathway strongly limits the projected multi-centennial Antarctic ice loss, a long-term collapse of the West Antarctic Ice Sheet cannot be fully excluded, and the associated sea-level rise could still be significant. By contrast, under SSP5-8.5, a West Antarctic Ice Sheet collapse is virtually certain in our ensemble. Therefore, even without specifically quantifying the ice-sheet response under a middle-of-the-road-scenario, our results suggest that both the likelihood of a long-term West Antarctic Ice Sheet collapse and the magnitude of its sea-level contribution increase for emissions above SSP1-2.6. This suggests that current mitigation efforts may not be sufficient to avoid self-sustained Antarctic ice loss. We acknowledge that the original manuscript did not present the results in this direction, which we have corrected in the revised version.

I hope these suggestions seem relevant to you and will help improve the manuscript. My major comments may be somewhat lengthy, but I would like to reiterate my congratulations on this excellent article, which is truly impressive. I have cited my previous work, but that doesn't necessarily mean you must cite it as well. I've also noted some minor corrections below, and at the end, you'll find the references I've mentioned. Hope it helps, and good luck with your revisions!

Eliot Jager, University of Helsinki

Minor comments

L39-40: Without results with SSP2-4.5, the sentence is overstated. SSP2-4.5 is currently the path closest to 'business as usual'. This does not mean that other paths are not possible, but 'strong mitigation efforts' today aim to shift from a trajectory of +3°C (SSP2- 4.5) to 2°C (SSP1-2.6), not from +5°C (SSP5-8.5).

Following our response on the interpretation of ice-sheet changes under SSP1-2.6 above, we have replaced this sentence by: *“Yet, even under strong mitigation, a significant amount of sea-level rise could still result from West Antarctic ice loss. Our results suggest that current mitigation efforts may not be sufficient to avoid self-sustained Antarctic ice loss, making emission decisions taken in the coming years decisive for future sea-level rise.”*

L54: you can cite Aschwanden et al. (2021) here.

Thank you for this suggestion. We have included the reference in the revised manuscript.

L59: Would Seroussi et al. (2024) be a more recent/relevant reference here?

In the revised manuscript, we have restructured and reformulated the Introduction. A reference to Seroussi et al. (2024) has been included.

L64: It's a bit odd here to cite the IPCC report, as it does not produce projections, and the fact that one section in this report is devoted to assessing 'Projections Beyond 2100.' The two other references are sufficient.

We have removed this reference.

L67-68: I'm not sure which reference the term “multi-centennial timescales” comes from. In Hinkel et al. (2019), 'long-term' is defined as “longer than 30 years,” with a typical horizon of “30 to 100 years and more.” Nuclear plants can be considered, but their typical lifetime is less than 200 years. Nichols et al. (2021) mention post-2100 but do not provide a typical range. In Wal et al. (2022), they indeed assess up to 2300, but I'm not convinced that many stakeholders in coastal adaptation planning are interested in such a timescale. While there are certainly some, italicizing it gives the impression that most, if not all, require such information, whereas they are primarily interested in the short term.

Thank you for this helpful comment. To avoid confusion, the reference to coastal planning stakeholders has been removed.

L69-70: I prefer the term “ice sheet model structure” over “ice-sheet model” to distinguish it from “parametric uncertainty.”

Agreed. We have replaced “ice-sheet model” by “ice sheet model structure” or “structural uncertainty” in the revised manuscript, where applicable.

L142-143: I agree in terms of the median, but isn't there a significant difference in the 90% confidence interval? I'm not completely certain about this; Graph 2.a is not very clear. Could you add different boxplots of your results next to E21? And/or add hatching to the shading?

Thank you for your helpful suggestions on Figure 2. Following your advice, we have revised this figure to improve readability: in addition to updating the coloring and line styles representing the two ice-sheet models and emission scenarios, we have added box-and-whiskers showing the Antarctic sea-level contribution projected by the ice-sheet models in 2100 and 2300.

We agree that there are some differences in the 90% confidence intervals between the emission scenarios, especially for the PISM ensemble. However, the median projections of each ice-sheet model are very similar for the emission scenarios, and the SSP5-8.5 ranges consistently overlap those of SSP1-2.6. Therefore, we believe that the original statement regarding the little scenario-dependance of the projections by 2100 is valid.

Fig. 4: It is difficult to compare panels c and d due to the difference in units.

Agreed. This figure has been removed and replaced by Figure 5 in the revised manuscript, showing the explained variance in the Antarctic contribution to sea-level change based on the ANOVA applied to each ice-sheet model.

L217-222: I am quite convinced by the explanation, but it also appears that the PICO-SSP-2.6 members from Kori exhibit a positive correlation, or at least the trend is not flat (as indicated by the empty diamond in Fig. 4.a).

This is correct. We attribute this stronger sensitivity of the Kori-ULB ensemble even when using PICO to the different initialisation approaches applied to the ice-sheet models. It is very likely that the spin-up approach used for PISM makes it less sensitive to oceanic changes than Kori-ULB, initializing with a transient inverse simulation.

In the revised manuscript, the difference in sensitivity between the two ice-sheet models is much clearer due to the ANOVA uncertainty quantification that you suggested. We have therefore removed the reference to a lower sensitivity of PICO, and instead emphasize differences in ensemble design and initialization approaches.

Section "Certain long-term future of the Antarctic Ice Sheet":

Please define "long-term" somewhere. I suggest using "very long-term" or "multi-centennial timescale" for a 2500 horizon and "millennium timescale" for a 3000 horizon.

The "long-term" future of the Antarctic Ice Sheet refers to the (committed) ice-sheet evolution until the year 3000. This has been specified in the first sentence of this section. In addition, we have clarified the timescales of assessment throughout the entire revised manuscript.

L320: "Under high warming" is not the most accurate description for SSP5-8.5. "Excessively high warming," "low-probability high warming," or "very high warming" might be more precise. I believe "very high warming" is the term used in the latest IPCC report.

We have replaced "high warming" by "very strong warming" in the revised manuscript.

L326-327: In your sentence, you state that the "onset of collapse and future rates" depend mainly on "model uncertainties," but they also depend significantly on the future level of warming/human emissions! You might refer to this uncertainty as "uncertainties in future human emissions."

We have reformulated this sentence to better reflect the role of future emissions as a source of uncertainty: *"while our results indicate that the long-term fate of the ice sheet is strongly determined by the emission scenario, uncertainties remain regarding the timing and pace of multi-decadal to centennial sea-level rise"*. This part of the Discussion has also been substantially restructured in the revised manuscript to emphasize the remaining uncertainties in the timing and pace of ice-sheet mass loss. We hope that this clarifies the message.

L342: "in Antarctic viscoelastic properties" -> do you mean the mantle of the AIS?

We mean both the viscosity of the mantle and the thickness of the lithosphere beneath the Antarctic Ice Sheet. We have reformulated this sentence for clarification.

L345: "ice-sheet model uncertainties" -> here, I think you are referring to the model structure, i.e., the uncertainty arising from using two different ice sheet models, not the parametric uncertainty of each. I recall pointing this out elsewhere; please check if other parts of the text need revision for consistency and use the term "structure." It doesn't have much to do with this comment, and there's no definition of 'model structure', but on these questions of sensitivity analysis this ebook is very cool: <https://uc-ebook.org/>

Indeed, thank you for pointing this out. We have clarified this throughout the revised manuscript. In particular, we have changed "ice-sheet model" to "ice-sheet model structure" or "structural uncertainty", if applicable. And thanks for the interesting and useful reference!

L351-352: "unless massive mitigation efforts are made" -> it would be necessary to include results with SSP2-4.5 to assess this. Perhaps this pathway alone is sufficient to prevent "substantial long-term Antarctic changes," without the need for massive mitigation efforts.

Following our response on the interpretation of ice-sheet changes under SSP1-2.6 above, this sentence has been modified to *"Our study shows that substantial long-term Antarctic changes -- some of which may be irreversible on human timescales -- may occur even if massive mitigation efforts are made."*

L354-355: "Unmitigated emissions" more closely aligns with SSP2-4.5, not SSP5-8.5.

Discussions for CMIP7 seem to suggest that this scenario might not be included in CMIP7 (e.g., van Vuuren et al. (2025)), or it could be referred to as “the emissions world avoided” (TEWA) scenario (Malte Meinshausen et al. (2024)).

Thank you for pointing this out. We have reformulated the description of the different emission pathways throughout the manuscript. In particular, “unmitigated emissions” (or similar) is replaced by “very high emissions”.

L358-359: Same here, past years have been decisive to pass from SSP5-8.5 to SSP2-4.5, but is it enough? I think this is a really interesting question, but would require to test at least SSP2-4.5.

We agree that estimates of the Antarctic sea-level contribution under a middle-of-the-road scenario such as SSP2-4.5 would be generally very valuable, and have called for the climate modelling community to provide more projections beyond 2100. Nevertheless, even under SSP1-2.6, a collapse of the West Antarctic Ice Sheet cannot be excluded. Following the revised interpretation of ice-sheet changes under SSP1-2.6 (discussed above in more detail), we have reformulated this sentence as “*These findings suggest that ongoing mitigation efforts may not be sufficient to prevent self-sustained Antarctic ice loss, making decisions on emissions reductions taken in the coming years and decades decisive for multi-centennial sea-level rise.*”

“Ice-sheet models” section:

It would be beneficial to include a table with one column for each model to provide a quick overview of the common and differing characteristics of the two models (take a look to ISMIP papers for examples).

Thank you for the suggestion. We have included such a table in the Supporting Information (Supplementary Table 1) of the revised manuscript.

L404: What is the exponent of the power-law? Additionally, it's not clear which specific equation from the cited paper is being used here.

We have specified the sliding law used in PISM as well as the choice of the exponent in the revised manuscript.

L493-494: The sum of 400 and 300 differs from the 1400 members mentioned earlier. Please clarify that you create one ensemble for each SSP.

We have clarified the design of the ensembles and, in particular, the number of ensemble members in the revised manuscript.

“Calibration” section:

- Do you apply a weighting to each model, or do you combine members from both models?

We calibrate each ice-sheet model's ensemble independently using the same Bayesian framework. This allows us to ensure equal representation of both ice-sheet models. This has been clarified in the revised manuscript.

- Does it make sense to have the same structural model error, σ_{mod} , for both model? Perhaps one model is inherently better? This can be assessed using the method I proposed in my major comment.

The structural error here is intended to account for the fact that models are a simplification of reality, which is a limitation common to both ice-sheet models. Assigning different structural errors to each ice-sheet model would imply giving more weight or trust to one model over the other, which is something we wanted to avoid. This approach also accounts for the possibility that one model may agree better with the observations 'for the wrong reasons'. Note that the effect of the structural error on the distributions strongly depends on the relative breadth of the original ensemble designs.

- L524: Independence between observations is a condition that is rarely fully met in glaciology (if one catchment is losing mass, it is likely that the adjacent one is also losing mass and will continue to do so in the near future), because our systems are not chaotic.

We agree that the assumption of independence between observational constraints is a simplification and that correlations likely exist. However, as stated in the manuscript, we assume independence and normality, as commonly done in Bayesian calibration frameworks, particularly when correlation is unknown or difficult to estimate.

- Fig. A1: It appears strange that your median is further from the observation after calibration.

We understand that this is counter-intuitive. We believe that this could arise for two reasons.

First, this may be explained by the distributions' shapes. For both ice-sheet models, the prior distributions are characterized by (i) a long tail towards mass loss and (ii) a mode above the observations, resulting in a median quite in line with observations, even though the mode is too high. By applying the calibration, the posterior median is shifted away and looks worse, but the posterior modes are in fact closer to the observations, as expected in a Bayesian calibration. Note that, unless the posterior distributions were symmetric, we would not expect the medians to also align with the observations.

In addition, this can also be a consequence of the calibration being performed using regional mass balance constraints. Both ice-sheet model ensembles show regional biases, which influence the continental-scale mass balance: The Kori-ULB ensemble shows a bias towards mass gain in East Antarctica, while the PISM ensemble underestimates West Antarctic mass loss. These regional biases persist in the calibrated ensembles and lead to posterior medians falling outside the observational range. Note that the posterior median is overall closer to the observations than the prior in the revised manuscript, where only continental mass balance constraints are used for the calibration.

- In Jager et al. (2024), we found that σ_i has a significant influence, and can lead to either underfitting or overfitting. "Perhaps here, taking σ_{mod} as 10 times the observational error σ_{obs} is too large and leads to under-fitting. This can be assessed using the method I've proposed in my major comment.

We agree that the σ_{mod} is a free parameter and has a strong influence on the posterior distribution. We hope that Extended Data Figures A1e–I along with the normalized CRPS values in the revised manuscript are helpful in showing the benefit of the calibration. In addition, Supplementary Table 3 shows the effect of varying the structural error on both the posterior median and the 5–95% probability intervals. Reducing the assumed structural error does not substantially affect the posterior distribution.

To address the possibility of over-/underfitting more thoroughly, we have also added Supplementary Figure 4 in the revised manuscript, which displays the normalised CRPS values and the prior/posterior distributions under different assumptions for the structural error σ_{mod} . For both ice-sheet models, we find that assuming a structural error of 3 times σ_{obs} improves the continental-scale CRPS while limiting overfitting (at the regional scale). Note that we would like to maintain identical values of the structural error for both ice-sheet models, as discussed in our response to a previous related comment.

Fig. S2:

- Isn't it strange that MAR appears to be a better choice for Kori, but RACMO is more suitable for PISM?

Thank you for this comment. There is indeed a difference between both ice-sheet models' posterior distributions with respect to the atmospheric climatology. However, this difference is small (the prior and posterior distributions nearly overlapped), and it is now reversed in the updated calibration. It presumably arises from compensation errors in surface mass balance vs dynamic ice loss. It may also be explained by differences in how the positive degree-day schemes are implemented in the two ice-sheet models. Overall, we believe that this difference reflects a calibration artefact rather than a robust model preference.

- g vs h: Are you certain of your values? There is a significant difference in the order of magnitude between the 2 plots.

Apologies for the confusion. For Kori-ULB, the uncertainty range of Γ_{eff} depends on the value of M_{param} , which corresponds to the chosen sub-shelf melt parameterisation. Therefore, we here show the [0 1] range as defined by the Latin hypercube sampling. We have clarified this in the figure caption.

Table S2: For PISM, there are significant changes between the prior and posterior ensembles, yet when I look at Fig. A1.b, the prior and posterior ensembles seem to have similar performance during the historical period. Are you sure you are not reducing the range for the wrong reasons?

We hope that Extended Data Figures A1e–I along with the normalized CRPS values in the revised manuscript are helpful in showing the benefit of the calibration and for the discussion of the interplay between the different ice-sheet regions.

Fig. A1:

- Can you plot the same charts as in a and b but for the different sectors? I still find it strange that the calibration for PISM misses some observations in the shading and that the median of the posterior is further from the observation than the prior for Kori.

Again, we hope that Extended Data Figures A1e–I along with the normalized CRPS values in the revised manuscript are helpful in showing the benefit of the calibration. We also refer to our previous response to explain these aspects of the calibration.

- Can you add a map of observed elevation rates for visual comparison? e.g. Smith et al. 2020

We agree that a map of observed elevation change rates is useful for visual comparison. However, to avoid adding more information to an already dense figure, we have included this map in as Supplementary Figure 5 in the revised manuscript and added a reference to it in the caption of Extended Data Figure A1.

References:

Seroussi, H., Pelle, T., Lipscomb, W. H., Abe-Ouchi, A., Albrecht, T., Alvarez-Solas, J., et al. (2024). Evolution of the Antarctic Ice Sheet over the next three centuries from an ISMIP6 model ensemble. *Earth's Future*, 12, e2024EF004561. <https://doi.org/10.1029/2024EF004561>

Klose, A. K., Coulon, V., Pattyn, F., and Winkelmann, R.: The long-term sea-level commitment from Antarctica, *The Cryosphere*, 18, 4463–4492, <https://doi.org/10.5194/tc-18-4463-2024>, 2024.

Coulon, V., Klose, A. K., Kittel, C., Edwards, T., Turner, F., Winkelmann, R., and Pattyn, F.: Disentangling the drivers of future Antarctic ice loss with a historically calibrated ice-sheet model, *The Cryosphere*, 18, 653–681, <https://doi.org/10.5194/tc-18-653-2024>, 2024.

Seroussi, H., Verjans, V., Nowicki, S., Payne, A. J., Goelzer, H., Lipscomb, W. H., Abe-Ouchi, A., Agosta, C., Albrecht, T., Asay-Davis, X., Barthel, A., Calov, R., Cullather, R., Dumas, C., Galton-Fenzi, B. K., Gladstone, R., Golledge, N. R., Gregory, J. M., Greve, R., Hattermann, T., Hoffman, M. J., Humbert, A., Huybrechts, P., Jourdain, N. C., Kleiner, T., Larour, E., Leguy, G. R., Lowry, D. P., Little, C. M., Morlighem, M., Pattyn, F., Pelle, T., Price, S. F., Quiquet, A., Reese, R., Schlegel, N.-J., Shepherd, A., Simon, E., Smith, R. S., Straneo, F., Sun, S., Trusel, L. D., Van Breedam, J., Van Katwyk, P., van de Wal, R. S. W., Winkelmann, R., Zhao, C., Zhang, T., and Zwinger, T.: Insights into the vulnerability of Antarctic glaciers from the ISMIP6 ice sheet model ensemble and associated uncertainty, *The Cryosphere*, 17, 5197–5217, <https://doi.org/10.5194/tc-17-5197-2023>, 2023.

Jager, E., Gillet-Chaulet, F., Champollion, N., Millan, R., Goelzer, H., and Mougnot, J.: The future of Upernavik Isstrøm through the ISMIP6 framework: sensitivity analysis and Bayesian calibration of ensemble prediction, *The Cryosphere*, 18, 5519–5550, <https://doi.org/10.5194/tc-18-5519-2024>, 2024.

IPCC, 2023: Summary for Policymakers. In: *Climate Change 2023: Synthesis Report. Contribution of Working Groups I, II and III to the Sixth Assessment Report of the Intergovernmental Panel on Climate Change* [Core Writing Team, H. Lee and J. Romero (eds.)]. IPCC, Geneva, Switzerland, pp. 1-34, doi: 10.59327/IPCC/AR6-9789291691647.001

Rafferty, A. E., Zimmer, A., Frierson, D. M. W., Startz, R., and Liu, P.: Less than 2 °C warming by 2100 unlikely, *Nat. Clim. Change*, 7, 637–641, <https://doi.org/10.1038/nclimate3352>, 2017.

Hausfather, Z. and Moore, F. C.: Net-zero commitments could limit warming to below 2 °C, *Nature*, 604, 247–248, <https://doi.org/10.1038/d41586-022-00874-1>, 2022.

Hausfather, Z. and Peters, G.: Emissions – the “business as usual” story is misleading, *Nature*, 577, 618–620, <https://doi.org/10.1038/d41586-020-00177-3>, 2020.

Pielke Jr, R., Burgess, M. G., and Ritchie, J.: Plausible 2005–2050 emissions scenarios project between 2 °C and 3 °C of warming by 2100, *Environ. Res. Lett.*, 17, 024027, <https://doi.org/10.1088/1748-9326/ac4ebf>, 2022.

Hezel, P. J., Fichet, T., and Massonnet, F.: Modeled Arctic sea ice evolution through 2300 in CMIP5 extended RCPs, *The Cryosphere*, 8, 1195–1204, <https://doi.org/10.5194/tc-8-1195-2014>, 2014.

Aschwanden, A., Bartholomaus, T. C., Brinkerhoff, D. J., and Truffer, M.: Brief communication: A roadmap towards credible projections of ice sheet contribution to sea level, *The Cryosphere*, 15, 5705–5715, <https://doi.org/10.5194/tc-15-5705-2021>, 2021.

Hill, E. A., Urruty, B., Reese, R., Garbe, J., Gagliardini, O., Durand, G., Gillet-Chaulet, F., Gudmundsson, G. H., Winkelmann, R., Chekki, M., Chandler, D., and Langebroek, P. M.: The stability of present-day Antarctic grounding lines – Part 1: No indication of marine ice sheet instability in the current geometry, *The Cryosphere*, 17, 3739–3759, <https://doi.org/10.5194/tc-17-3739-2023>, 2023.

van Vuuren, D., O'Neill, B., Tebaldi, C., Chini, L., Friedlingstein, P., Hasegawa, T., Riahi, K., Sanderson, B., Govindasamy, B., Bauer, N., Eyring, V., Fall, C., Frieler, K., Gidden, M., Gohar, L., Jones, A., King, A., Knutti, R., Kriegler, E., Lawrence, P., Lennard, C., Lowe, J., Mathison, C., Mehmood, S., Prado, L., Zhang, Q., Rose, S., Ruane, A., Schleussner, C.-F., Seferian, R., Sillmann, J., Smith, C., Sörensson, A., Panickal, S., Tachiiri, K., Vaughan, N., Vishwanathan, S., Yokohata, T., and Ziehn, T.: The Scenario Model Intercomparison Project for CMIP7 (ScenarioMIP-CMIP7) , *EGUsphere* [preprint], <https://doi.org/10.5194/egusphere-2024-3765>, 2025.

Meinshausen, M., Schleussner, C.-F., Beyer, K., Bodeker, G., Boucher, O., Canadell, J. G., Daniel, J. S., Diongue-Niang, A., Driouech, F., Fischer, E., Forster, P., Grose, M., Hansen, G., Hausfather, Z., Ilyina, T., Kikstra, J. S., Kimutai, J., King, A. D., Lee, J.-Y., Lennard, C., Lissner, T., Nauels, A., Peters, G. P., Pirani, A., Plattner, G.-K., Pörtner, H., Rogelj, J., Rojas, M., Roy, J., Samset, B. H., Sanderson, B. M., Séférian, R., Seneviratne, S., Smith, C. J., Szopa, S., Thomas, A., Urge-Vorsatz, D., Velders, G. J. M., Yokohata, T., Ziehn, T., and Nicholls, Z.: A perspective on the next generation of Earth system model scenarios: towards representative emission pathways (REPs), *Geosci. Model Dev.*, 17, 4533–4559, <https://doi.org/10.5194/gmd-17-4533-2024>, 2024.

Ben Smith et al., Pervasive ice sheet mass loss reflects competing ocean and atmosphere processes. *Science* 368,1239-1242(2020). DOI:10.1126/science.aaz5845

References

- Bamber, J.L., Oppenheimer, M., Kopp, R.E., Aspinall, W.P., Cooke, R.M.: Ice sheet contributions to future sea-level rise from structured expert judgment. *Proceedings of the National Academy of Sciences* 116(23), 11195–11200 (2019) <https://doi.org/10.1073/pnas.1817205116>
- Beadling, R.L., Russell, J.L., Stouffer, R.J., Mazloff, M., Talley, L.D., Goodman, P.J., Sallee, J.-B., Hewitt, H.T., Hyder, P., Pandde, A.: Representation of Southern Ocean Properties across Coupled Model Intercomparison Project Generations: CMIP3 to CMIP6. *Journal of Climate* 33(15), 6555–6581 (2020) <https://doi.org/10.1175/JCLI-D-19-0970.1>
- Bracegirdle, T., Holmes, C., Hosking, J., Marshall, G., Osman, M., Patterson, M., Rackow, T.: Improvements in circumpolar Southern Hemisphere Extratropical Atmospheric Circulation in CMIP6 compared to CMIP5. *Earth and Space Science* 7(6), 2019–001065 (2020) <https://doi.org/10.1029/2019EA001065>
- Bulthuis, K., Arnst, M., Sun, S., Pattyn, F.: Uncertainty quantification of the multi-centennial response of the Antarctic ice sheet to climate change. *The Cryosphere* 13(4), 1349–1380 (2019) <https://doi.org/10.5194/tc-13-1349-2019>
- Canadell, J. G., Monteiro, P. M. S., Costa, M. H., Cotrim da Cunha, L., Cox, P. M., Eliseev, A. V., Henson, S., Ishii, M., Jaccard, S., Koven, C., Lohila, A., Patra, P. K., Piao, S., Rogelj, J., Syampungani, S., Zaehle, S., & Zickfeld, K. (2021). Global Carbon and other Biogeochemical Cycles and Feedbacks. In *Climate Change 2021: The Physical Science Basis. Contribution of Working Group I to the Sixth Assessment Report of the Intergovernmental Panel on Climate Change* [Masson-Delmotte, V., P. Zhai, A. Pirani, S.L. Connors, C. Péan, S. Berger, N. Caud, Y. Chen, L. Goldfarb, M.I. Gomis, M. Huang, K. Leitzell, E. Lonnoy, J.B.R. Matthews, T.K. Maycock, T. Waterfield, O. Yelekçi, R. Yu, and B. Zhou (eds.)] Cambridge University Press, 673–816. <https://doi.org/10.1017/9781009157896.007>
- Chen, D., Rojas, M., Samset, B. H., Cobb, K., Diongue Niang, A., Edwards, P., Emori, S., Faria, S. H., Hawkins, E., Hope, P., Huybrechts, P., Meinshausen, M., Mustafa, S. K., Plattner, G.-K., & Tréguier, A.-M. (2021). Framing, Context, and Methods. In *Climate Change 2021: The Physical Science Basis. Contribution of Working Group I to the Sixth Assessment Report of the Intergovernmental Panel on Climate Change* [Masson-Delmotte, V., P. Zhai, A. Pirani, S.L. Connors, C. Péan, S. Berger, N. Caud, Y. Chen, L. Goldfarb, M.I. Gomis, M. Huang, K. Leitzell, E. Lonnoy, J.B.R. Matthews, T.K. Maycock, T. Waterfield, O. Yelekçi, R. Yu, and B. Zhou (eds.)] Cambridge University Press, 147–286. <https://doi.org/10.1017/9781009157896.003>
- Coulon, V., Klose, A.K., Kittel, C., Edwards, T., Turner, F., Winkelmann, R., Pattyn, F.: Disentangling the drivers of future Antarctic ice loss with a historically calibrated ice-sheet model. *The Cryosphere* 18(2), 653–681 (2024) <https://doi.org/10.5194/tc-18-653-2024>
- DeConto, R.M., Pollard, D.: Contribution of Antarctica to past and future sea level rise. *Nature* 531, 591–597 (2016) <https://doi.org/10.1038/nature17145>

DeConto, R.M., Pollard, D., Alley, R.B., Velicogna, I., Gasson, E., Gomez, N., Sadai, S., Condrón, A., Gilford, D.M., Ashe, E.L., Kopp, R.E., Li, D., Dutton, A.: The Paris Climate Agreement and future sea-level rise from Antarctica. *Nature* 593, 83–89 (2021) <https://doi.org/10.1038/s41586-021-03427-0>

Fox-Kemper, B., Hewitt, H.T., Xiao, C., Aalgeirsdottir, G., Drijfhout, S.S., Edwards, T.L., Golledge, N.R., Hemer, M., Kopp, R.E., Krinner, G., Mix, A., Notz, D., Nowicki, S., Nurhati, I.S., Ruiz, L., Sallée, J.-B., Slangen, A.B.A., Yu, Y.: Ocean, Cryosphere and Sea Level Change. In *Climate Change 2021: The Physical Science Basis. Contribution of Working Group I to the Sixth Assessment Report of the Intergovernmental Panel on Climate Change* [Masson-Delmotte, V., P. Zhai, A. Pirani, S.L. Connors, C. Péan, S. Berger, N. Caud, Y. Chen, L. Goldfarb, M.I. Gomis, M. Huang, K. Leitzell, E. Lonnoy, J.B.R. Matthews, T.K. Maycock, T. Waterfield, O. Yelekci, R. Yu, and B. Zhou (eds.)]. Cambridge University Press, 1211–1362 (2021) <https://doi.org/10.1017/9781009157896.011>

Garbe, J., Albrecht, T., Levermann, A., Donges, J.F., Winkelmann, R.: The hysteresis of the Antarctic Ice Sheet. *Nature* 585, 538–544 (2020) <https://doi.org/10.1038/s41586-020-2727-5>

Golledge, N.R., Kowalewski, D.E., Naish, T.R., Levy, R.H., Fogwill, C.J., Gasson, E.G.W.: The multi-millennial Antarctic commitment to future sea-level rise. *Nature* 526, 421–425 (2015) <https://doi.org/10.1038/nature15706>

Golledge, N. R., Keller, E. D., Gomez, N., Naughten, K. A., Bernales, J., Trusel, L. D., Edwards, T. L.: Global environmental consequences of twenty-first-century ice sheet melt. *Nature* 566(7742), 65–72 (2019) <https://doi.org/10.1038/s41586-019-0889-9>

Klose, A.K., Coulon, V., Pattyn, F., Winkelmann, R.: The long-term sea-level commitment from Antarctica. *The Cryosphere* 18(9), 4463–4492 (2024) <https://doi.org/10.5194/tc-18-4463-2024>

Lee, J.-Y., Marotzke, J., Bala, G., Cao, L., Corti, S., Dunne, J.P., Engelbrecht, F., Fischer, E., Fyfe, J.C., Jones, C., Maycock, A., Mutemi, J., Ndiaye, O., Panickal, S., Zhou, T.: Future Global Climate: Scenario-Based Projections and Near-Term Information. In *Climate Change 2021: The Physical Science Basis. Contribution of Working Group I to the Sixth Assessment Report of the Intergovernmental Panel on Climate Change* [Masson-Delmotte, V., P. Zhai, A. Pirani, S.L. Connors, C. Pean, S. Berger, N. Caud, Y. Chen, L. Goldfarb, M.I. Gomis, M. Huang, K. Leitzell, E. Lonnoy, J.B.R. Matthews, T.K. Maycock, T. Waterfield, O. Yelekci, R. Yu, and B. Zhou (eds.)]. Cambridge University Press, 553–672 (2021) <https://doi.org/10.1017/9781009157896.006>

Li, D., DeConto, R.M., Pollard, D.: Climate model differences contribute deep uncertainty in future Antarctic ice loss. *Science Advances* 9(7), 7082 (2023) <https://doi.org/10.1126/sciadv.add7082>

Lowry, D.P., Krapp, M., Golledge, N.R., Alevropoulos-Borrill, A.: The influence of emissions scenarios on future Antarctic ice loss is unlikely to emerge this century. *Communications Earth & Environment* 2, 221 (2021) <https://doi.org/10.1038/s43247-021-00289-2>

Morlighem, M., Rignot, E., Binder, T., Blankenship, D., Drews, R., Eagles, G., Eisen, O., Ferraccioli, F., Forsberg, R., Fretwell, P., Goel, V., Greenbaum, J. S., Gudmundsson, H., Guo, J., Helm, V., Hofstede, C., Howat, I., Humbert, A., Jokat, W., Karlsson, N.B., Lee, W.S., Matsuoka, K., Millan, R., Mouginot, J., Paden, J., Pattyn, F., Roberts, J., Rosier, S., Ruppel, A., Seroussi, H., Smith, E. C., Steinhage, D., Sun, B., van den Broeke, M. R., van Ommen, T. D., van Wessem, M., Young, D. A.: Deep glacial troughs and stabilizing ridges unveiled beneath the margins of the Antarctic ice sheet, *Nature Geoscience* 13, 132–137 (2020) <https://doi.org/10.1038/s41561-019-0510-8>

O'Neill, B.C., Tebaldi, C., Vuuren, D.P., Eyring, V., Friedlingstein, P., Hurtt, G., Knutti, R., Kriegler, E., Lamarque, J.-F., Lowe, J., Meehl, G.A., Moss, R., Riahi, K., Sanderson, B.M.: The Scenario Model Intercomparison Project (ScenarioMIP) for CMIP6. *Geoscientific Model Development* 9(9), 3461–3482 (2016) <https://doi.org/10.5194/gmd-9-3461-2016>

Otosaka, I.N., Shepherd, A., Ivins, E.R., Schlegel, N.-J., Amory, C., Broeke, M.R., Horwath, M., Joughin, I., King, M.D., Krinner, G., Nowicki, S., Payne, A.J., Rignot, E., Scambos, T., Simon, K.M., Smith, B.E., Sørensen, L.S., Velicogna, I., Whitehouse, P.L., A, G., Agosta, C., Ahlstrøm, A.P., Blazquez, A., Colgan, W., Engdahl, M.E., Fettweis, X., Forsberg, R., Gall'ee, H., Gardner, A., Gilbert, L., Gourmelen, N., Groh, A., Gunter, B.C., Harig, C., Helm, V., Khan, S.A., Kittel, C., Konrad, H., Langen, P.L., Lecavalier, B.S., Liang, C.-C., Loomis, B.D., McMillan, M., Melini, D., Mernild, S.H., Mottram, R., Mouginot, J., Nilsson, J., Noel, B., Pattle, M.E., Peltier, W.R., Pie, N., Roca, M., Sasgen, I., Save, H.V., Seo, K.-W., Scheuchl, B., Schrama, E.J.O., Schröder, L., Simonsen, S.B., Slater, T., Spada, G., Sutterley, T.C., Vishwakarma, B.D., Wessem, J.M., Wiese, D., Wal, W., Wouters, B.: Mass balance of the Greenland and Antarctic ice sheets from 1992 to 2020. *Earth System Science Data* 15(4), 1597–1616 (2023) <https://doi.org/10.5194/essd-15-1597-2023>

Ritz, C., Edwards, T.L., Durand, G., Payne, A.J., Peyaud, V., Hindmarsh, R.C.A.: Potential sea-level rise from Antarctic ice-sheet instability constrained by observations. *Nature* 528, 115–118 (2015) <https://doi.org/10.1038/nature16147>

Riahi, K., Schaeffer, R., Arango, J., Calvin, K., Guivarch, C., Hasegawa, T., Jiang, K., Kriegler, E., Matthews, R., Peters, G. P., Rao, A., Robertson, S., Sebbit, A. M., Steinberger, J., Tavoni, M., van Vuuren, D. P. (2022). Mitigation pathways compatible with long-term goals. In IPCC, 2022: Climate Change 2022: Mitigation of Climate Change. Contribution of Working Group III to the Sixth Assessment Report of the Intergovernmental Panel on Climate Change [P.R. Shukla, J. Skea, R. Slade, A. Al Khourdajie, R. van Diemen, D. McCollum, M. Pathak, S. Some, P. Vyas, R. Fradera, M. Belkacemi, A. Hasija, G. Lisboa, S. Luz, J. Malley, (eds.)] Cambridge University Press, 295–408. <https://doi.org/10.1017/9781009157926.005>

Schoof, C.: Ice sheet grounding line dynamics: Steady states, stability, and hysteresis. *Journal of Geophysical Research: Earth Surface* 112(F3), F03S28 (2007) <https://doi.org/10.1029/2006JF000664>

Seroussi, H., Nowicki, S., Payne, A.J., Goelzer, H., Lipscomb, W.H., Abe-Ouchi, A., Agosta, C., Albrecht, T., Asay-Davis, X., Barthel, A., Calov, R., Cullather, R., Dumas, C., Galton-Fenzi, B.K., Gladstone, R., Golledge, N.R., Gregory, J.M., Greve, R., Hattermann, T.,

Hoffman, M.J., Humbert, A., Huybrechts, P., Jourdain, N.C., Kleiner, T., Larour, E., Leguy, G.R., Lowry, D.P., Little, C.M., Morlighem, M., Pattyn, F., Pelle, T., Price, S.F., Quiquet, A., Reese, R., Schlegel, N.-J., Shepherd, A., Simon, E., Smith, R.S., Straneo, F., Sun, S., Trusel, L.D., Van Breedam, J., Wal, R.S.W., Winkelmann, R., Zhao, C., Zhang, T., Zwinger, T.: ISMIP6 Antarctica: a multi-model ensemble of the Antarctic ice sheet evolution over the 21st century. *The Cryosphere* 14(9), 3033–3070 (2020) <https://doi.org/10.5194/tc-14-3033-2020>

Seroussi, H., Pelle, T., Lipscomb, W.H., Abe-Ouchi, A., Albrecht, T., Alvarez- Solas, J., Asay-Davis, X., Barre, J.-B., Berends, C.J., Bernales, J., Blasco, J., Caillet, J., Chandler, D.M., Coulon, V., Cullather, R., Dumas, C., Galton-Fenzi, B.K., Garbe, J., Gillet-Chaulet, F., Gladstone, R., Goelzer, H., Golledge, N., Greve, R., Gudmundsson, G.H., Han, H.K., Hillebrand, T.R., Hoffman, M.J., Huybrechts, P., Jourdain, N.C., Klose, A.K., Langebroek, P.M., Leguy, G.R., Lowry, D.P., Mathiot, P., Montoya, M., Morlighem, M., Nowicki, S., Pattyn, F., Payne, A.J., Quiquet, A., Reese, R., Robinson, A., Saraste, L., Simon, E.G., Sun, S., Twarog, J.P., Trusel, L.D., Urruty, B., Van Breedam, J., Wal, R.S.W., Wang, Y., Zhao, C., Zwinger, T.: Evolution of the Antarctic Ice Sheet Over the Next Three Centuries From an ISMIP6 Model Ensemble. *Earth's Future* 12(9), 2024–004561 (2024) <https://doi.org/10.1029/2024EF004561>

Seroussi, H., Verjans, V., Nowicki, S., Payne, A.J., Goelzer, H., Lipscomb, W.H., Abe-Ouchi, A., Agosta, C., Albrecht, T., Asay-Davis, X., Barthel, A., Calov, R., Cullather, R., Dumas, C., Galton-Fenzi, B.K., Gladstone, R., Golledge, N.R., Gregory, J.M., Greve, R., Hattermann, T., Hoffman, M.J., Humbert, A., Huybrechts, P., Jourdain, N.C., Kleiner, T., Larour, E., Leguy, G.R., Lowry, D.P., Little, C.M., Morlighem, M., Pattyn, F., Pelle, T., Price, S.F., Quiquet, A., Reese, R., Schlegel, N.-J., Shepherd, A., Simon, E., Smith, R.S., Straneo, F., Sun, S., Trusel, L.D., Van Breedam, J., Van Katwyk, P., Wal, R.S.W., Winkelmann, R., Zhao, C., Zhang, T., Zwinger, T.: Insights into the vulnerability of Antarctic glaciers from the ISMIP6 ice sheet model ensemble and associated uncertainty. *The Cryosphere* 17(12), 5197–5217 (2023) <https://doi.org/10.5194/tc-17-5197-2023>

Weertman, J.: Stability of the Junction of an Ice Sheet and an Ice Shelf. *Journal of Glaciology* 13(67), 3–11 (1974) <https://doi.org/10.3189/S0022143000023327>

**Response to the comments of the reviewers for the manuscript
'From short-term uncertainties to long-term certainties in the future evolution of the
Antarctic Ice Sheet'**

by V. Coulon, A.K. Klose, T. Edwards, F. Turner, F. Pattyn, and R. Winkelmann

Reviewer #1 (Remarks to the Author)

The authors have undertaken a very thorough, comprehensive and thoughtful response to all reviewers' comments, and the manuscript is now in excellent shape. The tracked change document clearly evidences substantive rewriting and reorganisation of the manuscript text, to deliver a much improved and highly robust manuscript. The figures and diagrams are also much improved.

I particularly appreciate the very considered and positive way in which the authors approached suggestions for improving the overall messaging of the manuscript. As someone who is not a modeller, I was wary of providing such suggestions, feeling slightly out of my depth(!). However, I am delighted to see how effectively the changes made by the authors have improved the manuscript. The introduction is now focused, well structured, and justifies this study and the knowledge gap it addresses very effectively. The introduction, and the closing sections of the manuscript (lines 355-469), now accurately reflect and fairly represent, the quality, depth, and novelty of the research done in this study. I can easily identify and absorb the important information provided by this important study now. If I, as a non-modeller with slightly rusty mathematical and statistical skills, can do this then the authors can be confident that they are now able to target and impact a very broad audience with their manuscript.

I also fully understand the author team's logic where they did not implement recommendations from the reviewers, and agree with their decision making where changes have not been made.

I consider it a privilege to have been able to contribute (in a minor reviewer way) to the development of this manuscript, and it was a pleasure to read the very robust and focused revised manuscript. The authors have approached the reviews in a very positive and constructive way and have every right to be proud of what they have produced. To quote reviewer 3, the work is "truly impressive", as well as now being very accessible. I look forward to using the published version of the manuscript for both research and teaching purposes.

I have one minor comment (relating to lines 366-367) – I didn't think there was 6.91 m of West Antarctic ice to lose? Is this a typo?

**Neil Ross
Newcastle University**

We are glad to read that you are happy with the revised manuscript. Thanks a lot again for the constructive comments in the previous round of revisions, which were very helpful in improving our manuscript.

As described in the Methods section, the calculation of the ice-sheet sea-level contribution in this study is based on the method proposed by Goelzer et al. (2020), which includes the water-expulsion effect (Pan et al., 2021), next to other correction terms. Consequently, sea-level contributions depend on the magnitude of bedrock changes across the entire grid. By year 3000, this water-expulsion effect can become significant due to the substantial isostatic response to changes in ice loading in West Antarctica, adding to the changes in ice volume.

Reviewer #2 (Remarks to the Author)

I appreciate the effort that the authors have put into the revisions. I'm happy that they have addressed all of my comments. I have no additional suggestions and believe the manuscript is now in good shape and ready for publication.

Thank you very much again for the very helpful comments and suggestions in the previous round of revisions.

Reviewer #3 (Remarks to the Author)

This manuscript provides a robust analysis of the future evolution of the Antarctic Ice Sheet (AIS) and its contribution to sea level rise under two contrasting emission scenarios: low-emission (SSP1-2.6) and high-emission (SSP5-8.5). Using an ensemble modeling approach within a Bayesian framework, the study addresses uncertainties in ice-climate interactions (ocean and atmosphere) and ice sheet model structure (Kori and PISM).

The findings highlight that, for projections up to 2100, uncertainties are primarily driven by ice sheet model choice and ice-ocean interactions. Beyond 2100, however, the emission scenario becomes the dominant factor. Under SSP5-8.5, the study predicts a high likelihood of significant mass loss from the West Antarctic Ice Sheet (WAIS) and destabilization of subglacial basins in the East Antarctic Ice Sheet (EAIS), such as Wilkes. In contrast, adherence to SSP1-2.6, aligned with the Paris Agreement, substantially reduces the risk of WAIS collapse and prevents irreversible EAIS loss.

The manuscript is well-written, concise, and methodologically sound. Figures are clear, and the structure—from introduction to discussion—is well-organized. This work is a valuable contribution to the field, complementing prior studies like ISMIP and extending the understanding of ice-climate interactions and SSP uncertainties. The use of IPCC-aligned uncertainty terms enhances accessibility for a broader audience.

The authors have addressed all major concerns from my previous review, particularly regarding methodology and scenario interpretation. I now consider this paper ready for publication as is, with only minor optional refinements to further polish the details.

Thank you very much for the effort in thoroughly reviewing our manuscript and providing further suggestions for refinement. We have adapted the manuscript accordingly. Please see our detailed responses below.

Minor comments:

L36-38: I agree with both parts of the sentence, but I am unsure whether the second part is a direct consequence of the first, or if that is what you intended to show. Your results indicate that SSP1-2.6, which is more optimistic than current mitigation efforts, may not be sufficient to entirely avoid self-sustained Antarctic ice loss, and this loss is significantly less likely under SSP1-2.6 compared to SSP5-8.5, which represents a more pessimistic scenario than current mitigation efforts. These are two distinct findings, yet the current wording implies that the second result is a consequence of the first. I recommend rephrasing for clarity.

Thank you for this comment. Our results show that

- (1) Antarctic ice loss can be limited under low emissions (SSP1-2.6) compared to very high emissions (SSP5-8.5), especially by avoiding substantial ice-sheet retreat in East Antarctica.
- (2) Even though ice loss may be limited under SSP1-2.6 (with stronger mitigation efforts compared to current climate policies), widespread West Antarctic ice-sheet retreat cannot be fully excluded.

To our understanding, the statement in L36-38 reflects and summarizes these key results (1) and (2) of our manuscript in that “current mitigation efforts may not be sufficient to avoid self-sustained ice loss”, which means that stronger mitigation efforts compared to current climate policies are needed (i.e. “emission decisions taken in the coming years are decisive”).

L88-89: all ice-sheet models using inverse method are calibrated with satellite observations. Perhaps you mean transient satellite observations?

Thank you for this comment. We have changed “satellite observations” to “transient satellite observations”. Note that the calibration applied here is conceptually different to data assimilation in a snapshot (e.g. Gagliardini et al., 2013; Gillet-Chaulet et al., 2016; Morlighem et al., 2013) or transient inversion (Choi et al., 2023; Badgeley et al., 2025).

Badgeley, J.A., Morlighem, M., and Seroussi, H: Increased sea-level contribution from northwestern Greenland for models that reproduce observations, Proceedings of the National Academy of Sciences, 122 (25), e2411904122, <https://doi.org/10.1073/pnas.2411904122>, 2025.

Choi, Y., Seroussi, H., Morlighem, M., Schlegel, N.-J., and Gardner, A.: Impact of time-dependent data assimilation on ice flow model initialization and projections: a case study of Kjer Glacier, Greenland, *The Cryosphere*, 17, 5499–5517, <https://doi.org/10.5194/tc-17-5499-2023>, 2023.

Gagliardini, O., Zwinger, T., Gillet-Chaulet, F., Durand, G., Favier, L., de Fleurian, B., Greve, R., Malinen, M., Martín, C., Råback, P., Ruokolainen, J., Sacchettini, M., Schäfer, M., Seddik, H., and Thies, J.: Capabilities and performance of Elmer/Ice, a new-generation ice sheet model, *Geosci. Model Dev.*, 6, 1299–1318, <https://doi.org/10.5194/gmd-6-1299-2013>, 2013.

Gillet-Chaulet, F., Durand, G., Gagliardini, O., Mosbeux, C., Mouginot, J., Rémy, F., and Ritz, C.: Assimilation of surface velocities acquired between 1996 and 2010 to constrain the form of the basal friction law under Pine Island Glacier, *Geophysical Research Letters*, 43, 10,311–10,321, <https://doi.org/10.1002/2016GL069937>, 2016.

Morlighem, M., Seroussi, H., Larour, E., and Rignot, E.: Inversion of basal friction in Antarctica using exact and incomplete adjoints of a higher-order model, *Journal of Geophysical Research: Earth Surface*, 118, 1746–1753, <https://doi.org/10.1002/jgrf.20125>, 2013.

Fig. 2: I really like the updated style of the figures! The colors are visually appealing and work well, though I hope they won't confuse readers familiar with the IPCC reports, where light blue is often used for SSP1-1.9 and light red for SSP3-7.0. As it stands, the figure is very clear and effective. For panel b, however, the names of the GCMs on the bottom axis appear a bit crowded and stacked—this might just be my perspective, but it could benefit from slight adjustments to improve readability.

Thank you for this comment. We agree that the names of the GCMs on panel b were stacked. We have adjusted the orientation of the labels to improve readability.

L289-290: It seems that the negligible structural uncertainty by 2300 may result from opposing effects between the Antarctic Peninsula (where this uncertainty is quite significant) and the West Antarctic Ice Sheet. Additionally, since there are only two ice sheet models, when the medians of the two models intersect, their contribution to the overall uncertainty effectively drops to zero. However, I suspect that this uncertainty would begin to grow again beyond 2300. This changes the conclusion of this sentence somewhat.

We understand your point and agree that structural uncertainty in the Antarctic sea-level contribution is not strictly zero and may indeed increase beyond 2300, for example, due to contrasting behaviours between the Antarctic Peninsula and the West Antarctic Ice Sheet, or when including additional ice-sheet models. However, the main aim of this section is to highlight the shift in the dominant sources of uncertainty (from ice-sheet model structure to climate forcing) for this very high warming scenario. This shift results in relatively robust agreement between the ice-sheet models in terms of multi-centennial mass loss, and is particularly evident when including the SSP scenario in the ANOVA as a source of uncertainty as shown in Supplementary Figure 3.

To further stress this key message, we have added the following sentence at the end of this paragraph: “The increasing dominance of the climate pathway over ice-sheet model structure is also particularly evident when explicitly including the emission pathways in the ANOVA (Supplementary Fig. 3).”

L295-300: Similar to my previous comment, I think it is likely coincidental that both models produce the same median at this point. The curves appear quite different, and it seems probable that beyond 2300, PISM would project higher mass loss. For example, introducing a third model would likely alter these conclusions.

Please see our response to the previous related comment.

Fig. 4: Why is the total so far from 1 at the beginning? It looks much better in Fig. 5. From my perspective, this discrepancy might arise from the ISM/atmosphere and ISM/ocean uncertainties, which could be much higher initially—especially ISM/ocean. By 2100, most of the uncertainty comes from Kori, and its uncertainty is largely driven by ocean processes. Therefore, ISM/ocean should be the dominant source of uncertainty in 2100 in Fig. 4.

After reviewing the methodology, I realized this wasn’t immediately clear. In Fig. 4, you are not showing all ocean-related uncertainties but only the common source of uncertainty explored in both models: the ice-ocean heat flux. However, given the importance of other ocean-related uncertainties in Kori, it seems likely that these would account for much of the white space in Fig. 4. It would be helpful to include all ocean-related uncertainties in Fig. 4. Even though PISM does not explore sub-shelf melt parameterization, it uses Mpico, which falls within Kori’s parameter space. This suggests it should be possible to account for this uncertainty.

Here’s a potential approach: For Kori, you have the same number of members for each parameter ($400/5 = 80$ members per parameter). For Mpico (parameter 0), you have 80 members from Kori and 300 members from PISM. You could shuffle these 380 members to create five smaller groups of 80 members each. (I realize $5 \times 80 \neq 380$, but if you instead use groups of 20 members, this could work: 19 groups of 20 members with Mpico, 4 groups of 20 members with Plume, etc.). Afterward, you could compute the variance between these groups to estimate the variance due to sub-shelf melt parameterization. Dividing this variance by the total variance would give you the contribution of this uncertainty. A similar approach could be applied for oceanic present-day climatology.

For the variance from the interaction between ISM and ice shelf parameterization, I think the value should be exactly the same: though I haven’t thought through all the details and have just made a quick sketch, it seems that by using the previous method you split the variance due to the melt parameterization half in the main effect and half in the interaction. Another way to show this uncertainty is to attribute all the variance to the interaction: you compute the variance due to melt parameterization in the Kori ensemble and divide it by the total variance of both ensembles.

Thank you for this very helpful comment. It is correct that, since Figure 4 only displays the common parameters sampled in the ensembles of both ice-sheet models, uncertainty related to the sub-shelf melt parameterisation and the present-day ocean climatology appear as unexplained variance (within the white space).

We have followed your suggestion and have included all ensemble parameters in the revised Figure 4. To do this, we use Type-I (sequential) sums of squares, which allows variance to be attributed incrementally to each source of uncertainty and their interactions and have listed 'ice-sheet model' as the first parameter. Given the constant values of the 'sub-shelf melt parameterisation' and the 'ocean present-day climatology' in the PISM ensemble, this ensures that structural (ice-sheet model) differences between Kori-ULB and PISM are accounted for before attributing variance to within-model parameter perturbations, and avoids artificial inflation of 'sub-shelf melt parameterisation' and the 'ocean present-day climatology' effects that would have been confounded with the ice-sheet model structure. Variance fractions now sum closer to 1.

Finally, I didn't see any explicit mention that these sensitivity indices are computed on the calibrated ensemble (though I assume they are). It might be helpful to clarify this in your description of the sensitivity analysis, including how you account for the weights in this process. Also, if this is the case, the method proposed above would need to be modified slightly. Instead of forming groups with the same number of members, you would create groups with the same total weight.

We thank the reviewer for pointing this out. In fact, in the previous manuscript version, the ANOVA was computed on the uncalibrated ensemble. You are correct that this should have been clarified, and that it is more consistent to apply the sensitivity analysis on the calibrated ensemble. We have revised Figures 4 and 5 and compute the variance decomposition on the calibrated ensemble using weighted linear regression (`fitlm` in MATLAB), which allows us to include the calibration weights directly. Variance contributions are obtained from Type-I (sequential) weighted sums of squares and expressed as fractions of the total weighted variance. This ensures that the sensitivity analysis is fully consistent with the calibrated projections presented in the manuscript. The Methods section has been clarified accordingly. The scripts for reproducing our ANOVA analysis will be made available on Zenodo.

We would like to stress that while including the weighting in the ANOVA shifts absolute variance fractions slightly, as expected, it does not affect our main findings.

L453: A good citation to add for sliding and subglacial hydrology: Zhao, C., Gladstone, R., Zwinger, T. et al. Subglacial water amplifies Antarctic contributions to sea-level rise. Nat Commun 16, 3187 (2025). <https://doi.org/10.1038/s41467-025-58375-4>

Thank you for pointing out this reference. We have included it in the Discussion.

L466-469: Like in the abstract, I'm not sure I understand why the second part of the sentence is presented as a consequence of the first part. To me, the second part

seems to be a consequence of the lower likelihood of extreme sea level rise under SSP1-2.6.

Please see our response to the previous related comment.

Methods:

L518-519: I'm a bit confused about the q value. Does it depend on your spin-up calibration (L536-538)? In the end, do you obtain a single q value for all members, or does it vary between members based on the scoring described in L537?

Full-physics spin-up ensembles varying key model parameters including the basal sliding exponent q are run for the MAR and RACMO-based historical atmospheric climatologies, and the ensemble members performing well in the applied scoring are chosen for projecting the Antarctic ice-sheet evolution with PISM. The chosen initial ice-sheet states for the MAR- and RACMO-based climatology have distinct basal sliding exponents q . Likewise, for each historical atmospheric climatology, initial ice-sheet states with Kori-ULB are generated (based on transient inverse simulations), and are associated with respective basal friction fields. This has been clarified in the revised manuscript.

L592-593: I don't quite understand how natural variability is accounted for. Do you vary the temperature using a Gaussian distribution with a variance of 4°C and 5°C ?

The PDD approach assumes a daily temperature variability in the form of "white noise" with a standard deviation (as specified in the manuscript) around a monthly mean atmospheric temperature.

L666: Fortunately, the CRPS decreases across the entire domain, as expected since the data was used for calibration! It would have been a major red flag if that weren't the case. That said, while it doesn't always decrease at the regional scale, it does in most cases, which is a good sign for robustness.

Thanks a lot again for suggesting the CPRS to assess the effect of the calibration on our ensemble in your previous comments. This has been very helpful to improve our manuscript.

Hope it helps, and good luck with your revisions!

Eliot Jager, University of Helsinki

Summary:

This article examines the future of the Antarctic Ice Sheet (AIS) and its impact on sea level rise using two ensemble models calibrated within a Bayesian framework for two extreme emission scenarios: low emission (SSP1-2.6) and very-high emission (SSP5-8.5). Each ensemble addresses a broad spectrum of uncertainties associated with ice-climate interactions (ocean and atmosphere). The findings indicate that uncertainties in historical and short to long-term projections (up to 2100) predominantly stem from the selection of the ice sheet model (Kori or PISM), ice-climate interactions, and the climate model (AOGCM). In contrast, over multi-centennial to millennium timescale (from 2100 to 3000), the chosen emission scenario emerges as the most significant uncertainty factor. Notably, under the extremely high emission scenario (SSP5-8.5), there is a high likelihood of substantial mass loss from the West Antarctic Ice Sheet (WAIS) and a significant probability of loss from certain subglacial basins of the East Antarctic Ice Sheet (EAIS), such as Wilkes. Conversely, adhering to the Paris Agreement and limiting warming to 2 degrees (SSP1-2.6) considerably reduces the likelihood of WAIS loss and prevents self-sustained ice loss from the EAIS.

I enjoyed reading this manuscript, which is well-written and succinct! The figures are clear, as are the introduction, methods, results, and discussion sections. I believe this is an important paper and serves as a valuable complement to the ISMIP work on the AIS. It perfectly complements Seroussi et al. (2024) by more thoroughly exploring ice-climate interactions and SSP uncertainties. The goal of reaching a broader audience has been very successfully achieved by using terms characterizing uncertainties similar to those used by the IPCC (very likely, virtually certain, etc.).

Major comments:

I really appreciate the work of this team, and this article seems like an excellent culmination of their previous research. However, I am somewhat uncertain about the added value of this paper, as it primarily appears to summarize the findings for a broader audience from Klose et al. (2024). The main distinction seems to be the inclusion of the Bayesian calibration proposed in Coulon et al. (2024), which significantly affects the results for Kori due to differences in the prior ensemble compared to Klose et al. (2024), but has minimal impact on the PISM results. Additionally, there seems to be insufficient evidence demonstrating the superior performance of the posterior ensemble over the prior

ensemble during the observation period. Figure A1 indicates a significant reduction in uncertainty for the Kori posterior ensemble, yet this is not observed for PISM, and the posterior median occasionally falls outside the observational range for both models.

To enhance the paper and further distinguish it from Klose et al. (2024), I suggest two modifications:

1. **Rewrite the section “Role of uncertainties in modulating future Antarctic mass loss”:** Your ensemble adds substantial value to the findings of Seroussi et al. 2023 and 2024 by delving deeper into ice-climate interactions and SSP uncertainty. It would be beneficial to include a figure showing the time evolution of Sobol indices/percentage of variance (similar to Fig. 9 in Seroussi et al. 2023, Fig. 15 in Seroussi et al. 2024, and Fig. 6 in Jager et al. 2024). This would provide strong visual support for this section and clarify which sources of uncertainty are most significant up to the year 2300. Additionally, a new paragraph comparing this with Fig. 15 from Seroussi et al. 2024 could emphasize the importance of considering uncertainty from ice-climate interactions and SSP.
2. **Demonstrate the improvement of the ensemble through Bayesian calibration:** In Jager et al. 2024, I proposed a methodology to determine whether the posterior ensemble is better than the prior ensemble. Similar to practices in machine learning and hydrology, it involves splitting your observation dataset into a training set and a testing set. I believe this method could be easily applied to your study, but if you have another approach to demonstrate the benefits of Bayesian calibration, I would be delighted to learn about it. Here is a suggestion for implementing it:
 - a. Split the ensemble into a training set and a testing set. Given that you have 18 observations across 3 sectors, consider using observations from only 2 sectors (5 per sector) as the training set, with the remainder as the testing set.
 - b. Use the training set for Bayesian calibration to determine your weights.
 - c. Evaluate the performance of the posterior ensemble against the prior one using the testing set. Various metrics, such as the continuous rank probability score (CRPS) from Eq. 3 in Jager et al. 2024, can be used for this comparison. Additional metrics are available here: [\[https://www.cawcr.gov.au/projects/verification/\]](https://www.cawcr.gov.au/projects/verification/).
 - d. Restart with another training and testing set.

Show the improvement of the ensemble through Bayesian calibration. In Jager et al. 2024, I’ve proposed a methodology to check if the posterior ensemble is better than the prior

ensemble, similar to what is done in machine learning or in hydrology. I think it can easily be applied to your study, but if you think of another way to show the benefit of Bayesian calibration I'd be delighted. Here is a proposition for doing it:

I've noticed that some statements in the paper may not fully align with the widely accepted understanding of SSPs, emissions, and mitigation efforts, or at least differ from my personal understanding. This issue seems to extend beyond your study and reflects a broader trend in our community, where SSP5-8.5 is still often treated as a 'business-as-usual' scenario, which it is not. According to the latest IPCC Summary for Policymakers (2022), the policies currently implemented are projected to lead to a warming of about 3.2°C (ranging between 2.2°C and 3.5°C, with medium confidence). This projection is slightly higher than SSP2-4.5 (around 2.8°C) but lower than SSP3-7.0 (around 3.9°C), and significantly lower than SSP5-8.5 (around 4.8°C). Other literature also suggests that we are not currently on a trajectory consistent with SSP5-8.5 (Raftery et al., 2017; Hausfather and Peters, 2020; Hausfather and Moore, 2022; Pielke et al., 2022). Given this context, you have two options:

1. **Perform new simulations with an intermediate SSP such as SSP2-4.5:** This approach would allow you to explore whether current policies are sufficient to prevent significant contributions from the AIS, or if more aggressive policies are needed. It also provides valuable information to stakeholders by representing a 'most probable/likely path', serving as a median path between two less likely extremes.
2. **Revise some of your statements downwards:** While this option is simpler as it does not require running new simulations (and I understand that there are limited climate models that have run until 2300 with SSP2-4.5, though some exist from CMIP5 with RCP4.5, e.g., Hezel et al. 2014), it would significantly reduce the impact of your paper. Specifically, it would necessitate revising the last high-impact sentence in the abstract: 'Our results imply that strong mitigation efforts are needed in the coming years to decades to reduce the risk of self-sustained sea-level rise over the next centuries.'" In this second case, it would also be appropriate to add a paragraph to discuss this limitation of the study in the 'Next decades decisive for Antarctica's long-term future' section. This would be an opportunity to push for studies with better sampling of SSPs, which seem to be the main source of uncertainty after 2100.

I hope these suggestions seem relevant to you and will help improve the manuscript. My major comments may be somewhat lengthy, but I would like to reiterate my congratulations on this excellent article, which is truly impressive. I have cited my previous

work, but that doesn't necessarily mean you must cite it as well. I've also noted some minor corrections below, and at the end, you'll find the references I've mentioned. Hope it helps, and good luck with your revisions!

Eliot Jager, University of Helsinki

Minor comments:

L39-40: Without results with SSP2-4.5, the sentence is overstated. SSP2-4.5 is currently the path closest to 'business as usual'. This does not mean that other paths are not possible, but 'strong mitigation efforts' today aim to shift from a trajectory of +3°C (SSP2-4.5) to 2°C (SSP1-2.6), not from +5°C (SSP5-8.5).

L54: you can cite Aschwanden et al. (2021) here.

L59: Would Seroussi et al. (2024) be a more recent/relevant reference here?

L64: It's a bit odd here to cite the IPCC report, as it does not produce projections, and the fact that one section in this report is devoted to assessing 'Projections Beyond 2100.' The two other references are sufficient.

L67-68: I'm not sure which reference the term "multi-centennial timescales" comes from. In Hinkel et al. (2019), 'long-term' is defined as "longer than 30 years," with a typical horizon of "30 to 100 years and more." Nuclear plants can be considered, but their typical lifetime is less than 200 years. Nichols et al. (2021) mention post-2100 but do not provide a typical range. In Wal et al. (2022), they indeed assess up to 2300, but I'm not convinced that many stakeholders in coastal adaptation planning are interested in such a timescale. While there are certainly some, italicizing it gives the impression that most, if not all, require such information, whereas they are primarily interested in the short term.

L69-70: I prefer the term "ice sheet model structure" over "ice-sheet model" to distinguish it from "parametric uncertainty."

L142-143: I agree in terms of the median, but isn't there a significant difference in the 90% confidence interval? I'm not completely certain about this; Graph 2.a is not very clear. Could you add different boxplots of your results next to E21? And/or add hatching to the shading?

Fig. 4: It is difficult to compare panels c and d due to the difference in units.

L217-222: I am quite convinced by the explanation, but it also appears that the PICO-SSP-2.6 members from Kori exhibit a positive correlation, or at least the trend is not flat (as indicated by the empty diamond in Fig. 4.a).

Section "Certain long-term future of the Antarctic Ice Sheet": Please define "long-term" somewhere. I suggest using "very long-term" or "multi-centennial timescale" for a 2500 horizon and "millennium timescale" for a 3000 horizon.

L320: "Under high warming" is not the most accurate description for SSP5-8.5. "Excessively high warming," "low-probability high warming," or "very high warming" might be more precise. I believe "very high warming" is the term used in the latest IPCC report.

L326-327: In your sentence, you state that the "onset of collapse and future rates" depend mainly on "model uncertainties," but they also depend significantly on the future level of warming/human emissions! You might refer to this uncertainty as "uncertainties in future human emissions."

L342: "in Antarctic viscoelastic properties" -> do you mean the mantle of the AIS?

L345: "ice-sheet model uncertainties" -> here, I think you are referring to the model structure, i.e., the uncertainty arising from using two different ice sheet models, not the parametric uncertainty of each. I recall pointing this out elsewhere; please check if other parts of the text need revision for consistency and use the term "structure." It doesn't have much to do with this comment, and there's no definition of 'model structure', but on these questions of sensitivity analysis this ebook is very cool: <https://uc-ebook.org/>

L351-352: "unless massive mitigation efforts are made" -> it would be necessary to include results with SSP2-4.5 to assess this. Perhaps this pathway alone is sufficient to prevent "substantial long-term Antarctic changes," without the need for massive mitigation efforts.

L354-355: "Unmitigated emissions" more closely aligns with SSP2-4.5, not SSP5-8.5. Discussions for CMIP7 seem to suggest that this scenario might not be included in CMIP7 (e.g., van Vuuren et al. (2025)), or it could be referred to as "the emissions world avoided" (TEWA) scenario (Malte Meinshausen et al. (2024)).

L358-359: Same here, past years have been decisive to pass from SSP5-8.5 to SSP2-4.5, but is it enough? I think this is a really interesting question, but would require to test at least SSP2-4.5.

"Ice-sheet models" section: It would be beneficial to include a table with one column for each model to provide a quick overview of the common and differing characteristics of the two models (take a look to ISMIP papers for examples).

L404: What is the exponent of the power-law? Additionally, it's not clear which specific equation from the cited paper is being used here.

L493-494: The sum of 400 and 300 differs from the 1400 members mentioned earlier. Please clarify that you create one ensemble for each SSP.

“Calibration” section:

- Do you apply a weighting to each model, or do you combine members from both models?
- Does it make sense to have the same structural model error, σ_i^{mod} , for both model? Perhaps one model is inherently better? This can be assessed using the method I proposed in my major comment.
- L524: Independence between observations is a condition that is rarely fully met in glaciology (if one catchment is losing mass, it is likely that the adjacent one is also losing mass and will continue to do so in the near future), because our systems are not chaotic.
- Fig. A1: It appears strange that your median is further from the observation after calibration.
- In Jager et al. (2024), we found that σ_i has a significant influence, and can lead to either underfitting or overfitting. "Perhaps here, taking σ_i^{mod} as 10 times the observational error σ_i^{obs} is too large and leads to under-fitting. This can be assessed using the method I've proposed in my major comment.

Fig. S2: - Isn't it strange that MAR appears to be a better choice for Kori, but RACMO is more suitable for PISM?

- g vs h: Are you certain of your values? There is a significant difference in the order of magnitude between the 2 plots.

Table S2: For PISM, there are significant changes between the prior and posterior ensembles, yet when I look at Fig. A1.b, the prior and posterior ensembles seem to have similar performance during the historical period. Are you sure you are not reducing the range for the wrong reasons?

Fig. A1: - Can you plot the same charts as in a and b but for the different sectors? I still find it strange that the calibration for PISM misses some observations in the shading and that the median of the posterior is further from the observation than the prior for Kori.

- Can you add a map of observed elevation rates for visual comparison? e.g. Smith et al. 2020

References:

Seroussi, H., Pelle, T., Lipscomb, W. H., Abe-Ouchi, A., Albrecht, T., Alvarez-Solas, J., et al. (2024). Evolution of the Antarctic Ice Sheet over the next three centuries from an ISMIP6 model ensemble. *Earth's Future*, 12, e2024EF004561.

<https://doi.org/10.1029/2024EF004561>

Klose, A. K., Coulon, V., Pattyn, F., and Winkelmann, R.: The long-term sea-level commitment from Antarctica, *The Cryosphere*, 18, 4463–4492, <https://doi.org/10.5194/tc-18-4463-2024>, 2024.

Coulon, V., Klose, A. K., Kittel, C., Edwards, T., Turner, F., Winkelmann, R., and Pattyn, F.: Disentangling the drivers of future Antarctic ice loss with a historically calibrated ice-sheet model, *The Cryosphere*, 18, 653–681, <https://doi.org/10.5194/tc-18-653-2024>, 2024.

Seroussi, H., Verjans, V., Nowicki, S., Payne, A. J., Goelzer, H., Lipscomb, W. H., Abe-Ouchi, A., Agosta, C., Albrecht, T., Asay-Davis, X., Barthel, A., Calov, R., Cullather, R., Dumas, C., Galton-Fenzi, B. K., Gladstone, R., Golledge, N. R., Gregory, J. M., Greve, R., Hattermann, T., Hoffman, M. J., Humbert, A., Huybrechts, P., Jourdain, N. C., Kleiner, T., Larour, E., Leguy, G. R., Lowry, D. P., Little, C. M., Morlighem, M., Pattyn, F., Pelle, T., Price, S. F., Quiquet, A., Reese, R., Schlegel, N.-J., Shepherd, A., Simon, E., Smith, R. S., Straneo, F., Sun, S., Trusel, L. D., Van Breedam, J., Van Katwyk, P., van de Wal, R. S. W., Winkelmann, R., Zhao, C., Zhang, T., and Zwinger, T.: Insights into the vulnerability of Antarctic glaciers from the ISMIP6 ice sheet model ensemble and associated uncertainty, *The Cryosphere*, 17, 5197–5217, <https://doi.org/10.5194/tc-17-5197-2023>, 2023.

Jager, E., Gillet-Chaulet, F., Champollion, N., Millan, R., Goelzer, H., and Mouginit, J.: The future of Upernavik Isstrøm through the ISMIP6 framework: sensitivity analysis and Bayesian calibration of ensemble prediction, *The Cryosphere*, 18, 5519–5550, <https://doi.org/10.5194/tc-18-5519-2024>, 2024.

IPCC, 2023: Summary for Policymakers. In: *Climate Change 2023: Synthesis Report. Contribution of Working Groups I, II and III to the Sixth Assessment Report of the Intergovernmental Panel on Climate Change* [Core Writing Team, H. Lee and J. Romero (eds.)]. IPCC, Geneva, Switzerland, pp. 1-34, doi: 10.59327/IPCC/AR6-9789291691647.001

Raftery, A. E., Zimmer, A., Frierson, D. M. W., Startz, R., and Liu, P.: Less than 2 °C warming by 2100 unlikely, *Nat. Clim. Change*, 7, 637–641, <https://doi.org/10.1038/nclimate3352>, 2017.

Hausfather, Z. and Moore, F. C.: Net-zero commitments could limit warming to below 2 °C, *Nature*, 604, 247–248, <https://doi.org/10.1038/d41586-022-00874-1>, 2022.

Hausfather, Z. and Peters, G.: Emissions – the “business as usual” story is misleading, *Nature*, 577, 618–620, <https://doi.org/10.1038/d41586-020-00177-3>, 2020.

Pielke Jr, R., Burgess, M. G., and Ritchie, J.: Plausible 2005–2050 emissions scenarios project between 2 °C and 3 °C of warming by 2100, *Environ. Res. Lett.*, 17, 024027, <https://doi.org/10.1088/1748-9326/ac4ebf>, 2022.

Hezel, P. J., Fichefet, T., and Massonnet, F.: Modeled Arctic sea ice evolution through 2300 in CMIP5 extended RCPs, *The Cryosphere*, 8, 1195–1204, <https://doi.org/10.5194/tc-8-1195-2014>, 2014.

Aschwanden, A., Bartholomaus, T. C., Brinkerhoff, D. J., and Truffer, M.: Brief communication: A roadmap towards credible projections of ice sheet contribution to sea level, *The Cryosphere*, 15, 5705–5715, <https://doi.org/10.5194/tc-15-5705-2021>, 2021.

Hill, E. A., Urruty, B., Reese, R., Garbe, J., Gagliardini, O., Durand, G., Gillet-Chaulet, F., Gudmundsson, G. H., Winkelmann, R., Chekki, M., Chandler, D., and Langebroek, P. M.: The stability of present-day Antarctic grounding lines – Part 1: No indication of marine ice sheet instability in the current geometry, *The Cryosphere*, 17, 3739–3759, <https://doi.org/10.5194/tc-17-3739-2023>, 2023.

van Vuuren, D., O'Neill, B., Tebaldi, C., Chini, L., Friedlingstein, P., Hasegawa, T., Riahi, K., Sanderson, B., Govindasamy, B., Bauer, N., Eyring, V., Fall, C., Frieler, K., Gidden, M., Gohar, L., Jones, A., King, A., Knutti, R., Kriegler, E., Lawrence, P., Lennard, C., Lowe, J., Mathison, C., Mehmood, S., Prado, L., Zhang, Q., Rose, S., Ruane, A., Schleussner, C.-F., Seferian, R., Sillmann, J., Smith, C., Sörensson, A., Panickal, S., Tachiiri, K., Vaughan, N., Vishwanathan, S., Yokohata, T., and Ziehn, T.: The Scenario Model Intercomparison Project for CMIP7 (ScenarioMIP-CMIP7), *EGUsphere* [preprint], <https://doi.org/10.5194/egusphere-2024-3765>, 2025.

Meinshausen, M., Schleussner, C.-F., Beyer, K., Bodeker, G., Boucher, O., Canadell, J. G., Daniel, J. S., Diongue-Niang, A., Driouech, F., Fischer, E., Forster, P., Grose, M., Hansen, G., Hausfather, Z., Ilyina, T., Kikstra, J. S., Kimutai, J., King, A. D., Lee, J.-Y., Lennard, C., Lissner, T., Nauels, A., Peters, G. P., Pirani, A., Plattner, G.-K., Pörtner, H., Rogelj, J., Rojas, M., Roy, J., Samset, B. H., Sanderson, B. M., Séférian, R., Seneviratne, S., Smith, C. J., Szopa, S., Thomas, A., Urge-Vorsatz, D., Velders, G. J. M., Yokohata, T., Ziehn, T., and

Nicholls, Z.: A perspective on the next generation of Earth system model scenarios: towards representative emission pathways (REPs), *Geosci. Model Dev.*, 17, 4533–4559, <https://doi.org/10.5194/gmd-17-4533-2024>, 2024.

Ben Smith *et al.*, Pervasive ice sheet mass loss reflects competing ocean and atmosphere processes. *Science* **368**,1239-1242(2020). DOI:[10.1126/science.aaz5845](https://doi.org/10.1126/science.aaz5845)